# DiffKANformer: Diffusion KAN Transformer for General Time Series Analysis

## Abstract

Time series analysis tasks such as forecasting, imputation, anomaly detection, and classification are crucial for applications spanning climate science, financial domain, retail, and cloud infrastructure. We present DiffKANformer, a conditional diffusion model that integrates Kolmogorov-Arnold Networks (KAN) for feature projection and a Diffusion KAN Transformer architecture for denoising, specifically engineered for time series analysis. DiffKANformer introduces two key innovations: *(i)* a KAN-based projection mechanism in the forward diffusion process that captures complex correlation between features, and *(ii)* a Diffusion KAN Transformer architecture that effectively models complex long-term dependencies through adaptive univariate functions. Our model achieves superior performance across four fundamental time series analysis tasks, significantly outperforming existing prominent models in forecasting (eight datasets), imputation (six datasets), classification (ten datasets) and anomaly detection (five datasets). Comprehensive ablation studies across all tasks validate the utility of each DiffKANformer component, demonstrating the model's robustness in diverse time series challenges.

## 1 Introduction

Time series analysis stands at the heart of countless critical applications across diverse domains. From financial market forecasting (Kim, 2003) and economic modeling (Henrique et al., 2019) to transportation planning (Huang et al., 2023), energy management (Dumas et al., 2022), and climate prediction (Li et al., 2024a; Rasul et al., 2022), the ability to understand temporal patterns drives decision-making in our most vital systems. Beyond forecasting, time series analysis enables missing data imputation in domains such as data mining (Friedman, 1962), IT infrastructure monitoring (Qu et al., 2024), satellite telemetry, oil and gas operations, and manufacturing (Zhan et al., 2021). Other critical tasks include anomaly detection for industrial maintenance (Xu et al., 2021) and time series classification for applications such as trajectory-based action recognition (Franceschi et al., 2019).

The field of time series analysis has evolved dramatically from conventional statistical methods such as ARIMA and state-space models to sophisticated deep neural network architectures including NLinear (Zeng et al., 2023), recurrent neural networks (Hewamalage et al., 2021), convolutional neural networks (Yue et al., 2022), and transformers (Vaswani et al., 2017). Given its immense practical value, time series analysis has attracted substantial research attention from deep learning researchers (Wen et al., 2022; Lim & Zohren, 2021). Appendix B has a review of major time series models.

Recently, diffusion models have emerged as a powerful paradigm in generative modeling, achieving unprecedented performance across diverse domains. Their remarkable success in image synthesis (Ho et al., 2020; Dhariwal & Nichol, 2021b), video generation (Harvey et al., 2022; Blattmann et al., 2023), and cross-modal applications (Saharia et al., 2022) has naturally sparked interest in applying their generative power to time series analysis. This has led to several promising conditional diffusion-based frameworks designed specifically for time series forecasting (Tashiro et al., 2021; Li et al., 2024b; Cao et al., 2024; Shen & Kwok, 2023; Shen et al., 2024; Lopez Alcaraz & Strodthoff, 2023). These conditional diffusion models rely on sophisticated conditioning networks and employ various architectural advances to model multi-resolution and nonlinear relations inherent in time series data. For instance, TimeDiff (Shen & Kwok, 2023) incorporates future mixup conditioning for forecasting, TimeGrad (Rasul et al., 2021) combines diffusion with RNN hidden states as condition-

ing information, CSDI (Tashiro et al., 2021) employs self-supervised masking to guide conditioning for non-autoregressive imputation, and CnDiff (Rishi et al., 2025) adapts the Diffusion Transformer (DiT) architecture with specialized conditioning networks for forecasting. However, despite their effectiveness and sophisticated conditioning mechanisms, these methods exhibit two critical limitations: they fail to fully exploit the inherent properties of diffusion models, and none provides a unified solution capable of addressing all aspects of time series analysis.

A deeper examination reveals that current diffusion-based time series models predominantly rely on transformer architectures, which have become popular in computer vision (Dosovitskiy et al., 2020a) and natural language processing (Vaswani et al., 2017). Although extensive research has explored alternatives to attention mechanisms (Liu et al., 2021; 2022d), these variants still use multilayer perceptrons (MLPs) as their core computational building blocks. Surprisingly, relatively few efforts (Shazeer, 2020) have focused on enhancing MLPs themselves, despite their fundamental role.

Addressing this issue is critical for time series modeling. MLPs, while theoretically capable of approximating any function if they have enough neurons (Hornik et al., 1989), encounter practical difficulties with complex temporal patterns. Specifically, MLPs that utilize ReLU-like activations have trouble capturing the periodic functions (Yu et al., 2024; Yang & Wang, 2024) that are common in time series tasks. Additionally, when gradient descent is applied to MLPs, it converges slowly for high-frequency components (Rahaman et al., 2019; Basri et al., 2020), which are widespread in real-world time series data.

To address these architectural limitations, we turn to Kolmogorov-Arnold Networks (KANs), which have recently emerged as a powerful alternative to traditional MLPs. KANs offer superior theoretical parameter efficiency, requiring fewer parameters to model complex functions (Liu et al., 2024c), and demonstrate particular strength in mathematical and symbolic regression tasks (Yu et al., 2024; Liu et al., 2024b). Their key innovation lies in the learnable basis functions employed at each neuron, typically parameterized by B-spline curves (Unser et al., 2002; Gordon & Riesenfeld, 1974). This design enables KANs to approximate intricate functions through spline basis summation, providing enhanced expressiveness for capturing complex temporal dynamics that traditional MLPs struggle to represent.

Building upon these insights, this paper introduces the DiffKANformer architecture that uses a Diffusion Kolmogorov-Arnold Transformer model with a learnable KAN-based data projection in the forward process. These modifications reduce the gap between the true negative log-likelihood and its variational approximation, leading to more effective time series modeling across diverse tasks. The main contributions of this paper are as follows.

1. We introduce the Diffusion KAN transformer framework for time series analysis, which effectively captures long-term dependencies by utilizing adaptive univariate functions. Integrating a KAN-based projection within the forward diffusion process helps to learn complex dependencies between features for enhanced expressiveness in the diffusion process.

2. Experimental results demonstrate that, on average, our model exceeds the performance of other leading time series models across all tasks at the time of writing.

3. This is the first diffusion-based model that shows superior performance in all time series analysis tasks, providing a unified solution for forecasting, imputation, classification, and anomaly detection.

## 2 METHODOLOGY

Consider a time series represented as $x_{0:H} = \{x_0, x_1, x_2, \ldots, x_H\}$, where each vector $x_i$ consists of one or more variables that can exhibit significant correlations at time $i$ Liu et al. (2023a). This time series can be applied to several tasks. In forecasting, the objective is to predict the future series $x_{H:H+T}$ based on observed data $x_{0:H}$. In the imputation task, certain data points in $x_{0:H}$ might be absent, represented as $\tilde{x}_{0:H} = \{\tilde{x}_0, \tilde{x}_1, \ldots, \tilde{x}_H\}$. The corresponding mask vector $m_{0:H} = \{m_0, m_1, \ldots, m_H\}$ indicates which entries are observed ($m_i = 1$ if observed, 0 if not). The time series imputation task is to estimate these missing values in $\tilde{x}_{0:H}$ using the available data and the mask, generating a complete series $\hat{x}_{0:H}$ that closely mirrors the original $x_{0:H}$. Time series classification involves categorizing multiple series $x_{0:H}$ into defined classes. Anomaly detection in time series data focuses on identifying unusual or unexpected patterns within the series. We define

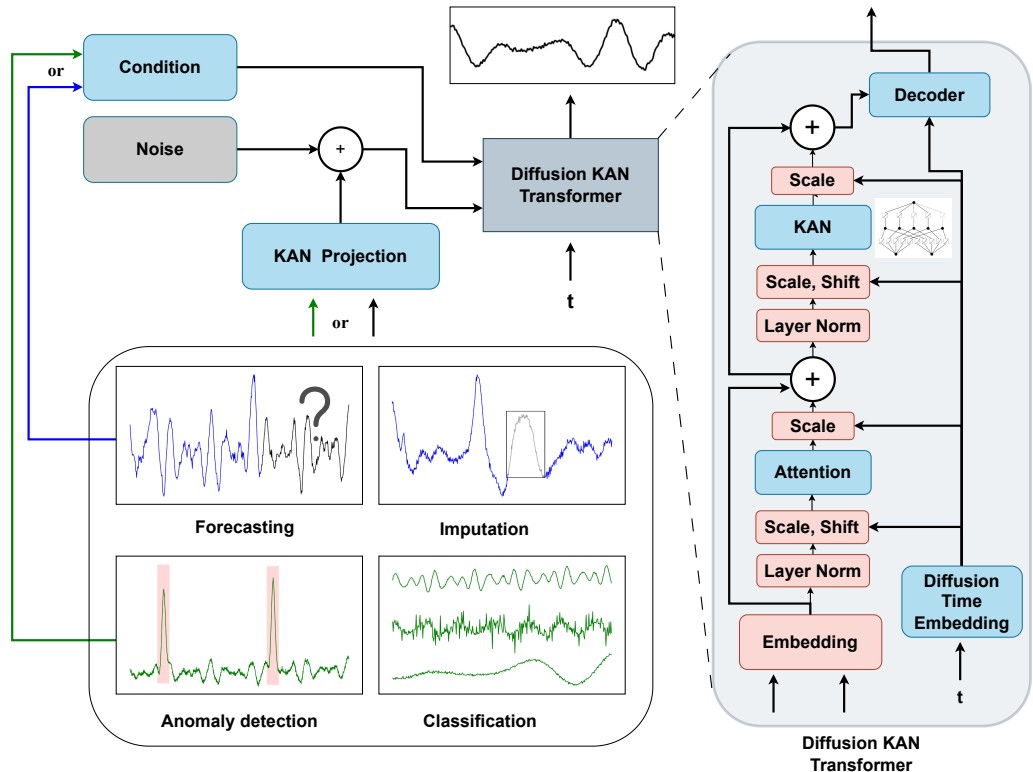

Figure 1: DiffKANformer integrates KAN projections with diffusion KAN transformers for various time series analysis. The architecture handles four key tasks through conditional inputs: forecasting (conditioning on past observations with noise added to future predictions), imputation (conditioning on masked inputs with noise applied only to masked regions), anomaly detection (learning robust representations for anomaly score), and classification (utilizing learned representations that are sent to a classification head). The core Diffusion KAN Transformer processes noised inputs through attention mechanisms, layer normalization, and embedding layers, enabling unified multi-task learning for time series analysis tasks.

output of our model as $x_{output}$, where $x_{output}$ is the future values to be predicted in the forecasting task, missing values to be predicted in the imputation task and the representation that is fed into the classification head and anomaly detection scores for those tasks. For ease of notation, we replace $x_{output}$ as **x**.

Current conditional diffusion models for time series, where the condition is past observations for forecasting, masked time series for imputation, and full series for anomaly detection and classification, convert these correlated data distributions into an isotropic Gaussian prior by adjusting data points and progressively adding fixed linear Gaussian noise to generate latent variables (Appendix C.1 and C.2). This method presents two issues: first, it confines diffusion models during the forward process, rendering them fixed and untrainable, and second, the MLP in the denoiser architecture won't be able to capture correlations between highly nonlinear complex relations between time series (Han et al., 2024b).

To address these challenges, we propose DiffKANformer, a nonlinear KAN-based projection framework that learns time-dependent distributions in the latent space during the forward process and employs the Diffusion KAN transformer block as the denoising architecture. Firstly, we discuss our diffusion loss formulation with KAN projection and subsequently delve into the diffusion KAN transformer architecture.

## 2.1 KAN DIFF FORMULATION AND OBJECTIVE

Let us define the nonlinear time-dependent KAN projection of data for marginal distributions as

$$q_\phi(\mathbf{x}^t|\mathbf{x}) = \mathcal{N}\left(\mathbf{x}^t; \sqrt{\bar{\alpha}_t}KAN_\phi(\mathbf{x}, t), (1 - \bar{\alpha}_t)I\right),$$

where $KAN_\phi(\mathbf{x}, t) : \mathbb{R}^{d \times H} \times [0, T] \mapsto \mathbb{R}^{d \times H}$ is a nonlinear KAN (Appendix C.3) parameterized by $\phi$ that applies a time-dependent projection to the data point $\mathbf{x}$ which helps to learn complex correlations between time series data. We now introduce a learnable condition to the forward process, denoted $c$ [$c$ = ConditionNetwork(input), where input varies by task], inspired by a similar formulation used in image diffusion Pandey et al. (2022). Consequently, the marginal distribution is

$$q_\phi(\mathbf{x}^t|\mathbf{x}, c) = \mathcal{N}(\mathbf{x}^t; \sqrt{\bar{\alpha}_t}KAN_\phi(\mathbf{x}, t) + (1 - \sqrt{\bar{\alpha}_t})c, (1 - \bar{\alpha}_t)I). \tag{1}$$

For $t = T$ along with an appropriately regulated noise schedule $\alpha_t$, $\bar{\alpha}_T \approx 0$ results in $q_\phi(\mathbf{x}^T|\mathbf{x}, c) \approx \mathcal{N}(c, I)$. Simply put, the Gaussian $\mathcal{N}(c, I)$ serves as our learnable prior distribution and inference requires executing our reverse process on it.

We get the following posterior distribution $\left(q_\phi(\mathbf{x}^{t-1}|\mathbf{x}^t, \mathbf{x}, c)\right)$ that satisfies Eq.(1) (Appendix D.1):

$$\mathbf{x}^{t-1} = \zeta_1\mathbf{x}^t + \zeta_2 c - KAN_\phi(\mathbf{x}, t)\zeta_1\sqrt{\bar{\alpha}_t} + KAN_\phi(\mathbf{x}, t-1)(\zeta_1\sqrt{\bar{\alpha}_t} + \zeta_0) + \sigma_{t-1|t}^2\epsilon$$

Consequently, the corresponding loss formulation of DiffKANformer is represented by the following (Appendix D.2).

$$\mathcal{L}_{\text{DiffKANformer}} = \mathbb{E}_{q_\phi}\left[\underbrace{D_{\text{KL}}\left(q_\phi\left(\mathbf{x}^T|\mathbf{x}, c\right) \| p\left(\mathbf{x}^T|c\right)\right)}_{\mathcal{L}_{\text{prior}}} - \underbrace{\sum_{t=1}^{T}\log p_\theta\left(\mathbf{x}|\mathbf{x}^t, c\right)}_{\mathcal{L}_{\text{rec}}}\right.$$
$$\left. + \underbrace{\sum_{t=1}^{T}D_{\text{KL}}\left(q_\phi\left(\mathbf{x}^{t-1}|\mathbf{x}^t, \mathbf{x}, c\right) \| p_\theta\left(\mathbf{x}^{t-1}|\mathbf{x}^t, c\right)\right)}_{\mathcal{L}_{\text{diff}}}\right]. \tag{2}$$

The details of each loss term and the detailed derivation are provided in the Appendix D.3.

## 2.2 DIFFUSION KAN TRANSFORMER

Beyond modifying the diffusion formulation through KAN projection, we fundamentally redesign the denoising architecture to better capture temporal dependencies in time series data. Traditional diffusion models employ convolutional U-Net architectures as denoisers (Ho et al., 2020), with various architectural modifications proposed, including adaptive normalization layers for conditional information injection Dhariwal & Nichol (2021a); Perez et al. (2018). Recently, transformer architectures have gained prominence in diffusion models through the introduction of Diffusion Transformers (DiT) Peebles & Xie (2023); Esser et al. (2024), demonstrating superior scalability and performance in computer vision tasks.

However, standard transformer architectures face critical limitations when applied to complex time series data Han et al. (2024a). At their core, transformers rely on two fundamental components: attention mechanisms and multi-layer perceptrons (MLPs). Although extensive research has explored alternatives to conventional attention mechanisms Liu et al. (2021); Wang et al. (2021), these variants predominantly depend on traditional MLPs as their computational backbone. This dependence becomes problematic for time series modeling, where MLPs face several inherent limitations despite their theoretical universal approximation capabilities Hornik et al. (1989).

Interestingly, comparatively few efforts are made to improve MLPs themselves in denoising architectures (Shazeer, 2020). We developed the Diffusion KAN Transformer (Figure 1), where we change the MLP in DiT with KAN along with the adaptive layer norm (adaLN) block (Peebles & Xie, 2023) where we replace the standard layer in the Diffusion KAN transformer blocks with the adaptive layer norm. Our scale and shift operations learn dimension-wise scale and shift parameters to provide fine-grained control over feature magnitudes and enable more stable training dynamics. Appendix E has more details with equations for our diffusion KAN transformer. The procedure for both training and inference is detailed in Appendix F.

## 3 EXPERIMENTS

We conducted extensive experiments to evaluate the performance of DiffKANformer on forecasting, imputation, classification, and anomaly detection tasks.

**Implementation Details** Table 1 provides an overview of the benchmarks. Further information about the datasets is available in Appendix G. Our DiffKANformer model is trained with the Adam optimizer, utilizing a learning rate of $10^{-4}$. The training procedure uses a batch size of 64, employs early stopping with a patience of 10, and continues for 100 epochs. Diffusion steps are implemented with a quadratic variance schedule, starting at $\beta_1 = 10^{-4}$ and escalating to $\beta_T = 10^{-1}$. All experiments run on a single Nvidia RTX A5000 GPU with 24GB. Figure 1 illustrates the architecture used in the DiffKANformer model. Additional information about the baselines and hyperparameters can be found in Appendix H and Appendix I.

Table 1: Summary of dataset benchmarks.

| Tasks | Benchmarks | Metrics | Series Length |
|---|---|---|---|
| Forecasting | Norpool, Casio, Traffic, Electricity, Weather, Exchange, ETTh1, ETTm1 | MSE, MAE | 96 to 720 |
| Imputation | ETT (4 subsets), Electricity, Weather | MSE, MAE | 96 |
| Classification | UEA (10 subsets) | Accuracy | 29 to 1751 |
| Anomaly Detection | SMD, MSL, SMAP, SWaT, PSM | Precision, Recall, F1-Score | 100 |

Main quantitative results are provided in the following. Appendix contains full quantitative results (Appendix M), qualitative results (Appendix N) and statistical significance analysis (Appendix L).

### 3.1 FORECASTING

**Setup** We performed experiments on eight real-world time series datasets for daily, weekly, and monthly forecasts Shen et al. (2024); Fan et al. (2022); Zhou et al. (2021); Wu et al. (2021). The length of the look-back window is selected from the set {96, 192, 336, 720, 1440}, determined by performance evaluations on the validation dataset averaged over ten runs with a fixed prediction length (Table 8). Here, the condition is the time series in the look-back window and the prediction is the future time series.

**Results** Table 2 presents the mean squared errors (MSEs) for multivariate forecasting. We see that our DiffKANformer has a rank of 1 in 4 out of the 8 datasets. Notable improvements are observed, particularly in complex datasets like Weather and ETTh1. In the other four datasets, DiffKANformer consistently holds Rank 2, with a very narrow margin from Rank 1 in each case. On average, the DiffKANformer model achieves a ranking of 1.5, demonstrating its state-of-the-art performance in comparison to other leading models in forecasting tasks. The results with mean absolute error (MAE) metrics are presented in Table 17 of the Appendix M.

### 3.2 IMPUTATION

**Setup** For imputation experiments, we use six datasets from the electricity and weather domains as benchmarks, including ETT (Zhou et al., 2021), Electricity [1], and Weather [2], all of which frequently encounter missing data issues. To evaluate model performance under varying levels of missing data, we randomly mask time points with ratios of 12.5%, 25%, 37.5%, 50%. In this context, masked time series is used as the condition for DiffKANformer and the processes of adding noise and denoising are restricted to the masked portions in the diffusion process.

**Results** Table 3 presents the results for the imputation task with the evaluation metric as the average MSE over various random masking ratios. Our model consistently delivers top-tier performance, recording the lowest error metrics across all datasets, with notably marked improvements on datasets such as ECL, ETTh1, ETTh2, ETTm1, and ETTm2. For example, in ETTm2, our method achieves an error rate of 0.016, significantly exceeding the nearest competitor. Overall, DiffKANformer secures an average rank of 1.5, clearly outperforming robust baselines like Nonstationary, TimesNet,

---

[1] https://archive.ics.uci.edu/ml/datasets/ElectricityLoadDiagrams20112014
[2] https://www.bgc-jena.mpg.de/wetter/

Table 2: Multivariate prediction of MSEs on eight real-world time series datasets (subscripts are the rank). CSDI and TiDE run out of memory on Traffic and Electricity. Results of all baselines are from (Shen et al., 2024). Baselines with * are run using TSLib (Wang et al., 2024b; Wu et al., 2022)

| Method | NorPool | Caiso | Traffic | ECL | Weather | Exchange | ETTh1 | ETTm1 | Rank |
|---|---|---|---|---|---|---|---|---|---|
| Ours | $\underline{0.544}_{(2)}$ | $\mathbf{0.089}_{(1)}$ | $\underline{0.374}_{(2)}$ | $\mathbf{0.144}_{(1)}$ | $\mathbf{0.293}_{(1)}$ | $\underline{0.016}_{(2)}$ | $\mathbf{0.401}_{(1)}$ | $\underline{0.338}_{(2)}$ | 1.5 |
| CnDiff* | $\mathbf{0.531}_{(1)}$ | $\underline{0.094}_{(2)}$ | $\underline{0.374}_{(2)}$ | $\underline{0.145}_{(2)}$ | $0.296_{(2)}$ | $0.016_{(2)}$ | $\underline{0.405}_{(2)}$ | $0.340_{(3)}$ | 2.0 |
| mr-Diff | $0.645_{(5)}$ | $0.127_{(5)}$ | $0.474_{(11)}$ | $0.155_{(7)}$ | $\underline{0.296}_{(2)}$ | $0.016_{(2)}$ | $0.411_{(5)}$ | $0.340_{(3)}$ | 5.0 |
| TimeDiff | $0.665_{(7)}$ | $0.136_{(11)}$ | $0.564_{(17)}$ | $0.193_{(16)}$ | $0.311_{(4)}$ | $0.018_{(15)}$ | $0.407_{(3)}$ | $\mathbf{0.336}_{(1)}$ | 9.2 |
| SSSD | $0.872_{(23)}$ | $0.195_{(25)}$ | $0.642_{(23)}$ | $0.255_{(27)}$ | $0.349_{(19)}$ | $0.061_{(30)}$ | $0.726_{(33)}$ | $0.464_{(26)}$ | 25.8 |
| CSDI | $1.010_{(34)}$ | $0.253_{(34)}$ | – | – | $0.356_{(22)}$ | $0.077_{(34)}$ | $0.497_{(19)}$ | $0.529_{(32)}$ | 29.2 |
| TimeGrad | $1.152_{(36)}$ | $0.258_{(35)}$ | $1.745_{(37)}$ | $0.736_{(37)}$ | $0.392_{(27)}$ | $0.079_{(35)}$ | $0.993_{(38)}$ | $0.874_{(36)}$ | 35.1 |
| D3VAE | $0.745_{(13)}$ | $0.241_{(32)}$ | $0.928_{(31)}$ | $0.286_{(30)}$ | $0.375_{(24)}$ | $0.200_{(37)}$ | $0.504_{(21)}$ | $0.362_{(13)}$ | 25.1 |
| CPF | $1.613_{(39)}$ | $0.383_{(37)}$ | $1.625_{(36)}$ | $0.793_{(38)}$ | $1.390_{(39)}$ | $\underline{0.016}_{(2)}$ | $0.730_{(34)}$ | $0.482_{(29)}$ | 31.8 |
| PSA-GAN | $1.501_{(38)}$ | $0.510_{(39)}$ | $1.614_{(35)}$ | $0.535_{(36)}$ | $1.220_{(37)}$ | $0.018_{(15)}$ | $0.623_{(32)}$ | $0.537_{(33)}$ | 33.1 |
| N-Hits | $0.716_{(11)}$ | $0.131_{(7)}$ | $0.386_{(4)}$ | $0.152_{(5)}$ | $0.323_{(10)}$ | $0.017_{(10)}$ | $0.498_{(20)}$ | $0.353_{(10)}$ | 9.6 |
| FiLM | $0.723_{(12)}$ | $0.179_{(21)}$ | $0.628_{(22)}$ | $0.210_{(20)}$ | $0.327_{(12)}$ | $\underline{0.016}_{(2)}$ | $0.426_{(10)}$ | $0.347_{(7)}$ | 13.2 |
| NBeats | $0.832_{(17)}$ | $0.141_{(12)}$ | $\mathbf{0.373}_{(1)}$ | $0.269_{(28)}$ | $1.344_{(38)}$ | $\underline{0.016}_{(2)}$ | $0.586_{(29)}$ | $0.391_{(21)}$ | 18.5 |
| Depts | $0.662_{(6)}$ | $0.106_{(4)}$ | $1.018_{(34)}$ | $0.319_{(32)}$ | $0.761_{(35)}$ | $0.020_{(18)}$ | $0.579_{(27)}$ | $0.380_{(19)}$ | 21.9 |
| TimeXer* | $0.715_{(10)}$ | $0.145_{(14)}$ | $0.435_{(9)}$ | $0.153_{(6)}$ | $0.315_{(7)}$ | $0.017_{(12)}$ | $0.424_{(9)}$ | $0.345_{(6)}$ | 9.1 |
| Crossformer* | $0.833_{(19)}$ | $0.127_{(6)}$ | $0.491_{(13)}$ | $0.149_{(4)}$ | $0.321_{(8)}$ | $0.069_{(32)}$ | $0.452_{(15)}$ | $0.348_{(8)}$ | 13.1 |
| PAttn* | $0.832_{(18)}$ | $0.162_{(18)}$ | $0.492_{(14)}$ | $0.183_{(15)}$ | $0.346_{(17)}$ | $\mathbf{0.015}_{(1)}$ | $0.427_{(11)}$ | $0.363_{(14)}$ | 13.5 |
| MultiPatchFormer* | $0.836_{(20)}$ | $0.155_{(17)}$ | $0.441_{(10)}$ | $0.167_{(10)}$ | $0.347_{(18)}$ | $0.017_{(14)}$ | $0.418_{(7)}$ | $0.369_{(16)}$ | 14.0 |
| iTransformer* | $0.864_{(22)}$ | $0.176_{(20)}$ | $0.429_{(6)}$ | $0.160_{(8)}$ | $0.354_{(21)}$ | $0.017_{(11)}$ | $0.440_{(14)}$ | $0.384_{(20)}$ | 15.2 |
| TimesNet* | $0.827_{(15)}$ | $0.146_{(15)}$ | $0.595_{(19)}$ | $0.179_{(14)}$ | $0.337_{(15)}$ | $0.021_{(20)}$ | $0.460_{(16)}$ | $0.376_{(18)}$ | 16.5 |
| Nonstationary* | $0.576_{(3)}$ | $0.134_{(9)}$ | $0.671_{(27)}$ | $0.179_{(13)}$ | $0.352_{(20)}$ | $0.023_{(22)}$ | $0.575_{(26)}$ | $0.463_{(25)}$ | 18.1 |
| TiDE* | $0.947_{(28)}$ | $0.236_{(31)}$ | – | $0.203_{(19)}$ | $0.459_{(29)}$ | $0.016_{(8)}$ | $0.435_{(12)}$ | $0.404_{(22)}$ | 24.8 |
| FedFormer | $0.873_{(24)}$ | $0.205_{(26)}$ | $0.591_{(18)}$ | $0.238_{(25)}$ | $0.342_{(16)}$ | $0.133_{(36)}$ | $0.541_{(24)}$ | $0.426_{(24)}$ | 24.1 |
| PatchTST | $0.851_{(21)}$ | $0.193_{(24)}$ | $0.831_{(30)}$ | $0.225_{(22)}$ | $0.782_{(36)}$ | $0.047_{(28)}$ | $0.526_{(23)}$ | $0.372_{(17)}$ | 25.1 |
| Scaleformer | $0.983_{(29)}$ | $0.207_{(28)}$ | $0.618_{(21)}$ | $0.195_{(17)}$ | $0.462_{(30)}$ | $0.036_{(25)}$ | $0.613_{(31)}$ | $0.481_{(28)}$ | 26.1 |
| Autoformer | $0.940_{(26)}$ | $0.226_{(30)}$ | $0.688_{(28)}$ | $0.201_{(18)}$ | $0.360_{(23)}$ | $0.056_{(29)}$ | $0.516_{(22)}$ | $0.565_{(34)}$ | 26.2 |
| ETSformer* | $1.011_{(35)}$ | $0.180_{(22)}$ | $0.955_{(32)}$ | $0.241_{(26)}$ | $0.473_{(31)}$ | $0.025_{(23)}$ | $0.599_{(30)}$ | $0.501_{(31)}$ | 28.8 |
| Pyraformer | $1.008_{(33)}$ | $0.273_{(36)}$ | $0.659_{(24)}$ | $0.273_{(29)}$ | $0.394_{(28)}$ | $0.032_{(24)}$ | $0.579_{(27)}$ | $0.493_{(30)}$ | 28.9 |
| Transformer | $1.004_{(32)}$ | $0.206_{(27)}$ | $0.671_{(26)}$ | $0.328_{(33)}$ | $0.388_{(26)}$ | $0.062_{(31)}$ | $0.759_{(35)}$ | $0.992_{(38)}$ | 31.0 |
| Informer | $0.985_{(30)}$ | $0.251_{(33)}$ | $0.664_{(25)}$ | $0.298_{(31)}$ | $0.385_{(25)}$ | $0.073_{(33)}$ | $0.775_{(36)}$ | $0.673_{(35)}$ | 31.0 |
| Reformer* | $0.907_{(25)}$ | $0.166_{(19)}$ | $0.713_{(29)}$ | $0.332_{(34)}$ | $0.682_{(34)}$ | $0.074_{(39)}$ | $0.953_{(37)}$ | $0.874_{(37)}$ | 31.8 |
| NLinear | $0.707_{(9)}$ | $0.135_{(10)}$ | $0.430_{(7)}$ | $0.147_{(3)}$ | $0.313_{(5)}$ | $0.019_{(17)}$ | $0.410_{(4)}$ | $0.349_{(9)}$ | 8.0 |
| SCINet | $0.613_{(4)}$ | $0.095_{(3)}$ | $0.434_{(8)}$ | $0.171_{(12)}$ | $0.329_{(13)}$ | $0.036_{(25)}$ | $0.465_{(17)}$ | $0.359_{(12)}$ | 11.8 |
| MICN* | $0.770_{(14)}$ | $0.133_{(8)}$ | $0.517_{(16)}$ | $0.168_{(11)}$ | $0.322_{(9)}$ | $0.017_{(13)}$ | $0.435_{(13)}$ | $0.357_{(11)}$ | 11.9 |
| TimeMixer* | $0.828_{(16)}$ | $0.183_{(23)}$ | $0.499_{(15)}$ | $0.164_{(9)}$ | $0.337_{(14)}$ | $0.016_{(9)}$ | $0.420_{(8)}$ | $0.366_{(15)}$ | 13.6 |
| DLinear | $0.670_{(8)}$ | $0.461_{(38)}$ | $0.389_{(5)}$ | $0.215_{(21)}$ | $0.488_{(32)}$ | $0.022_{(21)}$ | $0.415_{(6)}$ | $0.345_{(5)}$ | 17.0 |
| LightTS* | $0.943_{(27)}$ | $0.151_{(16)}$ | $0.601_{(20)}$ | $0.225_{(23)}$ | $0.324_{(11)}$ | $0.021_{(19)}$ | $0.465_{(18)}$ | $0.411_{(23)}$ | 19.6 |
| TSMixer* | $0.990_{(31)}$ | $0.144_{(13)}$ | $0.476_{(12)}$ | $0.225_{(24)}$ | $0.314_{(6)}$ | $0.037_{(27)}$ | $0.550_{(25)}$ | $0.478_{(27)}$ | 20.6 |
| LSTMa | $1.481_{(37)}$ | $0.217_{(29)}$ | $0.966_{(33)}$ | $0.414_{(35)}$ | $0.662_{(33)}$ | $0.403_{(38)}$ | $1.149_{(39)}$ | $1.030_{(39)}$ | 35.4 |

and Crossformer, which hold ranks of 2.2, 3.0, and 7.3, respectively. These findings emphasize our model's superior ability to manage missing data and accurately reconstruct masked time series portions with diverse temporal dynamics. The results with MSE for individual masking rations and MAE as the metric are provided in Tables 18 and 19 in the appendix M.

## 3.3 CLASSIFICATION

**Setup** For classification experiments, we perform sequence-level classification using DiffKAN-former to learn the representations that are fed into a classification head. The benchmark datasets are chosen to be 10 multivariate datasets from the UEA Time Series Classification Archive (Bagnall et al., 2018), which include tasks such as gesture, action, and audio recognition, as well as medical diagnosis through heartbeat monitoring, among other practical applications. Subsequently, we pre-process these datasets according to the guidelines provided in (Zerveas et al., 2021), noting that different subsets exhibit varying sequence lengths. The time series that must be classified is used as a condition for diffusion, and noise is added to this series, allowing the DiffKANformer to learn representations which are fed to the classification head. Training is performed end-to-end, including the classification head.

**Results** Table 4 presents accuracy as the metric for the classification task. Our DiffKANformer consistently attains the highest average accuracy of 0.738 along with the best average ranking of

Table 3: In the context of the Imputation task, we randomly took different proportions of time points specifically 12.5%, 25%, 37.5%, and 50% within 96-length time series and evaluated the performance using MSE. Average outcomes across four distinct masking ratios are presented, indicated by subscripts denoting ranks. For detailed outcomes, please refer to 18 and 19, which present the full results and the MAE metrics. Results of all baselines are run using TSLib (Wang et al., 2024b; Wu et al., 2022).

| Models | ECL | ETTh1 | ETTh2 | ETTm1 | ETTm2 | Weather | Rank |
|---|---|---|---|---|---|---|---|
| Ours | **0.075** $_{(1)}$ | **0.065** $_{(1)}$ | **0.037** $_{(1)}$ | 0.024 $_{(2)}$ | **0.016** $_{(1)}$ | 0.032 $_{(3)}$ | 1.5 |
| Nonstationary | 0.088 $_{(2)}$ | 0.075 $_{(2)}$ | 0.049 $_{(2)}$ | 0.026 $_{(3)}$ | 0.021 $_{(2)}$ | 0.030 $_{(2)}$ | 2.2 |
| TimesNet | 0.094 $_{(3)}$ | 0.089 $_{(4)}$ | 0.051 $_{(3)}$ | 0.027 $_{(4)}$ | 0.022 $_{(3)}$ | **0.029** $_{(1)}$ | 3.0 |
| Crossformer | 0.097 $_{(4)}$ | 0.148 $_{(12)}$ | 0.151 $_{(5)}$ | 0.060 $_{(9)}$ | 0.079 $_{(5)}$ | 0.043 $_{(9)}$ | 7.3 |
| LightTS | 0.108 $_{(6)}$ | 0.159 $_{(14)}$ | 0.152 $_{(6)}$ | 0.068 $_{(10)}$ | 0.076 $_{(4)}$ | 0.046 $_{(10)}$ | 8.3 |
| iTransformer | 0.099 $_{(5)}$ | 0.148 $_{(13)}$ | 0.139 $_{(4)}$ | 0.071 $_{(11)}$ | 0.082 $_{(6)}$ | 0.051 $_{(12)}$ | 8.5 |
| SSSD | 0.208 $_{(15)}$ | 0.078 $_{(3)}$ | 0.410 $_{(17)}$ | **0.023** $_{(1)}$ | 0.152 $_{(13)}$ | 0.032 $_{(4)}$ | 8.8 |
| Reformer | 0.162 $_{(11)}$ | 0.101 $_{(5)}$ | 0.226 $_{(12)}$ | 0.033 $_{(5)}$ | 0.165 $_{(14)}$ | 0.039 $_{(7)}$ | 9.0 |
| Pyraformer | 0.189 $_{(14)}$ | 0.116 $_{(8)}$ | 0.201 $_{(10)}$ | 0.037 $_{(7)}$ | 0.128 $_{(10)}$ | 0.034 $_{(5)}$ | 9.0 |
| Transformer | 0.165 $_{(12)}$ | 0.108 $_{(6)}$ | 0.256 $_{(14)}$ | 0.034 $_{(6)}$ | 0.203 $_{(15)}$ | 0.036 $_{(6)}$ | 9.8 |
| TiDE | 0.129 $_{(8)}$ | 0.169 $_{(15)}$ | 0.163 $_{(8)}$ | 0.090 $_{(14)}$ | 0.101 $_{(7)}$ | 0.052 $_{(13)}$ | 10.8 |
| DLinear | 0.128 $_{(7)}$ | 0.170 $_{(16)}$ | 0.162 $_{(7)}$ | 0.090 $_{(15)}$ | 0.101 $_{(8)}$ | 0.052 $_{(14)}$ | 11.2 |
| Informer | 0.171 $_{(13)}$ | 0.114 $_{(7)}$ | 0.366 $_{(16)}$ | 0.049 $_{(8)}$ | 0.245 $_{(16)}$ | 0.042 $_{(8)}$ | 11.3 |
| ETSformer | 0.136 $_{(10)}$ | 0.138 $_{(10)}$ | 0.234 $_{(13)}$ | 0.073 $_{(13)}$ | 0.142 $_{(12)}$ | 0.048 $_{(11)}$ | 11.5 |
| FiLM | 0.129 $_{(9)}$ | 0.174 $_{(17)}$ | 0.169 $_{(9)}$ | 0.090 $_{(16)}$ | 0.102 $_{(9)}$ | 0.057 $_{(15)}$ | 12.5 |
| FEDformer | 0.215 $_{(16)}$ | 0.128 $_{(9)}$ | 0.224 $_{(11)}$ | 0.072 $_{(12)}$ | 0.129 $_{(11)}$ | 0.073 $_{(16)}$ | 12.5 |
| Autoformer | 0.231 $_{(17)}$ | 0.147 $_{(11)}$ | 0.266 $_{(15)}$ | 0.740 $_{(17)}$ | 0.862 $_{(17)}$ | 0.210 $_{(17)}$ | 15.7 |

1.9, surpassing all other models. In most datasets, DiffKANformer achieves the highest accuracy (marked in bold) or ranks among the top performers. Notably, it achieves the best accuracy on datasets such as FaceDet (0.692), Handwriting (0.382), PEMS-SF (0.898), SpokenArab (0.994), and UWave (0.884). Although competing models such as TimesNet and Heartbeat deliver commendable results, particularly Heartbeat with a notable accuracy of 0.809 on the Heartbeat dataset, DiffKANformer stands out with its superior average ranking and overall accuracy. This indicates that the new method is more effective and robust for the time series classification task compared to existing state-of-the-art models.

Table 4: Classification task with accuracy as the metric. **to be read as former (e.g., In** is Informer). FiLM runs out of memory for the PEMS-SF dataset.

| Method | Ethanol | FaceDet | Handwriting | Heartbeat | Japanese | PEMS-SF | SCP1 | SCP2 | SpokenArab | UWave | Acc | Rank |
|---|---|---|---|---|---|---|---|---|---|---|---|---|
| Ours | 0.316 $_{(2)}$ | **0.692** $_{(1)}$ | **0.382** $_{(1)}$ | 0.750 $_{(8)}$ | **0.978** $_{(1)}$ | **0.898** $_{(1)}$ | 0.914 $_{(2)}$ | **0.578** $_{(1)}$ | **0.994** $_{(1)}$ | **0.884** $_{(1)}$ | 0.738 | 1.9 |
| TimesNet | 0.304 $_{(4)}$ | 0.674 $_{(5)}$ | 0.327 $_{(3)}$ | **0.809** $_{(1)}$ | **0.978** $_{(1)}$ | 0.884 $_{(2)}$ | 0.914 $_{(2)}$ | 0.555 $_{(3)}$ | 0.988 $_{(3)}$ | 0.881 $_{(2)}$ | 0.731 | 2.6 |
| Trans* | 0.277 $_{(10)}$ | 0.686 $_{(2)}$ | 0.377 $_{(2)}$ | 0.780 $_{(7)}$ | **0.978** $_{(1)}$ | 0.832 $_{(7)}$ | 0.907 $_{(4)}$ | 0.533 $_{(9)}$ | 0.989 $_{(2)}$ | 0.865 $_{(4)}$ | 0.722 | 4.7 |
| In* | 0.258 $_{(13)}$ | 0.669 $_{(7)}$ | 0.317 $_{(5)}$ | 0.780 $_{(2)}$ | 0.970 $_{(6)}$ | 0.826 $_{(8)}$ | 0.914 $_{(2)}$ | 0.561 $_{(2)}$ | 0.986 $_{(5)}$ | 0.862 $_{(5)}$ | 0.714 | 5.6 |
| Re* | 0.281 $_{(8)}$ | 0.681 $_{(3)}$ | 0.316 $_{(6)}$ | 0.775 $_{(4)}$ | 0.970 $_{(6)}$ | 0.815 $_{(9)}$ | 0.897 $_{(8)}$ | 0.533 $_{(9)}$ | 0.985 $_{(6)}$ | 0.868 $_{(3)}$ | 0.712 | 6.0 |
| iTrans* | 0.269 $_{(12)}$ | 0.669 $_{(7)}$ | 0.271 $_{(8)}$ | 0.756 $_{(6)}$ | 0.975 $_{(5)}$ | 0.872 $_{(3)}$ | **0.921** $_{(1)}$ | 0.544 $_{(5)}$ | 0.980 $_{(7)}$ | 0.859 $_{(6)}$ | 0.712 | 6.1 |
| FED* | 0.292 $_{(6)}$ | 0.670 $_{(6)}$ | 0.235 $_{(10)}$ | 0.765 $_{(5)}$ | **0.978** $_{(1)}$ | 0.843 $_{(5)}$ | 0.597 $_{(13)}$ | 0.527 $_{(11)}$ | 0.987 $_{(4)}$ | 0.587 $_{(13)}$ | 0.648 | 7.5 |
| LightTS | 0.304 $_{(4)}$ | 0.665 $_{(9)}$ | 0.205 $_{(12)}$ | 0.751 $_{(7)}$ | 0.962 $_{(12)}$ | 0.867 $_{(4)}$ | 0.911 $_{(6)}$ | 0.538 $_{(7)}$ | 0.977 $_{(8)}$ | 0.831 $_{(9)}$ | 0.701 | 7.8 |
| Pyra* | 0.311 $_{(3)}$ | 0.555 $_{(14)}$ | 0.325 $_{(4)}$ | 0.663 $_{(13)}$ | 0.970 $_{(6)}$ | 0.554 $_{(13)}$ | 0.802 $_{(11)}$ | 0.538 $_{(7)}$ | 0.968 $_{(11)}$ | 0.825 $_{(10)}$ | 0.651 | 9.2 |
| PatchTST | 0.273 $_{(11)}$ | 0.679 $_{(4)}$ | 0.265 $_{(9)}$ | 0.678 $_{(11)}$ | 0.959 $_{(13)}$ | 0.861 $_{(5)}$ | 0.849 $_{(10)}$ | 0.516 $_{(12)}$ | 0.966 $_{(12)}$ | 0.850 $_{(7)}$ | 0.690 | 9.6 |
| FiLM | 0.281 $_{(8)}$ | 0.645 $_{(10)}$ | 0.136 $_{(14)}$ | 0.731 $_{(9)}$ | 0.913 $_{(14)}$ | - | 0.890 $_{(9)}$ | 0.550 $_{(4)}$ | 0.977 $_{(8)}$ | 0.781 $_{(12)}$ | 0.656 | 9.8 |
| Cross* | **0.395** $_{(1)}$ | 0.625 $_{(12)}$ | 0.276 $_{(7)}$ | 0.546 $_{(14)}$ | 0.970 $_{(6)}$ | 0.809 $_{(10)}$ | 0.754 $_{(12)}$ | 0.461 $_{(14)}$ | 0.949 $_{(14)}$ | 0.834 $_{(8)}$ | 0.662 | 9.9 |
| DLinear | 0.254 $_{(14)}$ | 0.628 $_{(11)}$ | 0.221 $_{(11)}$ | 0.678 $_{(11)}$ | 0.967 $_{(10)}$ | 0.793 $_{(11)}$ | 0.914 $_{(2)}$ | 0.544 $_{(5)}$ | 0.964 $_{(13)}$ | 0.821 $_{(11)}$ | 0.678 | 10.1 |
| Auto* | 0.285 $_{(7)}$ | 0.586 $_{(13)}$ | 0.185 $_{(13)}$ | 0.721 $_{(10)}$ | 0.964 $_{(11)}$ | 0.786 $_{(12)}$ | 0.552 $_{(14)}$ | 0.511 $_{(13)}$ | 0.975 $_{(10)}$ | 0.518 $_{(14)}$ | 0.608 | 11.8 |

## 3.4 ANOMALY DETECTION

**Setup** Experiments are performed for unsupervised anomaly detection in time series, aiming to pinpoint abnormal time instances. Five popular anomaly detection benchmarks are used: SMD (Su et al., 2019), MSL (Hundman et al., 2018a), SMAP (Hundman et al., 2018b), SWaT (Mathur & Tippenhauer, 2016), and PSM (Abdulaal et al., 2021), which encompass applications in service monitoring, space and earth exploration, and water treatment. Using the pre-processing techniques of Anomaly Transformer (Xu et al., 2021), we segment the data set into successive non-overlapping

portions using a sliding window approach. In prior research, reconstruction has served as a standard method for unsupervised point-wise representation learning, where reconstruction error acts as a natural anomaly indicator. To ensure a fair evaluation, we modify only the base model used for reconstruction with our model, while maintaining the classical reconstruction error as the universal anomaly criterion across all experiments.

**Results** Table 5 summarizes the results of anomaly detection tasks using F1 score as evaluation metric. DiffKANformer excels as the leading method in most of the datasets. It achieves a notable margin over other leading models for MSL and SMAP datasets. For the other datasets, the F1 score are within top 4 ranks. Overall, DiffKANformer achieves the highest average F1 score of 90.26% with an impressive average rank of 2.4.

Table 5: Anomaly detection task. We calculate the F1-score (presented as a percentage) for each dataset. A higher F1-score signifies enhanced performance. For comprehensive results, refer to Table 20. All Baselines were executed with TSLib (Wang et al., 2024b; Wu et al., 2022).

| Model | MSL | PSM | SMAP | SMD | SWaT | Avg F1 | Rank |
|---|---|---|---|---|---|---|---|
| Ours | **90.38** $_{(1)}$ | 95.76 $_{(4)}$ | **91.84** $_{(1)}$ | 83.79 $_{(4)}$ | 90.16 $_{(2)}$ | 90.39 | 2.4 |
| TimesNet | 81.80 $_{(7)}$ | **97.40** $_{(1)}$ | 69.35 $_{(8)}$ | 84.59 $_{(3)}$ | **92.48** $_{(1)}$ | 85.12 | 4.0 |
| SSSD | 85.55 $_{(2)}$ | 92.97 $_{(9)}$ | 70.47 $_{(5)}$ | **88.88** $_{(1)}$ | 88.05 $_{(5)}$ | 85.18 | 4.4 |
| MICN | 79.41 $_{(14)}$ | 95.93 $_{(3)}$ | 70.52 $_{(4)}$ | 84.82 $_{(2)}$ | 89.85 $_{(3)}$ | 84.11 | 5.2 |
| DLinear | 81.87 $_{(6)}$ | 96.63 $_{(2)}$ | 68.97 $_{(15)}$ | 83.00 $_{(7)}$ | 85.73 $_{(7)}$ | 83.24 | 7.4 |
| iTransformer | 72.53 $_{(16)}$ | 94.32 $_{(5)}$ | 69.31 $_{(9)}$ | 81.41 $_{(9)}$ | 89.24 $_{(4)}$ | 81.36 | 8.6 |
| AnomalyTrans$^+$ | 83.59 $_{(3)}$ | 92.68 $_{(10)}$ | 70.15 $_{(6)}$ | 78.50 $_{(11)}$ | 79.24 $_{(17)}$ | 80.83 | 9.4 |
| FiLM | 63.88 $_{(18)}$ | 93.70 $_{(7)}$ | 69.01 $_{(14)}$ | 83.25 $_{(5)}$ | 85.72 $_{(8)}$ | 79.11 | 10.4 |
| LightTS | 80.99 $_{(11)}$ | 93.41 $_{(8)}$ | 68.92 $_{(17)}$ | 82.70 $_{(8)}$ | 81.74 $_{(9)}$ | 81.55 | 10.6 |
| Transformer | 80.90 $_{(12)}$ | 90.64 $_{(16)}$ | 73.26 $_{(3)}$ | 71.19 $_{(14)}$ | 79.53 $_{(10)}$ | 79.10 | 11.0 |
| KANAD | 81.07 $_{(10)}$ | 90.53 $_{(17)}$ | 69.10 $_{(12)}$ | 83.09 $_{(6)}$ | 79.30 $_{(11)}$ | 80.62 | 11.2 |
| Reformer | 81.10 $_{(8)}$ | 92.33 $_{(11)}$ | 69.26 $_{(11)}$ | 71.04 $_{(16)}$ | 79.30 $_{(11)}$ | 78.61 | 11.4 |
| Autoformer | 82.59 $_{(4)}$ | 88.23 $_{(18)}$ | 74.01 $_{(2)}$ | 71.16 $_{(15)}$ | 79.19 $_{(18)}$ | 79.04 | 11.4 |
| Pyraformer | 77.19 $_{(15)}$ | 92.24 $_{(13)}$ | 69.41 $_{(7)}$ | 78.97 $_{(10)}$ | 79.29 $_{(14)}$ | 79.42 | 11.8 |
| Informer | 81.08 $_{(9)}$ | 92.32 $_{(12)}$ | 69.31 $_{(9)}$ | 71.04 $_{(16)}$ | 79.29 $_{(14)}$ | 78.61 | 12.0 |
| Crossformer | 80.61 $_{(13)}$ | 94.30 $_{(6)}$ | 69.06 $_{(13)}$ | 74.50 $_{(13)}$ | 79.28 $_{(16)}$ | 79.55 | 12.2 |
| FEDformer | 82.22 $_{(5)}$ | 91.40 $_{(14)}$ | 68.96 $_{(16)}$ | 69.93 $_{(18)}$ | 79.30 $_{(11)}$ | 78.36 | 12.8 |
| ETSformer | 68.34 $_{(17)}$ | 91.09 $_{(15)}$ | 68.57 $_{(18)}$ | 76.15 $_{(12)}$ | 86.76 $_{(6)}$ | 78.18 | 13.6 |

## 4 MODEL ANALYSIS

### 4.1 KAN PROJECTION

In Table 6, we have shown empirical that DiffKANformer performs better than the ablation run without KAN Projection. To understand what is learned by KAN Projection, we take a trained model and explore the correlation between the features in the learned latent representation space of KAN Projection at different diffusion model timesteps. We observed that there are increasing correlations between the features in learned latent space at different timesteps as shown in Figure 2. Thus, we hypothesize that this increased correlation and time-dependent adaptability perhaps facilitate a more effective diffusion process for time series analysis. Similar observations of correlation in latent space that help diffusion models have been made in image and video diffusion works (Ge et al., 2023). In the space of diffusion models for time series, Tashiro et al. (2021) has noted that learning the correlation between feature and temporal space is necessary for time series imputation tasks.

### 4.2 KAN VS MLP FOR TIME SERIES ANALYSIS

Table 7 demonstrate that KANs consistently outperform traditional MLPs across multiple tasks and evaluation metrics. Most notably, KANs show substantial improvements in anomaly detection capabilities, with the DiTKAN model achieving an F1 score of 90.16 compared to 80.80 for the MLP projected DiT model representing a significant 11.6% improvement in detection performance. Sim-

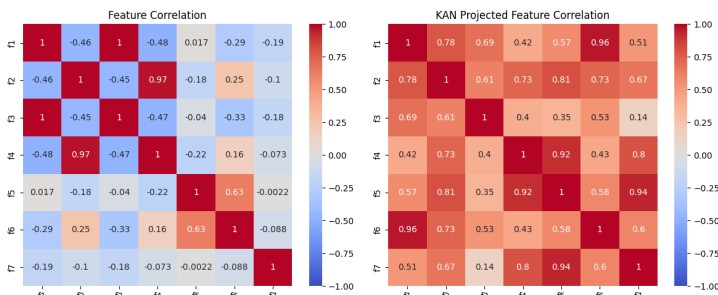

Figure 2: Correlation Matrix of Features for Actual Data and KAN Projection Data in ETTm1 for the Forecasting Task.

Table 6: Comparison of with and without kan projection across a variety of tasks. Various evaluation metrics are utilized, such as MSE for Imputation(25 % mask) and Forecasting, Accuracy for Anomaly Detection and classification.

| Task | Dataset | w KAN Projection | w/o KAN Projection |
|---|---|---|---|
| Forecasting | ETTh1 | **0.4015** | 0.9495 |
| | Weather | **0.2930** | 0.6960 |
| Imputation | ETTh1 | **0.0520** | 0.0649 |
| | ETTm1 | **0.0210** | 0.0454 |
| | ETTm2 | **0.0140** | 0.0309 |
| Classification | Handwriting | **0.3822** | 0.2115 |
| | PEMS | **0.8984** | 0.8359 |
| | SCP2 | **0.5781** | 0.4922 |
| Anomaly Detection | MSL | **0.9791** | 0.9765 |
| | PSM | **0.9783** | 0.9725 |
| | SMAP | **0.9793** | 0.9797 |
| | SMD | **0.9876** | 0.9863 |
| | SwAT | **0.9783** | 0.9703 |

Table 7: Ablation study comparing MLP and KAN architectures across different tasks. Results show forecasting performance (MSE), anomaly detection capability (F1 score), and classification accuracy for DiffKANformer with different neural network components

| | Model Architecture Combinations | | | |
|---|---|---|---|---|
| Task Dataset, Metric | DiT + MLP | DiT + KAN | DiTKAN + MLP | DiTKAN + KAN |
| Forecasting Weather, MSE | 0.325 | 0.320 | 0.308 | **0.293** |
| Imputation ETTm2, MSE | 0.035 | 0.028 | 0.034 | **0.013** |
| Anomaly Detection SwAT, F1 | 80.80 | 88.33 | 87.35 | **90.16** |
| Classification PEMS-SF, Acc. | 0.828 | 0.838 | 0.844 | **0.898** |

ilarly, for classification tasks, KANs deliver superior accuracy with DiTKAN and KAN projection reaching 89.8% accuracy versus 82.8% for the DiT model. While the improvements in forecasting tasks are more modest, KANs still maintain competitive or slightly better performance in MSE metrics for both weather forecasting and ETTm1 imputation tasks. These results suggest that KANs' learnable activation functions and enhanced representational capacity provide meaningful advantages over traditional MLPs for time series analysis.

In addition to these ablation studies, we also evaluated the efficiency and performance of the model through computational time analysis (Appendix K) and statistical significance analysis (Appendix L). The sensitivity analysis for different diffusion hyperparameters is in Appendix J.1, and for various look-back windows is in Appendix J.2. Results show competitive performance and robustness.

## 5 CONCLUSION

This paper introduces DiffKANformer, a diffusion-based framework that advances time series analysis through two key innovations: KAN-based projection in the forward diffusion process and the Diffusion KAN Transformer architecture. DiffKANformer achieves state-of-the-art performance in forecasting (8 datasets), imputation (6 datasets), classification (10 datasets), and anomaly detection (5 datasets), representing the first diffusion model to excel across all major time series analysis tasks. The integration of Kolmogorov-Arnold Networks addresses fundamental limitations of traditional MLPs in capturing complex temporal patterns, while our learnable KAN projection enables more effective modeling of complex correlation between features. These architectural advances, validated through extensive ablation studies, establish a new paradigm for time series modeling that bridges the gap between generative modeling and time series analysis.

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

## A   LLM USAGE

LLM based grammar check tool has been used to polish language after it was written by the authors.

## B   RELATED WORK

Classical time series methods include ARIMA (Shumway et al., 2017), Holt-Winter (Hyndman & Athanasopoulos, 2018), and Prophet (Sean & Taylor, 2018). Real-world time series often exhibit a complexity that exceeds these predefined frameworks, which restricts the effectiveness of these conventional approaches in practical applications. Thus several deep learning methods have been proposed for time series modeling.

Basis expansion has been utilized by several models for time-series analysis. FiLM (Zhou et al., 2022a) employed Fourier analysis and low-rank matrix approximation to reduce noise, along with Legendre polynomial projection to maintain historical data representations. NBeats (Oreshkin et al., 2019) is an interpretable architecture that combines polynomial trend modeling with Fourier techniques for detecting seasonality. Depts (Fan et al., 2022) builds on NBeats by adding a periodicity module for periodic series, while N-Hits (Challu et al., 2023) enhances the approach using multi-scale hierarchical interpolation.

Models such as SCINet (Liu et al., 2022a) utilize a recursive strategy involving downsampling, convolution, and interaction, to harness temporal dependencies in downsampled subsequences. NLinear (Zeng et al., 2023) implements a basic approach by normalizing the time series before using a linear layer for prediction. DLinear (Zeng et al., 2023) employs a seasonal-trend decomposition akin to Autoformer (Wu et al., 2021).

**Transformer Based Models:** Several models utilized the transformer (Vaswani et al., 2017) and its variations for time series analysis. Informer (Zhou et al., 2021) used sparse attention to minimize computational load and employed a generative-style decoder for rapid long-sequence in one forward pass. Autoformer (Wu et al., 2021) substituted traditional self-attention with an autocorrelation layer, while Fedformer (Zhou et al., 2022b) incorporated frequency domain mapping through a frequency-enhanced module. Pyraformer (Liu et al., 2022b) introduced pyramidal attention for multi-resolution representation, and Scaleformer (Shabani et al., 2023) adopted a shared-weight multilevel forecasting approach, from broad to fine scales. Inspired by vision transformers (Dosovitskiy et al., 2020b), PatchTST (Nie et al., 2023) partitions time series data into subseries patches and uses self-supervised pre-training to extract local semantic features.

**Diffusion Based Models:** TimeGrad (Rasul et al., 2021) pioneered the use of conditional diffusion models, leveraging autoregressive prediction informed by RNN hidden states. CSDI (Tashiro et al., 2021), applies non-autoregressive generation through self-supervised masking but relies on dual transformers, facing boundary inconsistencies and computational challenges with large datasets. SSSD (Lopez Alcaraz & Strodthoff, 2023) aims to reduce CSDI's computational load using structured state-space models, but it continues with masking-based conditioning, preserving boundary issues. TimeDiff (Shen & Kwok, 2023) introduced future mixup and autoregressive initialization within an encoder-decoder scheme for denoising. TMDM (Li et al., 2024b) integrates transformers with a diffusion process for probabilistic multivariate time series forecasting. TimeDiT (Cao et al., 2024) is a foundational model for time series, which employed a transformer type architecture to capture temporal dependencies and employs diffusion processes to generate samples. mr-Diff(Shen et al., 2024) is a recent study utilizing a multi-resolution strategy, incorporating fine-to-coarse patterns as latent variables to aid in denoising. CN-Diff (Rishi et al., 2025) introduced a generative framework that employs novel non-linear transformations and learnable conditions in the forward process for time series forecasting.

## C   PRELIMINARIES

### C.1   DIFFUSION MODEL FORMULATION

Diffusion models are generative frameworks that progressively corrupt data through a forward noising process and then learn to reverse this process to generate realistic samples. For a data point

$x \sim q(x)$, the forward process produces increasingly noisy latent variables $x^0, x^1, \ldots, x^T$, until $x^T$ resembles pure Gaussian noise. The generative model is trained to reverse this chain, starting from noise and reconstructing $x^0 \approx x$.

In the standard diffusion model, the forward process follows a linear Gaussian Markov chain (Sohl-Dickstein et al., 2015; Ho et al., 2020). In denoising diffusion probabilistic models (DDPM), at step $t$, $x^t$ is generated by modifying the previous state $x^{t-1}$ (multiplied by $\sqrt{\alpha_t}$) with zero-mean Gaussian noise and variance $(1-\alpha_t)$, that is, $q(x^t|x^{t-1}) = \mathcal{N}\left(x^t; \sqrt{\alpha_t}x^{t-1}, (1 - \alpha_t)I\right)$. We derive the marginal distribution as $q(x^t|x) = \mathcal{N}(x^t; \sqrt{\bar{\alpha}_t}x, (1 - \bar{\alpha}_t)I)$, where $\bar{\alpha}_t$ is defined as $\prod_{s=1}^t \alpha_s$. Hence, $x^t = \sqrt{\bar{\alpha}_t}x + \sqrt{1 - \bar{\alpha}_t}\epsilon$, with $\epsilon \sim \mathcal{N}(0, I)$. Leveraging these marginal distributions, the joint distribution for the latent variables $x^0, x^1, x^2, \ldots, x^T$ is

$$q(x^{0:T}|x) = \prod_{t=1}^T q(x^t \mid x^{t-1}).$$

The forward process is typically fixed, lacking trainable parameters, and it is designed so that $q(x^0|x) \approx \delta(x^0 - x)$ and $q(x^T|x) \approx \mathcal{N}(x^T; 0, I)$. If accessing $q(x^{t-1}|x^t)$ were feasible, we could sample from $x^T \sim \mathcal{N}(x^T; 0, I)$ and reverse it to yield $x^0 \sim q(x^0) \approx q(x)$. The distribution $q(x^{t-1}|x^t)$ depends implicitly on $q(x)$, forming a complex relationship. Hence, we approximate the reverse process through a Markov chain adopting the form

$$p_\theta(x^{0:T}) = p(x^T) \prod_{t=1}^T p_\theta(x^{t-1}|x^t),$$

with $p(x^T) = \mathcal{N}(x^T; 0, I)$. The integration of the forward process $q$ with the reverse process $p_\theta$ is akin to a (hierarchical) variational autoencoder (Kingma, 2013a; Rezende et al., 2014). During training, the standard variational bound of the negative log-likelihood is minimized. In the case of DDPM (Ho et al., 2020), the loss to be minimized is

$$\mathcal{L} = \mathbb{E}_q\left[\underbrace{D_{\mathrm{KL}}\left(q(x^T|x) \,\|\, p(x^T)\right)}_{\mathcal{L}_{\mathrm{prior}}} - \underbrace{\log p_\theta(x|x^0)}_{\mathcal{L}_{\mathrm{rec}}} + \sum_{t=1}^T \underbrace{D_{\mathrm{KL}}\left(q(x^{t-1}|x^t, x) \,\|\, p_\theta(x^{t-1}|x^t)\right)}_{\mathcal{L}_{\mathrm{diff}}}\right].$$

Given the fixed nature of process $q$ and distribution $p_\theta(x^T) = p(x^T)$, the prior term $\mathcal{L}_{prior}$ can be disregarded as it does not depend on parameters $\theta$. Since $\log p_\theta(x|x^0)$ is often modeled by a Gaussian distribution with low variance, the reconstruction term $\mathcal{L}_{rec}$ also remains unaffected by $\theta$. Consequently, the diffusion term $\mathcal{L}_{diff}$ is the only part that the model parameters $\theta$ influence. $\mathcal{L}_{diff}$ is the sum of Kullback–Leibler (KL) divergences between the posterior distribution in the forward process $q(x^{t-1}|x^t, x)$ and the assumed normal distribution $p_\theta(x^{t-1}|x^t) = \mathcal{N}(x^{t-1}; \mu_\theta(x^t, t), \Sigma_\theta(x^t, t))$ in the reverse process. Here, the variance $\Sigma_\theta(x^t, t)$ is set to $\sigma_t^2 I$, while the mean $\mu_\theta(x^t, t)$ is learned by a neural network with parameters $\theta$. This process is typically viewed as a noise estimation or data prediction problem (Benny & Wolf, 2022). To estimate noise, the network $\epsilon_\theta$ forecasts the noise in the diffused input $x^t$ and then computes $\mu_\theta(x^t, t)$ using the formula $\left(\frac{1}{\sqrt{\alpha_t}}x^t - \frac{1-\alpha_t}{\sqrt{1-\bar{\alpha}_t}\sqrt{\alpha_t}}\epsilon_\theta(x^t, t)\right)$. The parameter $\theta$ is learned by minimizing the loss function $\mathcal{L}_\epsilon = \mathbb{E}_{x,\epsilon,t}\left[\|\epsilon - \epsilon_\theta(x^t, t)\|^2\right]$. Alternatively, in the data prediction approach, a denoising network $x_\theta$ is employed to derive an estimate $x_\theta(x^t, t)$ of the clean data $x^0$ from $x^t$, and then set to

$$\mu_\theta(x^t, t) = \frac{\sqrt{\alpha_t}(1 - \bar{\alpha}_{t-1})}{1 - \bar{\alpha}_t}x^t + \frac{\sqrt{\bar{\alpha}_{t-1}}\beta_t}{1 - \bar{\alpha}_t}x_\theta(x^t, t).$$

Here, the parameter $\theta$ is learned by minimizing the loss..

$$\mathcal{L}_x = \mathbb{E}_{x,\epsilon,t}\left\|x - x_\theta(x^t, t)\right\|^2.$$

For time series diffusion models, forecasting $x_\theta$ has been found to be more effective than predicting $\epsilon_\theta$, as shown in (Feng et al., 2024; Shen & Kwok, 2023).

## C.2 Conditional Diffusion models for time series analysis

In time series analysis, the prediction target $x_{\text{output}}$ depends on the specific task at hand. For forecasting, the goal is to predict future values $x_{1:H}$ given past observations $x_{-L+1:0} \in \mathbb{R}^{d \times L}$, where $L$ is the lookback window, $H$ is the forecast horizon, and $d$ is the number of variables. For imputation, the aim is to reconstruct missing regions within the input sequence. For classification, the model learns a latent representation of the input time series that is subsequently mapped to class labels through a classification head. For anomaly detection, the objective is to identify irregular or abnormal patterns within the sequence.

When employing conditional diffusion models for these tasks, the generative distribution is expressed as $p_\theta(x_{\text{output}}^{0:T}|c) = p_\theta(x_{\text{output}}^T) \prod_{t=1}^{T} p_\theta(x_{\text{output}}^{t-1}|x_{\text{output}}^t, c)$, where $x_{\text{output}}^T \sim \mathcal{N}(0, I)$ and $c$ denotes the condition, which varies according to the task (e.g., past observations for forecasting, masked inputs for imputation, the full sequence for classification and anomaly detection) (Rasul et al., 2021; Shen & Kwok, 2023; Shen et al., 2024; Li et al., 2024b).

At each step $t$, the denoising transition is modeled as $p_\theta(x_{\text{output}}^{t-1}|x_{\text{output}}^t, c) = \mathcal{N}\left(x_{\text{output}}^{t-1}; \mu_\theta(x_{\text{output}}^t, t|c), \sigma_t^2 I\right)$. During inference, generation begins with $\hat{x}_{\text{output}}^T \sim \mathcal{N}(0, I)$. By iteratively applying the denoising step until $t = T$, we obtain the final prediction $\hat{x}_{\text{output}}$, which corresponds to future values in forecasting, completed sequences in imputation, class-discriminative features in classification, or anomaly-indicative representations in detection tasks.

## C.3 Kolmogrov-Arnold Networks

The Kolmogorov-Arnold representation theorem Hecht-Nielsen (1987) states that any multivariate continuous function $f$, defined on a bounded domain, can be expressed as a finite composition of continuous univariate functions and addition. Specifically, for a smooth function $f : [0,1]^n \to \mathbb{R}$, it can be represented as:

$$f(x_1, \ldots, x_n) = \sum_{q=1}^{2n+1} \Phi_q \left( \sum_{p=1}^{n} \Phi_{q,p}(x_p) \right).$$

Here, each function $\Phi_{q,p} : [0,1] \to \mathbb{R}$ and $\Phi_q : \mathbb{R} \to \mathbb{R}$ are continuous. This means that the $(2d + 1)(d + 1)$ univariate functions $\Phi_q$ and $\phi_{q,p}$ are enough for an exact representation of a $d$-variate function. Unlike traditional multi-layer perceptrons (MLPs), which employ fixed activation functions at each node, (Liu et al., 2024c) define a generalized Kolmogorov-Arnold layer which replaces s each weight parameter with a univariate function. The resulting functional form for deeper KAN can be expressed as : $\text{KAN}(x) = (\Phi_{L-1} \circ \Phi_{L-2} \circ \cdots \circ \Phi_0)(x)$ where $\Phi_l, l \in [0, 1, \cdots, L-1]$ is a KAN layer, and the output dimension of each KAN layer can be expressed as: $[n_0, n_1, \cdots, n_{L-1}]$. Therefore, the transform process of $j$-th feature in $l$-th layer can be formed as: $x_{l,j} = \sum_{i=1}^{n_{l-1}} \Phi_{l-1,j,i}(x_{l-1,i})$, $j = 1, \cdots, n_l$. where $\Phi$ consists of two parts: the spline function and the residual activation function with learnable parameters $w_a, w_b$: $\Phi(x) = w_a \text{SiLU}(x) + w_b \text{Spline}(x)$. where $\text{Spline}(\cdot)$ is a linear combination of B-spline functions $\text{Spline}(x) = \sum_i c_i B_i(x)$.

# D Formal Derivations

## D.1 Forward Posterior

Given:

$$q_\phi(\mathbf{x}^t|\mathbf{x}, c) = \mathcal{N}(\mathbf{x}^t; \sqrt{\bar{\alpha}_t} KAN_\phi(\mathbf{x}, t) + (1 - \sqrt{\bar{\alpha}_t})c, (1 - \bar{\alpha}_t)I) \tag{3}$$

From (3), We can write,

$$
\begin{aligned}
\mathbf{x}^t &= \sqrt{\bar{\alpha}_t} KAN_\phi(\mathbf{x}, t) + (1 - \sqrt{\bar{\alpha}_t})c + \sqrt{1 - \bar{\alpha}_t}\epsilon \\
&= \sqrt{\bar{\alpha}_t} KAN_\phi(\mathbf{x}, t) + (1 - \sqrt{\bar{\alpha}_t})c + \sqrt{1 - \alpha_t\bar{\alpha}_{t-1}}\epsilon \\
&= \sqrt{\bar{\alpha}_t} KAN_\phi(\mathbf{x}, t) + (1 - \sqrt{\bar{\alpha}_t})c + \sqrt{1 - \alpha_t}\epsilon + \sqrt{\alpha_t - \alpha_t\bar{\alpha}_{t-1}}\epsilon \\
&= \sqrt{\bar{\alpha}_t} KAN_\phi(\mathbf{x}, t) + (1 - \sqrt{\bar{\alpha}_t})c + \sqrt{1 - \alpha_t}\epsilon + \sqrt{\alpha_t}\sqrt{1 - \bar{\alpha}_{t-1}}\epsilon \\
&= \sqrt{\bar{\alpha}_t} KAN_\phi(\mathbf{x}, t) + (1 - \sqrt{\bar{\alpha}_t})c + \sqrt{1 - \alpha_t}\epsilon + \sqrt{\alpha_t}\left(\mathbf{x}^{t-1} - \sqrt{\bar{\alpha}_{t-1}} KAN_\phi(\mathbf{x}, t-1) - (1 - \sqrt{\bar{\alpha}_{t-1}})c\right) \\
&= \sqrt{\alpha_t}\mathbf{x}^{t-1} + \sqrt{\bar{\alpha}_t}\left(KAN_\phi(\mathbf{x}, t) - KAN_\phi(\mathbf{x}, t-1)\right) + (1 - \sqrt{\bar{\alpha}_t})c + \sqrt{1 - \alpha_t}\epsilon
\end{aligned}
$$

(4)

By introducing nonlinear, time-dependent projection of data, along with condition, results in a forward process that is non-Markovian as:

$$
q_\phi(\mathbf{x}^t|\mathbf{x}^{t-1}, \mathbf{x}, c) = \mathcal{N}(\mathbf{x}^t; \sqrt{\alpha_t}\mathbf{x}^{t-1} + \sqrt{\bar{\alpha}_t}\left(KAN_\phi(\mathbf{x}, t) - KAN_\phi(\mathbf{x}, t-1)\right) + (1 - \sqrt{\bar{\alpha}_t})c, (1 - \alpha_t)I)
$$

(5)

From (5 and 3), Posterior distribution can be derived as:

$$
\begin{aligned}
q_\phi\left(\mathbf{x}^{t-1}|\mathbf{x}^t, \mathbf{x}, c\right) &\propto \frac{q_\phi(\mathbf{x}^{t-1}, \mathbf{x}^t, \mathbf{x}, c)}{q_\phi(\mathbf{x}^t, \mathbf{x}, c)} \\
&\propto \frac{q_\phi(\mathbf{x}^t|\mathbf{x}^{t-1}, \mathbf{x}, c)}{q_\phi(\mathbf{x}^t, \mathbf{x}, c)} q_\phi(\mathbf{x}^{t-1}, \mathbf{x}, c) \\
&\propto q_\phi(\mathbf{x}^t|\mathbf{x}^{t-1}, \mathbf{x}, c) q_\phi(\mathbf{x}^{t-1}|\mathbf{x}, c)
\end{aligned}
$$

(6)

$$
\propto \exp\left(-\frac{1}{2}\left(\frac{\left(\mathbf{x}^t - \sqrt{\alpha_t}\mathbf{x}^{t-1} - \sqrt{\bar{\alpha}_t}(KAN_\phi(\mathbf{x}, t) - KAN_\phi(\mathbf{x}, t-1)) - (1 - \sqrt{\alpha_t})c\right)^2}{1 - \alpha_t}\right.\right.
$$
$$
\left.\left. +\frac{\left(\mathbf{x}^{t-1} - \sqrt{\bar{\alpha}_{t-1}} KAN_\phi(\mathbf{x}, t-1) - (1 - \sqrt{\bar{\alpha}_{t-1}})c\right)^2}{1 - \bar{\alpha}_{t-1}}\right)\right)
$$

$$
\propto \exp\left(-\frac{1}{2}\left((\frac{\alpha_t}{1 - \alpha_t} + \frac{1}{1 - \bar{\alpha}_{t-1}})(\mathbf{x}^{t-1})^2\right.\right.
$$
$$
-\frac{\left(2\sqrt{\alpha_t}\mathbf{x}^{t-1}(\mathbf{x}^t - \sqrt{\bar{\alpha}_t}(KAN_\phi(\mathbf{x}, t) - KAN_\phi(\mathbf{x}, t-1)) - (1 - \sqrt{\alpha_t})c\right)}{1 - \alpha_t}
$$
$$
\left.\left. -\frac{2\mathbf{x}^{t-1}}{1 - \bar{\alpha}_{t-1}}\left(\sqrt{\bar{\alpha}_{t-1}} KAN_\phi(\mathbf{x}, t-1) + (1 - \sqrt{\bar{\alpha}_{t-1}})c\right)\right)\right)
$$

(7)

By reformulating the above propotional form, akin to normal distribution, we can see that,

$$
q_\phi(\mathbf{x}^{t-1}|\mathbf{x}^t, \mathbf{x}, c) = \mathcal{N}(\mathbf{x}^{t-1}; \mu_{t-1|t}, \sigma_{t-1|t}^2 I)
$$

(8)

Where,

$$
\mu_{t-1|t} = \left(\frac{\sqrt{\alpha_t}\mathbf{x}^t - \sqrt{\bar{\alpha}_t}\sqrt{\alpha_t}(KAN_\phi(\mathbf{x}, t) - KAN_\phi(\mathbf{x}, t-1)) - \sqrt{\alpha_t}(1 - \sqrt{\alpha_t})c}{1 - \alpha_t}\right. +
$$
$$
\left. \frac{\sqrt{\bar{\alpha}_{t-1}} KAN_\phi(\mathbf{x}, t-1) + (1 - \sqrt{\bar{\alpha}_{t-1}})c}{1 - \bar{\alpha}_{t-1}}\right)\sigma_{t-1|t}^2
$$

(9)

$$
\sigma_{t-1|t}^2 = \frac{(1 - \alpha_t)(1 - \bar{\alpha}_{t-1})}{1 - \bar{\alpha}_t}
$$

(10)

Mean($\mu_{t-1|t}$) can be simplified as,

$$
\begin{aligned}
\mu_{t-1|t} &= \Bigg( \frac{\sqrt{\alpha_t}\mathbf{x}^t - \sqrt{\bar{\alpha}}_t\sqrt{\alpha_t}(KAN_\phi(\mathbf{x},t) - KAN_\phi(\mathbf{x},t-1)) - \sqrt{\alpha_t}(1 - \sqrt{\alpha_t})c}{(1-\alpha_t)} \frac{(1-\alpha_t)(1-\bar{\alpha}_{t-1})}{1-\bar{\alpha}_t} \\
&\qquad + \frac{\sqrt{\bar{\alpha}}_{t-1}KAN_\phi(\mathbf{x},t-1) + (1-\sqrt{\bar{\alpha}}_{t-1})c}{(1-\bar{\alpha}_{t-1})} \frac{(1-\alpha_t)(1-\bar{\alpha}_{t-1})}{1-\bar{\alpha}_t} \Bigg) \\
&= \Big( \sqrt{\alpha_t}\mathbf{x}^t - \sqrt{\bar{\alpha}}_t\sqrt{\alpha_t}(KAN_\phi(\mathbf{x},t) - KAN_\phi(\mathbf{x},t-1)) - \sqrt{\alpha_t}(1 - \sqrt{\alpha_t})c \Big) \frac{(1-\bar{\alpha}_{t-1})}{1-\bar{\alpha}_t} \\
&\qquad + \Big( \sqrt{\bar{\alpha}}_{t-1}KAN_\phi(\mathbf{x},t-1) + (1-\sqrt{\bar{\alpha}}_{t-1})c \Big) \frac{(1-\alpha_t)}{1-\bar{\alpha}_t}
\end{aligned}
\tag{11}
$$

$$
\mu_{t-1|t} = \zeta_1\mathbf{x}^t + \zeta_2 c - KAN_\phi(\mathbf{x},t)\zeta_1\sqrt{\bar{\alpha}}_t + KAN_\phi(\mathbf{x},t-1)(\zeta_1\sqrt{\bar{\alpha}}_t + \zeta_0)
$$

Where,

$$
\zeta_0 = \frac{1-\alpha_t}{1-\bar{\alpha}_t}\sqrt{\bar{\alpha}}_{t-1}; \qquad \zeta_1 = \frac{(1-\bar{\alpha}_{t-1})}{1-\bar{\alpha}_t}\sqrt{\alpha_t}; \qquad \zeta_2 = \left( 1 + \frac{(\sqrt{\bar{\alpha}}_t - 1)(\sqrt{\alpha_t} + \sqrt{\bar{\alpha}}_{t-1})}{1-\bar{\alpha}_t} \right);
$$

## D.2 Loss Formulation

Although our intention was not to explicitly make the forward process non-Markovian, our formulation has resulted in a non-Markovian forward process Eq.5. So, we can make use of joint distribution from DDIM (Song et al., 2020) as:

$$
q(x^{0:T}|x) = q(x^T|x) \prod_{t=1}^T q(x^{t-1}|x^t, x),
$$

Next, we define Non-Markovian trainable generative process $p_\theta(x_{0:T})$ as:

$$
p_\theta(x^{0:T}) = p(x^T) \prod_{t=1}^T p_\theta(x^{t-1}|x^t)p_\theta(x^0|x^t),
\tag{12}
$$

$$
\text{where } p(x^T) = \mathcal{N}(x^T; 0, I).
\tag{13}
$$

This formulation is similar to Equation (55) in DDIM (Song et al., 2020). However, the objective in the referenced work is to minimize the number of sampling time steps; therefore, certain time steps are employed for image generation, and others are included in the variational objective, ultimately demonstrating that the loss formulation aligns with DDPM.

Our definition of trainable reverse generative process (13) is a result of introducing a KAN projection in the marginal distribution, which makes it challenging to learn from the reverse Markovian process. However, even without non-linearity, the resulting loss will be similar to that of DDPM,

Now, by incorporating this into our variational loss term, the variational loss is given as:

$$
\mathbb{E}_{q_\phi}\Bigg[ \underbrace{D_{\mathrm{KL}}\left( q_\phi\left(\mathbf{x}^T|\mathbf{x},c\right) \| p\left(\mathbf{x}^T|c\right)\right)}_{\mathcal{L}_{\mathrm{prior}}} - \underbrace{\sum_{t=1}^T \log p_\theta\left(\mathbf{x}|\mathbf{x}^t,c\right)}_{\mathcal{L}_{\mathrm{rec}}}
$$
$$
+ \underbrace{\sum_{t=1}^T D_{\mathrm{KL}}\left( q_\phi\left(\mathbf{x}^{t-1}|\mathbf{x}^t,\mathbf{x},c\right) \| p_\theta\left(\mathbf{x}^{t-1}|\mathbf{x}^t,c\right)\right)}_{\mathcal{L}_{\mathrm{diff}}} \Bigg].
\tag{14}
$$

## D.3 Variational Objective

To calculate the diffusion term $\mathcal{L}_{diff}$ of the objective (2), we need to compute the KL divergence between the forward posterior distribution $q_\phi(\mathbf{x}^{t-1}|\mathbf{x}^t, \mathbf{x}, c)$ and the reverse distribution $p_\theta(\mathbf{x}^{t-1}|\mathbf{x}^t, c)$.

Since we use parameterization $p_\theta(\mathbf{x}^{t-1}|\mathbf{x}^t, c) = q_\phi(\mathbf{x}^{t-1}|\mathbf{x}^t, \hat{\mathbf{x}}_\theta(x^t, t), c)$, both of these distributions are normal distributions with the same variance, so we can evaluate the KL divergence between them analytically as follows:

$$D_{KL}\left(q_\phi(\mathbf{x}^{t-1}|\mathbf{x}^t, \mathbf{x}, c)\|p_\theta(\mathbf{x}^{t-1}|\mathbf{x}^t, c)\right) =$$

$$\frac{1}{2\sigma_{t-1|t}^2}\left\|\zeta_1\sqrt{\bar{\alpha}_t}(KAN_\phi(\mathbf{x}, t) - KAN_\phi(\hat{\mathbf{x}}_\theta(\mathbf{x}^t, t), t)) - (\zeta_1\sqrt{\bar{\alpha}_t} + \zeta_0)\left(KAN_\phi(\hat{\mathbf{x}}_\theta(\mathbf{x}^t, t-1), t) - KAN_\phi(\mathbf{x}, t-1)\right)\right\|_2^2.$$

$$(15)$$

We can compute the prior term as follows:

$$\begin{aligned}
D_{KL}\left(q_\phi(\mathbf{x}^T|\mathbf{x}, c)\|p(\mathbf{x}^T|c)\right) &= \frac{1}{2}\left[\log\frac{|I|}{\sigma_T^2 I} - d + \text{Tr}\{I^{-1}\sigma_T^2 I\} + \left\|\sqrt{\bar{\alpha}_T}KAN_\phi(\mathbf{x}, T) + (1 - \sqrt{\bar{\alpha}_T})c - c\right\|_2^2\right] \\
&= \frac{1}{2}\left[-d\log\sigma_T^2 + d\sigma_T^2 + \left\|\sqrt{\bar{\alpha}_T}KAN_\phi(\mathbf{x}, T) - \sqrt{\bar{\alpha}_T}c\right\|_2^2\right] \\
&= \frac{1}{2}\left(d\left(\sigma_T^2 - \log\sigma_T^2 - 1\right) + \bar{\alpha}_T\left\|KAN_\phi(\mathbf{x}, T) - c\right\|_2^2\right). \\
&= \frac{1}{2}\bar{\alpha}_T\left\|KAN_\phi(\mathbf{x}, T) - c\right\|_2^2.
\end{aligned}$$

$$(16)$$

The reconstruction term is considered for all latent variables similar to VAE(Kingma, 2013b). This formulation is inspired from Rishi et al. (2025).

## E    DIFFUSION KAN TRANSFORMER BLOCK EQUATIONS

Our Diffusion KAN Transformer replaces MLPs in the standard DiT architecture with KAN layers. The other key components of the architecture are as follows.

**Adaptive Layer Normalization with KAN Integration:** Following the success of adaLN-Zero in DiT (Peebles & Xie, 2023), we employ adaptive layer normalization (adaLN) for injecting conditional information. The conditioning vector, derived from timestep $t$ and condition $c$, modulates both the normalization parameters and the KAN activations:

$$\text{adaLN}(\mathbf{h}, \mathbf{c}) = \gamma_\mathbf{c} \odot \frac{\mathbf{h} - \mu(\mathbf{h})}{\sigma(\mathbf{h})} + \beta_\mathbf{c} \tag{17}$$

where $\gamma_\mathbf{c}$ and $\beta_\mathbf{c}$ are regression outputs from the conditioning vector $\mathbf{c}$.

**Temporal Conditioning Mechanism:** For time series analysis, we enhance the conditioning mechanism to specifically handle temporal dependencies. The timestep embedding $t$ undergoes sinusoidal encoding followed by

$$\mathbf{e}_t = (\text{SinusoidalEmbed}(t)) \tag{18}$$

**Scale and Shift Operations:** We incorporate additional scale and shift operators within each transformer block, similar to recent advances in diffusion transformers (Liu et al., 2024a). These operations provide fine-grained control over feature magnitudes and enable more stable training dynamics as mentioned in Section 2.2.

The complete Diffusion KAN Transformer block can be expressed as:

$$\mathbf{h}' = \text{adaLN}(\mathbf{h}, \mathbf{c}) \tag{19}$$

$$\mathbf{h}_{\text{attn}} = \text{MultiHeadAttention}(\mathbf{h}') + \mathbf{h} \tag{20}$$

$$\mathbf{h}_{\text{norm}} = \text{adaLN}(\mathbf{h}_{\text{attn}}, \mathbf{c}) \tag{21}$$

$$\mathbf{h}_{\text{out}} = \text{KAN}(\mathbf{h}_{\text{norm}}) + \mathbf{h}_{\text{attn}} \tag{22}$$

## F    ALGORITHM

The training and inference algorithms used in our paper are as follows.

---

**Algorithm 1** Training

---

**Input:** $\mathbf{x}(x_{1:H}), x_{-L+1:0}$
**repeat**
  $\epsilon \sim \mathcal{N}(0, I), t \sim U[1, T]$
  $\mathbf{x}^t \sim q_\phi(\mathbf{x}^t|\mathbf{x}, c)$
  Compute the loss in Eq. (2)
  Take numerical optimization step on: $\nabla \mathcal{L}_{\text{DiffKANformer}}$
**until** until converged

---

**Algorithm 2** Inference

---

$\mathbf{x}^T \sim \mathcal{N}(c, I)$
**for** $t = T$ **to** 1 **do**
  $\hat{\mathbf{x}} = \hat{\mathbf{x}}_\theta(\mathbf{x}^t, t, c)$
  $\mathbf{x}^{t-1} = \zeta_1 \mathbf{x}^t + \zeta_2 c - KAN_\phi(\hat{\mathbf{x}}, t)\zeta_1\sqrt{\bar{\alpha}_t} + KAN_\phi(\hat{\mathbf{x}}, t-1)(\zeta_1\sqrt{\bar{\alpha}_t} + \zeta_0) + \sigma_{t-1|t}^2 \epsilon$
**end for**
$\hat{\mathbf{x}} \sim p(\mathbf{x}|\mathbf{x}_0, c)$

---

**Algorithm 3** DDPM Inference

---

1: $\mathbf{x}^T \sim \mathcal{N}(0, I)$
2: **for** $t = T$ **to** 1 **do**
3:   $\hat{\mathbf{x}} = \hat{\mathbf{x}}_\theta(\mathbf{x}^t, t)$
4:   $\mathbf{x}^{t-1} = \frac{1}{\sqrt{\alpha_t}}\left(\mathbf{x}^t - \frac{1-\alpha_t}{\sqrt{1-\bar{\alpha}_t}}\hat{\mathbf{x}}\right) + \sigma_t\epsilon$
5: **end for**
6: **return** $\mathbf{x}_0$

---

## G EXTENDED DATASET DETAILS

### G.1 FORECASTING

These datasets are *(1)* Caiso[3] - hourly electricity loads over eight years from various California regions; *(2)* Traffic[4] - hourly road occupancy rates from San Francisco Bay area sensors; *(3)* Electricity[5] - hourly electricity use of 321 clients over two years; *(4)* Weather[6] - 10-minute meteorological data for 2020-2021; *(5)* NorPool[7] - eight years of hourly energy production data from various European countries; *(6)* Exchange[8] - daily exchange rates for eight countries: Australia, United Kingdom, Canada, Switzerland, China, Japan, New Zealand and Singapore Lai et al. (2018); *(7)* ETTh1 and *(8)* ETTm1[9] are benchmarks for China's electricity transformer temperature data, over two years, ETTh1 reported hourly and ETTm1 every 15 minutes Zhou et al. (2021). Additional details regarding the dataset's characteristics are available in Table 8.

---

[3]https://www.energyonline.com/Data/
[4]https://pems.dot.ca.gov/
[5]https://archive.ics.uci.edu/ml/datasets/ElectricityLoadDiagrams20112014
[6]https://www.bgc-jena.mpg.de/wetter/
[7]https://data.nordpoolgroup.com/power-system/production/
[8]https://github.com/laiguokun/multivariate-time-series-data
[9]https://github.com/zhouhaoyi/ETDataset

Following Shen et al. (2024), we adopt a train:validation:test ratio of 7:1:2 for Exchange, Weather, Wind, Traffic and Electricity data sets, and 6:2:2 for ETTh1 and Ettm1 data sets. For Norpool, the training data consists of observations before April 1, 2020; validation data spans from April 1 to October 1, 2020; and testing data is after October 1, 2020. For Caiso, the training set includes data before January 1, 2020, the validation period extends from January 1 to October 1, 2020, and the test set is post-October 1, 2020. Given that data sets exhibit different sampling intervals, we focus on prediction tasks with suitable prediction lengths based on the characteristics of the dataset (Shen & Kwok, 2023; Shen et al., 2024).

Table 8: Dataset characteristics for Forecasting

| dataset | dim | #observations | freq. | $H$ (steps) |
|---|---|---|---|---|
| NorPool | 18 | 70,128 | 1 hour | 1 month (720) |
| Caiso | 10 | 74,472 | 1 hour | 1 month (720) |
| Weather | 21 | 52,696 | 10 mins | 1 week (672) |
| ETTm1 | 7 | 69,680 | 15 mins | 2 days (192) |
| Wind | 7 | 48,673 | 15 mins | 2 days (192) |
| Traffic | 862 | 17,544 | 1 hour | 1 week (168) |
| Electricity | 321 | 26,304 | 1 hour | 1 week (168) |
| ETTh1 | 7 | 17,420 | 1 hour | 1 week (168) |
| Exchange | 8 | 7,588 | 1 day | 2 weeks (14) |

### G.2 IMPUTATION, CLASSIFICATION AND ANOMALY DETECTION

The descriptions of the datasets utilized for Imputation, Classification, and Anomaly Detection are presented in Table 9. These configurations are derived from TSLib (Wu et al., 2022; Wang et al., 2024b).

Table 9: Dataset descriptions. The dataset size is organized in (Train, Validation, Test).

| Tasks | Dataset | Dim | Series Length | Dataset Size | Information (Frequency) |
|---|---|---|---|---|---|
| Imputation | ETTm1, ETTm2 | 7 | 96 | (34465, 11521, 11521) | Electricity (15 mins) |
| | ETTh1, ETTh2 | 7 | 96 | (8545, 2881, 2881) | Electricity (15 mins) |
| | Electricity | 321 | 96 | (18317, 2633, 5261) | Electricity (15 mins) |
| | Weather | 21 | 96 | (36792, 5217, 10540) | Weather (10 mins) |
| Classification (UEA) | EthanolConcentration | 3 | 1751 | (261, 0, 263) | Alcohol Industry |
| | FaceDetection | 144 | 62 | (5890, 0, 3524) | Face (250Hz) |
| | Handwriting | 3 | 152 | (150, 0, 850) | Handwriting |
| | Heartbeat | 61 | 405 | (204, 0, 205) | Heart Beat |
| | JapaneseVowels | 12 | 29 | (270, 0, 370) | Voice |
| | PEMS-SF | 963 | 144 | (267, 0, 173) | Transportation (Daily) |
| | SelfRegulationSCP1 | 6 | 896 | (268, 0, 293) | Health (256Hz) |
| | SelfRegulationSCP2 | 7 | 1152 | (200, 0, 180) | Health (256Hz) |
| | SpokenArabicDigits | 13 | 93 | (6599, 0, 2199) | Voice (1025Hz) |
| | UWaveGestureLibrary | 3 | 315 | (120, 0, 320) | Gesture |
| Anomaly Detection | SMD | 38 | 100 | (566724, 141681, 708420) | Server Machine |
| | MSL | 55 | 100 | (14653, 11664, 73729) | Spacecraft |
| | SMAP | 25 | 100 | (108146, 27037, 427617) | Spacecraft |
| | SWaT | 51 | 100 | (396000, 99000, 449919) | Infrastructure |
| | PSM | 25 | 100 | (105984, 26497, 87841) | Server Machine |

## H BASELINES

We have extensively evaluated our DiffKANformer model by benchmarking it against various baseline models using a range of methodologies across different tasks, from basis expansion methods to diffusion techniques. These evaluations include: (i) models based on diffusion (Shen et al., 2024; Shen & Kwok, 2023; Rasul et al., 2021; Tashiro et al., 2021; Lopez Alcaraz & Strodthoff, 2023; Rishi et al., 2025); (ii) basis expansion methods, as described in (Challu et al., 2023; Zhou et al., 2022a; Fan et al., 2022; Oreshkin et al., 2019); (iii) other generative approaches such as GAN and VAE, referenced in (Li et al., 2022; Rangapuram et al., 2023; Jeha et al., 2022). We have also compared DiffKANformer with transformer models (Shabani et al., 2023; Nie et al., 2023; Zhou et al.,

2022b; Wu et al., 2021; Liu et al., 2022b; Zhou et al., 2021; Kitaev et al., 2020; Liu et al., 2022c; Woo et al., 2022; Wang et al., 2024c; Tan et al., 2024; Naghashi et al., 2025; Liu et al., 2023b; Wu et al., 2022; Liu et al., 2022c; Das et al., 2023a; Woo et al., 2022; Kitaev et al., 2020; Zhang & Yan, 2023) and additional deep learning methodologies (Liu et al., 2022a; Zeng et al., 2023; Bahdanau et al., 2015; Zhou et al., 2024; Chen et al., 2023; Wang et al., 2024a; 2023; Zhang et al., 2022; Xu et al., 2021).

# I IMPLEMENTATION DETAILS

The MLP modules within the embeddings of the condition and noised input, as well as the decoder, are from TiDE (Das et al., 2023a). This MLP framework is acknowledged as an effective component for universal time series analysis models (Das et al., 2023b). In the diffusion transformer block (DiT), MLP is replaced with KAN, with the diffusion time step adjusted by adaptive layer normalization (Peebles & Xie, 2023; Esser et al., 2024). Hyperparameters are tuned specifically regarding embedding layers, encoder layers, time steps and hidden dimensions while other hyperparameters remain constant as detailed in Section 3. Optimal hyperparameters are selected via cross-validation and summarized for various datasets and tasks in Table 10. In the condition network, for condition $c$, a dense layer is used for tasks such as forecasting and anomaly detection, while a transformer is utilized for imputation and classification tasks. The implementation of KAN is inspired by Yang & Wang (2024).

Table 10: Best hyperparameters for all our time series tasks.

| Task | Dataset | Timesteps | Hidden Dim | Depth | Emb | Heads |
|---|---|---|---|---|---|---|
| Forecasting | weather | 100 | 256 | 1 | 4 | 8 |
| | Caiso | 100 | 128 | 3 | 5 | 8 |
| | ETTh1 | 100 | 128 | 3 | 5 | 8 |
| | Exchange | 200 | 64 | 2 | 2 | 4 |
| | ECL | 100 | 256 | 3 | 5 | 8 |
| | Norpool | 200 | 256 | 2 | 1 | 4 |
| | ETTm1 | 100 | 128 | 1 | 4 | 8 |
| | Traffic | 200 | 256 | 3 | 2 | 8 |
| Imputation | ETTh1 | 200 | 128 | 2 | 1 | 1 |
| | ETTh2 | 200 | 128 | 2 | 1 | 1 |
| | ETTm2 | 200 | 128 | 2 | 1 | 1 |
| | ECL | 200 | 128 | 2 | 1 | 4 |
| | ETTm1 | 200 | 128 | 2 | 1 | 8 |
| | weather | 200 | 32 | 2 | 1 | 4 |
| Classification | EthanolConcentration | 200 | 128 | 4 | 4 | 8 |
| | FaceDetection | 200 | 256 | 1 | 4 | 8 |
| | Handwriting | 100 | 256 | 1 | 1 | 8 |
| | Heartbeat | 200 | 256 | 4 | 4 | 1 |
| | PEMS-SF | 200 | 256 | 1 | 1 | 8 |
| | SelfRegulationSCP2 | 100 | 256 | 4 | 1 | 8 |
| | SpokenArabicDigits | 200 | 256 | 1 | 1 | 8 |
| | UWaveGestureLibrary | 200 | 128 | 2 | 1 | 8 |
| | JapaneseVowels | 1000 | 32 | 1 | 1 | 8 |
| | SelfRegulationSCP1 | 200 | 256 | 1 | 1 | 8 |
| Anomaly Detection | MSL | 100 | 32 | 4 | 1 | 4 |
| | PSM | 100 | 64 | 1 | 1 | 8 |
| | SMAP | 100 | 64 | 4 | 1 | 4 |
| | SMD | 100 | 32 | 1 | 1 | 1 |
| | SWaT | 100 | 32 | 4 | 4 | 4 |

## J  SENSITIVITY ANALYSIS

### J.1  DIFFUSION HYPERPARAMETERS

Table 11 presents the metrics for different tasks involving DiffKANformer, dependent on T (diffusion timesteps). In Classification, the accuracy is evaluated, along with Forecasting via MSE and MAE, and for Anomaly Detection, both Accuracy and F1 score are assessed. Notably, setting T = 200 leads to improved outcomes in both classification and anomaly detection. Meanwhile, T = 100 is more effective for forecasting.

Table 11: Ablation Study with varying diffusion time steps

| Diffusion time | Classification (Ethanol) | Anomaly Detection (MSL) | | Forecasting (ETTm1) | |
|---|---|---|---|---|---|
| Datasets | Accuracy | Acc | F1 | MSE | MAE |
| 50 | 0.261 | 0.969 | 0.839 | 0.344 | 0.380 |
| 100 | 0.261 | 0.965 | 0.815 | **0.337** | **0.375** |
| 200 | **0.316** | **0.980** | **0.903** | 0.339 | 0.377 |
| 500 | 0.234 | 0.979 | 0.899 | 0.351 | 0.390 |

A quadratic variance schedule is employed to establish the variance of Gaussian noise $\beta_k$. In this study, the parameter $\beta_T$ is selected from the set {0.001, 0.01, 0.1, 0.9}, while $\beta_1$ remains fixed at $10^{-4}$. As shown in Table 12, the optimal setting with $\beta_T = 0.1$ consistently yields improved results.

Table 12: Ablation study by varying variance upper bound $(\beta_T)$

| Tasks (Datasets) | Classification (Ethanol) | Anomaly Detection (MSL) | | Forecasting (ETTm1) | |
|---|---|---|---|---|---|
| $\beta_T$ Metric | Accuracy | Acc | F1 | MSE | MAE |
| 0.001 | 0.246 | 0.974 | 0.868 | 0.345 | 0.381 |
| 0.01 | 0.261 | 0.973 | 0.867 | 0.339 | 0.376 |
| 0.1 | **0.316** | **0.980** | **0.879** | **0.337** | **0.375** |
| 0.9 | 0.250 | 0.964 | 0.806 | 0.337 | 0.376 |

### J.2  LOOK BACK WINDOW FOR FORECASTING

Table 13 presents the prediction MSE and MAE for DiffKANformer utilizing various lengths of the lookback window. The values considered are L = {96, 192, 336, 720, 1440}. It is evident that on the datasets ETTh1, and weather (corresponding to 168-step, and 672-step-ahead predictions, respectively), satisfactory performance is achieved when L is set to 720.

Table 13: Predicting MSE and MAE for different Lookback windows $(L)$ for Multivariate forecasting

| Look back window | 96 | | 192 | | 336 | | 720 | | 1440 | |
|---|---|---|---|---|---|---|---|---|---|---|
| Datasets | MSE | MAE | MSE | MAE | MSE | MAE | MSE | MAE | MSE | MAE |
| ETTh1 | 0.518 | 0.487 | 0.445 | 0.443 | 0.428 | 0.439 | **0.401** | **0.425** | 0.426 | 0.443 |
| Weather | 0.365 | 0.368 | 0.342 | 0.362 | 0.331 | 0.361 | **0.307** | **0.336** | 0.394 | 0.412 |

## K  COMPUTATIONAL ANALYSIS

We evaluate the computational efficiency of our proposed DiffKANformer in terms of training time, inference time, and model size, with results summarized in Tables 14 and 15. Although the current

KAN implementation is still in its early stage and not fully parallelizable, leading to moderately higher training times compared to the most optimized baselines, this limitation is primarily due to immature implementation rather than deficiencies of the architecture itself. With more efficient and parallelized implementations, the training efficiency of KAN-based models is expected to improve substantially. Importantly, DiffKANformer achieves competitive accuracy while maintaining a significantly smaller parameter footprint (0.5M parameters), which translates into faster inference and lower memory requirements compared to most diffusion-based baselines. These results highlight the potential of DiffKANformer as a lightweight yet effective framework for time series analysis, offering a promising trade-off between accuracy, efficiency, and scalability.

Table 14: Training time (ms) for different models and sequence lengths ($H$) for ETTh1 univariate.

|  | $H = 96$ | $H = 168$ | $H = 192$ | $H = 336$ | $H = 720$ | MSE |
|---|---|---|---|---|---|---|
| DiffKANformer | 0.92 | 0.90 | 0.97 | 0.84 | 0.71 | 0.066 |
| CN-Diff | 0.21 | 0.26 | 0.27 | 0.31 | 0.39 | 0.066 |
| mr-Diff | 0.59 | 0.69 | 0.71 | 0.74 | 0.82 | 0.066 |
| TimeDiff | 0.71 | 0.75 | 0.77 | 0.82 | 0.85 | 0.066 |
| TimeGrad | 2.11 | 2.42 | 3.21 | 4.22 | 5.93 | 0.078 |
| CSDI | 5.72 | 7.09 | 7.59 | 10.59 | 17.21 | 0.083 |
| SSSD | 16.98 | 19.34 | 22.64 | 32.12 | 52.93 | 0.097 |

Table 15: Inference time (ms) for different models and sequence lengths ($H$) for ETTh1 univariate.

| Model | Trainable Params | $H = 96$ | $H = 168$ | $H = 192$ | $H = 336$ | $H = 720$ |
|---|---|---|---|---|---|---|
| DiffKANformer | 0.5M | 10.1 | 9.9 | 9.8 | 10.0 | 9.8 |
| CN-Diff | 1.1M | 6.2 | 6.7 | 6.9 | 7.8 | 9.1 |
| mr-Diff | 1.4M | 12.5 | 14.3 | 14.9 | 16.8 | 27.5 |
| TimeDiff | 1.7M | 16.2 | 17.3 | 17.6 | 26.5 | 34.6 |
| TimeGrad | 3.1M | 870.2 | 1620.9 | 1854.5 | 3119.7 | 6724.1 |
| CSDI | 10M | 90.4 | 128.3 | 142.8 | 398.9 | 513.1 |
| SSSD | 32M | 418.6 | 590.2 | 645.4 | 1054.2 | 2516.9 |

## L  STATISTICAL SIGNIFICANCE ANALYSIS

Acknowledging that the performance of deep models in time series analysis can be affected by various random initializations. Table 16 presents the outcomes from various runs across several tasks, including forecasting, imputation, anomaly detection, and classification. In the forecasting task (ETTh1) and imputation task (ETTm1), we measure performance using the mean squared error (MSE). For anomaly detection (MSL), we utilize the F1-score for evaluation, and for classification (Ethanol Concentration), accuracy is employed as the metric. We note that classification accuracy and the F1-score for anomaly detection are consistent across different runs, indicating stable and reliable performance reflecting that multiple runs are predicting same categories. Conversely, the MSE for forecasting and imputation exhibits minor fluctuations across runs, highlighting the statistical significance of our model.

## M  FULL RESULTS

### M.1  FORECASTING

Table 17 illustrates the mean absolute error (MAE) outcomes for multivariate time series prediction tasks, with an average rank of 2.6. It is evident that DiffKANformer continues to demonstrate a superior overall performance for MAE as the evaluation metric.

Table 16: Statistical significance analysis across five different runs over four different task

| | Forecasting | Imputation | Anomaly Detection | Classification |
|---|---|---|---|---|
| Datasets(metric) | ETTh1(MSE) | ETTm1(MSE) | MSL(F1) | Ethanol Conc(Accuracy) |
| 0 | 0.4016211 | 0.0214681 | 0.9038665 | 0.3822115 |
| 1 | 0.4016621 | 0.0213752 | 0.9038665 | 0.3822115 |
| 2 | 0.4016901 | 0.0215876 | 0.9038665 | 0.3822115 |
| 3 | 0.4016121 | 0.0209875 | 0.9038665 | 0.3822115 |
| 4 | 0.4016319 | 0.0223561 | 0.9038665 | 0.3822115 |
| mean | 0.40164346 | 0.0215549 | 0.9038665 | 0.3822115 |
| std deviation | 0.0000287742 | 0.0004482766 | 0.0 | 0.0 |

Table 17: Multivariate prediction of MAEs on eight real-world time series datasets (subscripts are the rank). CSDI and TiDE run out of memory on Traffic and Electricity. Results of all baselines are from Shen et al. (2024). Baselines with * are run using TSLib (Wang et al., 2024b; Wu et al., 2022)

| Method | NorPool | Caiso | Traffic | Electricity | Weather | Exchange | ETTh1 | ETTm1 | Rank |
|---|---|---|---|---|---|---|---|---|---|
| Ours | $\underline{0.565}_{(2)}$ | $\mathbf{0.189}_{(1)}$ | $\underline{0.268}_{(2)}$ | $\underline{0.243}_{(2)}$ | $0.325_{(4)}$ | $\mathbf{0.079}_{(1)}$ | $0.425_{(7)}$ | $\underline{0.372}_{(2)}$ | 2.6 |
| CnDiff* | $\mathbf{0.554}_{(1)}$ | $\underline{0.191}_{(2)}$ | $0.270_{(6)}$ | $\underline{0.243}_{(2)}$ | $\underline{0.324}_{(2)}$ | $\mathbf{0.079}_{(1)}$ | $0.421_{(3)}$ | $0.378_{(7)}$ | 3.0 |
| mr-Diff | $0.604_{(5)}$ | $0.219_{(5)}$ | $0.320_{(14)}$ | $0.252_{(6)}$ | $\underline{0.324}_{(2)}$ | $0.082_{(7)}$ | $0.422_{(4)}$ | $0.373_{(3)}$ | 5.8 |
| TimeDiff | $0.611_{(6)}$ | $0.234_{(12)}$ | $0.384_{(20)}$ | $0.305_{(18)}$ | $\mathbf{0.312}_{(1)}$ | $0.091_{(16)}$ | $0.430_{(10)}$ | $\mathbf{0.372}_{(1)}$ | 10.5 |
| TimeGrad | $0.821_{(35)}$ | $0.339_{(34)}$ | $0.849_{(37)}$ | $0.630_{(37)}$ | $0.381_{(27)}$ | $0.193_{(34)}$ | $0.719_{(37)}$ | $0.605_{(37)}$ | 34.8 |
| CSDI | $0.777_{(32)}$ | $0.345_{(35)}$ | – | – | $0.374_{(25)}$ | $0.194_{(35)}$ | $0.438_{(12)}$ | $0.442_{(27)}$ | 27.7 |
| SSSD | $0.753_{(26)}$ | $0.295_{(25)}$ | $0.398_{(27)}$ | $0.363_{(27)}$ | $0.350_{(16)}$ | $0.127_{(25)}$ | $0.561_{(32)}$ | $0.406_{(21)}$ | 24.9 |
| D3VAE | $0.692_{(14)}$ | $0.331_{(32)}$ | $0.483_{(31)}$ | $0.372_{(29)}$ | $0.380_{(26)}$ | $0.301_{(37)}$ | $0.502_{(26)}$ | $0.391_{(15)}$ | 26.2 |
| CPF | $0.889_{(37)}$ | $0.424_{(37)}$ | $0.714_{(36)}$ | $0.643_{(38)}$ | $0.781_{(39)}$ | $0.082_{(7)}$ | $0.597_{(35)}$ | $0.472_{(29)}$ | 32.2 |
| PSA-GAN | $0.890_{(38)}$ | $0.477_{(38)}$ | $0.697_{(35)}$ | $0.533_{(36)}$ | $0.578_{(37)}$ | $0.087_{(14)}$ | $0.546_{(30)}$ | $0.488_{(33)}$ | 32.6 |
| N-Hits | $0.643_{(10)}$ | $0.221_{(6)}$ | $\underline{0.268}_{(2)}$ | $0.245_{(4)}$ | $0.335_{(7)}$ | $0.085_{(10)}$ | $0.480_{(20)}$ | $0.388_{(12)}$ | 8.9 |
| FiLM | $0.646_{(11)}$ | $0.278_{(22)}$ | $0.398_{(27)}$ | $0.320_{(20)}$ | $0.336_{(8)}$ | $\mathbf{0.079}_{(1)}$ | $0.436_{(11)}$ | $0.374_{(4)}$ | 13.0 |
| Depts | $0.611_{(6)}$ | $0.204_{(4)}$ | $0.568_{(34)}$ | $0.401_{(32)}$ | $0.394_{(30)}$ | $0.100_{(18)}$ | $0.491_{(24)}$ | $0.412_{(23)}$ | 21.4 |
| NBeats | $0.832_{(36)}$ | $0.235_{(13)}$ | $\mathbf{0.265}_{(1)}$ | $0.370_{(28)}$ | $0.420_{(31)}$ | $0.081_{(6)}$ | $0.521_{(27)}$ | $0.409_{(24)}$ | 20.5 |
| Scaleformer | $0.769_{(30)}$ | $0.310_{(27)}$ | $0.379_{(18)}$ | $0.304_{(17)}$ | $0.438_{(32)}$ | $0.138_{(27)}$ | $0.579_{(34)}$ | $0.475_{(30)}$ | 26.9 |
| PatchTST | $0.710_{(20)}$ | $0.293_{(24)}$ | $0.411_{(30)}$ | $0.348_{(26)}$ | $0.555_{(36)}$ | $0.147_{(28)}$ | $0.489_{(23)}$ | $0.392_{(16)}$ | 25.4 |
| FedFormer | $0.744_{(23)}$ | $0.317_{(29)}$ | $0.385_{(21)}$ | $0.341_{(24)}$ | $0.347_{(14)}$ | $0.233_{(36)}$ | $0.484_{(21)}$ | $0.413_{(24)}$ | 24.0 |
| Autoformer | $0.751_{(25)}$ | $0.321_{(30)}$ | $0.392_{(26)}$ | $0.313_{(19)}$ | $0.354_{(17)}$ | $0.167_{(30)}$ | $0.484_{(21)}$ | $0.496_{(34)}$ | 25.2 |
| Pyraformer | $0.781_{(33)}$ | $0.371_{(36)}$ | $0.390_{(22)}$ | $0.379_{(30)}$ | $0.385_{(28)}$ | $0.112_{(23)}$ | $0.493_{(25)}$ | $0.435_{(26)}$ | 27.9 |
| Informer | $0.757_{(27)}$ | $0.336_{(33)}$ | $0.391_{(24)}$ | $0.383_{(31)}$ | $0.364_{(21)}$ | $0.192_{(33)}$ | $0.605_{(36)}$ | $0.542_{(35)}$ | 30.0 |
| Transformer | $0.765_{(28)}$ | $0.321_{(30)}$ | $0.410_{(29)}$ | $0.405_{(33)}$ | $0.370_{(24)}$ | $0.178_{(32)}$ | $0.567_{(33)}$ | $0.592_{(36)}$ | 30.6 |
| Crossformer | $0.704_{(19)}$ | $0.225_{(8)}$ | $0.269_{(5)}$ | $0.247_{(5)}$ | $0.369_{(23)}$ | $0.170_{(31)}$ | $0.459_{(17)}$ | $0.390_{(14)}$ | 15.2 |
| NonStationary* | $0.581_{(3)}$ | $0.232_{(11)}$ | $0.382_{(19)}$ | $0.281_{(14)}$ | $0.356_{(18)}$ | $0.105_{(22)}$ | $0.528_{(28)}$ | $0.443_{(28)}$ | 17.9 |
| Timexer* | $0.647_{(12)}$ | $0.230_{(10)}$ | $0.275_{(7)}$ | $0.252_{(8)}$ | $0.333_{(6)}$ | $0.086_{(11)}$ | $0.429_{(9)}$ | $0.376_{(6)}$ | 8.6 |
| Timesnet* | $0.696_{(15)}$ | $0.239_{(14)}$ | $0.320_{(15)}$ | $0.281_{(15)}$ | $0.349_{(15)}$ | $0.103_{(21)}$ | $0.456_{(16)}$ | $0.394_{(18)}$ | 16.1 |
| Multipatchformer* | $0.701_{(18)}$ | $0.245_{(16)}$ | $0.292_{(8)}$ | $0.256_{(10)}$ | $0.342_{(11)}$ | $0.086_{(13)}$ | $\underline{0.420}_{(2)}$ | $0.387_{(11)}$ | 11.1 |
| PAttn* | $0.698_{(17)}$ | $0.248_{(17)}$ | $0.317_{(12)}$ | $0.267_{(11)}$ | $0.341_{(10)}$ | $0.080_{(5)}$ | $0.429_{(8)}$ | $0.384_{(10)}$ | 11.2 |
| Reformer* | $0.717_{(22)}$ | $0.272_{(21)}$ | $0.390_{(23)}$ | $0.409_{(34)}$ | $0.613_{(38)}$ | $0.650_{(39)}$ | $0.740_{(38)}$ | $0.689_{(38)}$ | 31.6 |
| iTransformer* | $0.711_{(21)}$ | $0.264_{(19)}$ | $0.292_{(9)}$ | $0.252_{(7)}$ | $0.346_{(13)}$ | $0.086_{(12)}$ | $0.439_{(13)}$ | $0.397_{(19)}$ | 14.1 |
| ETSFormer* | $0.783_{(34)}$ | $0.291_{(23)}$ | $0.525_{(33)}$ | $0.341_{(25)}$ | $0.500_{(34)}$ | $0.112_{(24)}$ | $0.555_{(31)}$ | $0.484_{(32)}$ | 29.5 |
| TiDE* | $0.745_{(24)}$ | $0.315_{(28)}$ | – | $0.291_{(16)}$ | $0.392_{(29)}$ | $0.079_{(4)}$ | $0.424_{(6)}$ | $0.399_{(20)}$ | 18.1 |
| SCINet | $0.601_{(4)}$ | $0.193_{(3)}$ | $0.335_{(16)}$ | $0.280_{(13)}$ | $0.344_{(12)}$ | $0.137_{(26)}$ | $0.463_{(19)}$ | $0.389_{(13)}$ | 13.2 |
| NLinear | $0.636_{(8)}$ | $0.223_{(7)}$ | $0.293_{(10)}$ | $\mathbf{0.239}_{(1)}$ | $0.328_{(5)}$ | $0.091_{(16)}$ | $\mathbf{0.418}_{(1)}$ | $0.375_{(5)}$ | 6.6 |
| DLinear | $0.640_{(9)}$ | $0.497_{(39)}$ | $\underline{0.268}_{(2)}$ | $0.336_{(22)}$ | $0.444_{(33)}$ | $0.102_{(19)}$ | $0.442_{(14)}$ | $0.378_{(7)}$ | 18.1 |
| LSTMa | $0.974_{(39)}$ | $0.305_{(26)}$ | $0.510_{(32)}$ | $0.444_{(35)}$ | $0.501_{(35)}$ | $0.534_{(38)}$ | $0.782_{(39)}$ | $0.699_{(39)}$ | 35.4 |
| MICN* | $0.682_{(13)}$ | $0.230_{(9)}$ | $0.306_{(11)}$ | $0.277_{(12)}$ | $0.365_{(22)}$ | $0.089_{(15)}$ | $0.449_{(15)}$ | $0.393_{(17)}$ | 14.2 |
| TSMixer* | $0.774_{(31)}$ | $0.244_{(15)}$ | $0.369_{(17)}$ | $0.336_{(23)}$ | $0.357_{(19)}$ | $0.149_{(29)}$ | $0.538_{(29)}$ | $0.477_{(31)}$ | 24.2 |
| LightTS* | $0.765_{(29)}$ | $0.253_{(18)}$ | $0.391_{(25)}$ | $0.327_{(21)}$ | $0.362_{(20)}$ | $0.102_{(20)}$ | $0.462_{(18)}$ | $0.418_{(25)}$ | 22.0 |
| TimeMixer* | $0.698_{(16)}$ | $0.265_{(20)}$ | $0.319_{(13)}$ | $0.255_{(9)}$ | $0.341_{(9)}$ | $0.084_{(9)}$ | $0.422_{(5)}$ | $0.381_{(9)}$ | 11.2 |

## M.2 IMPUTATION

Table 18 and 19 provides a comprehensive analysis of the MSE and MAE of several imputation models across various datasets with differing levels of missing data (12.5%, 25%, 37.5% and 50%). Our model's excellence is evident in the "Avg Rank" row, where it secures the top position with

a rank of 1.5. These findings collectively underscore the robustness of our imputation method, particularly as the proportion of missing data increases.

Table 18: Full results for the imputation task with MSE metrics. We randomly mask 12.5%, 25%, 37.5% and 50% time points to compare the model performance under different missing degrees. + in the Transformers indicates the name of + former .

| Dataset | | Ours | NS+ | TimesNet | Cross+ | LightTS | iTrans+ | SSSD | Pyra+ | Re+ | Trans+ | TiDE | DLinear | In+ | ETS* | FED* | FiLM | Auto+ |
|---|---|---|---|---|---|---|---|---|---|---|---|---|---|---|---|---|---|---|
| ECL | 0.125 | **0.060** | 0.078 | 0.088 | 0.079 | 0.077 | 0.073 | 0.197 | 0.177 | 0.148 | 0.149 | 0.085 | 0.085 | 0.149 | 0.117 | 0.184 | 0.085 | 0.196 |
| | 0.25 | **0.070** | 0.085 | 0.091 | 0.092 | 0.099 | 0.090 | 0.205 | 0.184 | 0.155 | 0.161 | 0.114 | 0.113 | 0.165 | 0.127 | 0.206 | 0.114 | 0.215 |
| | 0.375 | **0.079** | 0.091 | 0.096 | 0.104 | 0.119 | 0.107 | 0.214 | 0.190 | 0.166 | 0.170 | 0.142 | 0.141 | 0.179 | 0.141 | 0.225 | 0.142 | 0.242 |
| | 0.5 | **0.092** | 0.099 | 0.102 | 0.114 | 0.137 | 0.128 | 0.217 | 0.204 | 0.178 | 0.178 | 0.174 | 0.173 | 0.193 | 0.158 | 0.247 | 0.174 | 0.269 |
| | Avg | **0.075** | 0.088 | 0.094 | 0.097 | 0.108 | 0.099 | 0.208 | 0.189 | 0.162 | 0.165 | 0.129 | 0.128 | 0.171 | 0.136 | 0.215 | 0.129 | 0.231 |
| ETTh1 | 0.125 | 0.047 | 0.047 | 0.062 | 0.120 | 0.114 | 0.098 | **0.046** | 0.077 | 0.060 | 0.065 | 0.111 | 0.111 | 0.071 | 0.086 | 0.072 | 0.117 | 0.094 |
| | 0.25 | **0.052** | 0.064 | 0.080 | 0.139 | 0.146 | 0.125 | 0.065 | 0.102 | 0.084 | 0.118 | 0.150 | 0.150 | 0.095 | 0.118 | 0.103 | 0.154 | 0.118 |
| | 0.375 | **0.077** | 0.083 | 0.097 | 0.156 | 0.175 | 0.156 | 0.090 | 0.128 | 0.113 | 0.113 | 0.186 | 0.188 | 0.127 | 0.152 | 0.140 | 0.191 | 0.160 |
| | 0.5 | **0.083** | 0.107 | 0.116 | 0.176 | 0.203 | 0.212 | 0.110 | 0.156 | 0.146 | 0.135 | 0.229 | 0.229 | 0.165 | 0.196 | 0.197 | 0.233 | 0.217 |
| | Avg | **0.065** | 0.075 | 0.089 | 0.148 | 0.159 | 0.148 | 0.078 | 0.116 | 0.101 | 0.108 | 0.169 | 0.170 | 0.114 | 0.138 | 0.128 | 0.174 | 0.147 |
| ETTh2 | 0.125 | **0.029** | 0.039 | 0.040 | 0.126 | 0.113 | 0.095 | 0.184 | 0.141 | 0.144 | 0.182 | 0.108 | 0.108 | 0.279 | 0.132 | 0.127 | 0.110 | 0.212 |
| | 0.25 | **0.025** | 0.046 | 0.047 | 0.143 | 0.140 | 0.119 | 0.336 | 0.186 | 0.201 | 0.252 | 0.143 | 0.144 | 0.354 | 0.189 | 0.176 | 0.150 | 0.219 |
| | 0.375 | **0.037** | 0.052 | 0.054 | 0.155 | 0.166 | 0.148 | 0.477 | 0.221 | 0.246 | 0.281 | 0.179 | 0.179 | 0.402 | 0.264 | 0.247 | 0.188 | 0.274 |
| | 0.5 | **0.055** | 0.060 | 0.061 | 0.181 | 0.189 | 0.192 | 0.642 | 0.257 | 0.313 | 0.308 | 0.221 | 0.217 | 0.430 | 0.351 | 0.348 | 0.227 | 0.360 |
| | Avg | **0.037** | 0.049 | 0.051 | 0.151 | 0.152 | 0.139 | 0.410 | 0.201 | 0.226 | 0.256 | 0.163 | 0.162 | 0.366 | 0.234 | 0.224 | 0.169 | 0.266 |
| ETTm1 | 0.125 | 0.017 | 0.018 | 0.019 | 0.051 | 0.052 | 0.045 | **0.016** | 0.027 | 0.024 | 0.022 | 0.056 | 0.056 | 0.028 | 0.042 | 0.034 | 0.056 | 1.103 |
| | 0.25 | 0.021 | 0.022 | 0.023 | 0.054 | 0.062 | 0.060 | **0.020** | 0.034 | 0.033 | 0.030 | 0.076 | 0.076 | 0.040 | 0.061 | 0.051 | 0.077 | 0.811 |
| | 0.375 | 0.026 | 0.028 | 0.029 | 0.060 | 0.073 | 0.077 | **0.024** | 0.040 | 0.035 | 0.037 | 0.099 | 0.099 | 0.056 | 0.084 | 0.079 | 0.100 | 0.561 |
| | 0.5 | 0.032 | 0.034 | 0.036 | 0.067 | 0.086 | 0.101 | 0.032 | 0.049 | 0.047 | 0.045 | 0.128 | 0.128 | 0.073 | 0.104 | 0.125 | 0.129 | 0.484 |
| | Avg | 0.024 | 0.026 | 0.027 | 0.060 | 0.068 | 0.071 | **0.023** | 0.037 | 0.033 | 0.034 | 0.090 | 0.090 | 0.049 | 0.073 | 0.072 | 0.090 | 0.740 |
| ETTm2 | 0.125 | **0.011** | 0.018 | 0.019 | 0.069 | 0.062 | 0.052 | 0.088 | 0.077 | 0.105 | 0.178 | 0.066 | 0.066 | 0.174 | 0.081 | 0.060 | 0.066 | 1.059 |
| | 0.25 | **0.014** | 0.020 | 0.021 | 0.069 | 0.073 | 0.070 | 0.128 | 0.101 | 0.142 | 0.215 | 0.089 | 0.088 | 0.244 | 0.115 | 0.090 | 0.089 | 0.957 |
| | 0.375 | **0.018** | 0.022 | 0.023 | 0.087 | 0.084 | 0.090 | 0.236 | 0.142 | 0.182 | 0.206 | 0.111 | 0.111 | 0.285 | 0.163 | 0.131 | 0.112 | 0.726 |
| | 0.5 | **0.022** | 0.025 | 0.026 | 0.091 | 0.087 | 0.117 | 0.158 | 0.192 | 0.230 | 0.213 | 0.138 | 0.138 | 0.277 | 0.209 | 0.237 | 0.138 | 0.705 |
| | Avg | **0.016** | 0.021 | 0.022 | 0.079 | 0.076 | 0.082 | 0.152 | 0.128 | 0.165 | 0.203 | 0.101 | 0.101 | 0.245 | 0.142 | 0.129 | 0.102 | 0.862 |
| weather | 0.125 | 0.028 | 0.025 | **0.023** | 0.039 | 0.038 | 0.038 | 0.036 | 0.029 | 0.031 | 0.030 | 0.038 | 0.038 | 0.036 | 0.036 | 0.042 | 0.041 | 0.304 |
| | 0.25 | 0.031 | 0.027 | 0.028 | 0.042 | 0.043 | 0.046 | 0.029 | 0.032 | 0.036 | 0.034 | 0.047 | 0.047 | 0.039 | 0.044 | 0.058 | 0.050 | 0.237 |
| | 0.375 | 0.032 | 0.031 | 0.031 | 0.044 | 0.048 | 0.055 | 0.030 | 0.036 | 0.040 | 0.040 | 0.056 | 0.056 | 0.044 | 0.052 | 0.077 | 0.062 | 0.151 |
| | 0.5 | 0.036 | 0.036 | 0.034 | 0.047 | 0.053 | 0.067 | **0.033** | 0.040 | 0.048 | 0.044 | 0.066 | 0.066 | 0.050 | 0.061 | 0.114 | 0.076 | 0.148 |
| | Avg | 0.032 | 0.030 | **0.029** | 0.043 | 0.046 | 0.051 | 0.032 | 0.034 | 0.039 | 0.036 | 0.052 | 0.052 | 0.042 | 0.048 | 0.073 | 0.057 | 0.210 |
| Avg MSE | | **0.041** | 0.048 | 0.049 | 0.095 | 0.102 | 0.104 | 0.151 | 0.118 | 0.114 | 0.128 | 0.117 | 0.117 | 0.165 | 0.129 | 0.140 | 0.120 | 0.408 |
| Avg Rank | | **1.5** | 2.2 | 3.0 | 7.3 | 8.3 | 8.5 | 8.8 | 9.0 | 9.0 | 9.8 | 10.8 | 11.2 | 11.3 | 11.5 | 12.5 | 12.5 | 15.7 |

Table 19: Full results for the imputation task with MAE metrics. We randomly mask 12.5%, 25%, 37.5% and 50% time points to compare the model performance under different missing degrees. + in the Transformers indicates the name of + former .

| Dataset | | Ours | NS+ | TimesNet | Cross+ | iTrans+ | LightTS | SSSD | Re+ | Pyra+ | Trans+ | DLinear | TiDE | In+ | ETS+ | FED+ | FiLM | Auto+ |
|---|---|---|---|---|---|---|---|---|---|---|---|---|---|---|---|---|---|---|
| ECL | 0.125 | **0.168** | 0.193 | 0.203 | 0.199 | 0.190 | 0.197 | 0.304 | 0.276 | 0.298 | 0.277 | 0.207 | 0.207 | 0.276 | 0.246 | 0.322 | 0.207 | 0.333 |
| | 0.25 | **0.184** | 0.201 | 0.208 | 0.217 | 0.214 | 0.228 | 0.308 | 0.282 | 0.299 | 0.285 | 0.243 | 0.244 | 0.288 | 0.257 | 0.340 | 0.243 | 0.345 |
| | 0.375 | **0.195** | 0.209 | 0.213 | 0.231 | 0.235 | 0.252 | 0.312 | 0.290 | 0.303 | 0.291 | 0.273 | 0.274 | 0.297 | 0.270 | 0.354 | 0.274 | 0.365 |
| | 0.5 | **0.212** | 0.218 | 0.221 | 0.244 | 0.258 | 0.271 | 0.313 | 0.297 | 0.310 | 0.296 | 0.303 | 0.305 | 0.306 | 0.285 | 0.370 | 0.304 | 0.382 |
| | Avg | **0.190** | 0.205 | 0.211 | 0.223 | 0.224 | 0.237 | 0.309 | 0.286 | 0.302 | 0.287 | 0.256 | 0.257 | 0.292 | 0.264 | 0.347 | 0.257 | 0.356 |
| ETTh1 | 0.125 | 0.152 | **0.146** | 0.168 | 0.237 | 0.219 | 0.235 | 0.151 | 0.172 | 0.200 | 0.181 | 0.232 | 0.231 | 0.188 | 0.209 | 0.194 | 0.238 | 0.222 |
| | 0.25 | **0.156** | 0.169 | 0.191 | 0.258 | 0.249 | 0.267 | 0.177 | 0.205 | 0.229 | 0.245 | 0.269 | 0.268 | 0.219 | 0.245 | 0.233 | 0.273 | 0.249 |
| | 0.375 | **0.189** | 0.192 | 0.209 | 0.277 | 0.279 | 0.294 | 0.205 | 0.239 | 0.259 | 0.238 | 0.302 | 0.299 | 0.252 | 0.277 | 0.275 | 0.303 | 0.288 |
| | 0.5 | **0.195** | 0.217 | 0.227 | 0.298 | 0.326 | 0.318 | 0.232 | 0.273 | 0.285 | 0.262 | 0.332 | 0.329 | 0.287 | 0.315 | 0.327 | 0.334 | 0.337 |
| | Avg | **0.173** | 0.181 | 0.199 | 0.268 | 0.268 | 0.278 | 0.191 | 0.222 | 0.243 | 0.232 | 0.284 | 0.282 | 0.236 | 0.261 | 0.257 | 0.287 | 0.274 |
| ETTh2 | 0.125 | **0.121** | 0.130 | 0.131 | 0.237 | 0.210 | 0.227 | 0.310 | 0.268 | 0.276 | 0.311 | 0.222 | 0.222 | 0.389 | 0.256 | 0.242 | 0.224 | 0.311 |
| | 0.25 | **0.111** | 0.142 | 0.144 | 0.254 | 0.238 | 0.256 | 0.417 | 0.317 | 0.328 | 0.374 | 0.259 | 0.258 | 0.451 | 0.316 | 0.290 | 0.263 | 0.327 |
| | 0.375 | **0.136** | 0.152 | 0.155 | 0.267 | 0.265 | 0.280 | 0.501 | 0.355 | 0.350 | 0.389 | 0.289 | 0.289 | 0.480 | 0.375 | 0.340 | 0.295 | 0.359 |
| | 0.5 | 0.168 | **0.164** | 0.165 | 0.290 | 0.302 | 0.300 | 0.581 | 0.411 | 0.377 | 0.406 | 0.319 | 0.321 | 0.503 | 0.442 | 0.399 | 0.326 | 0.413 |
| | Avg | **0.134** | 0.147 | 0.149 | 0.262 | 0.254 | 0.266 | 0.452 | 0.338 | 0.333 | 0.370 | 0.272 | 0.273 | 0.456 | 0.347 | 0.318 | 0.277 | 0.353 |
| ETTm1 | 0.125 | **0.084** | 0.090 | 0.092 | 0.158 | 0.147 | 0.156 | 0.088 | 0.110 | 0.113 | 0.103 | 0.161 | 0.161 | 0.117 | 0.144 | 0.130 | 0.162 | 0.875 |
| | 0.25 | **0.091** | 0.098 | 0.101 | 0.164 | 0.172 | 0.174 | 0.095 | 0.129 | 0.126 | 0.121 | 0.190 | 0.190 | 0.141 | 0.175 | 0.160 | 0.190 | 0.719 |
| | 0.375 | **0.101** | 0.110 | 0.111 | 0.171 | 0.195 | 0.190 | 0.103 | 0.130 | 0.139 | 0.134 | 0.217 | 0.217 | 0.168 | 0.205 | 0.196 | 0.217 | 0.565 |
| | 0.5 | **0.110** | 0.122 | 0.124 | 0.181 | 0.225 | 0.207 | 0.119 | 0.151 | 0.153 | 0.148 | 0.246 | 0.246 | 0.194 | 0.230 | 0.251 | 0.246 | 0.504 |
| | Avg | **0.096** | 0.105 | 0.107 | 0.171 | 0.184 | 0.182 | 0.101 | 0.127 | 0.133 | 0.126 | 0.204 | 0.203 | 0.155 | 0.189 | 0.185 | 0.204 | 0.666 |
| ETTm2 | 0.125 | **0.071** | 0.078 | 0.082 | 0.174 | 0.151 | 0.165 | 0.210 | 0.230 | 0.200 | 0.310 | 0.171 | 0.171 | 0.300 | 0.199 | 0.166 | 0.171 | 0.819 |
| | 0.25 | **0.080** | 0.083 | 0.087 | 0.174 | 0.178 | 0.182 | 0.256 | 0.272 | 0.236 | 0.342 | 0.199 | 0.200 | 0.368 | 0.241 | 0.206 | 0.200 | 0.748 |
| | 0.375 | 0.091 | **0.089** | 0.092 | 0.196 | 0.203 | 0.196 | 0.341 | 0.318 | 0.272 | 0.348 | 0.225 | 0.225 | 0.406 | 0.294 | 0.244 | 0.226 | 0.630 |
| | 0.5 | 0.102 | **0.096** | 0.099 | 0.195 | 0.232 | 0.202 | 0.272 | 0.359 | 0.307 | 0.343 | 0.252 | 0.251 | 0.386 | 0.331 | 0.331 | 0.251 | 0.587 |
| | Avg | **0.086** | 0.087 | 0.090 | 0.185 | 0.191 | 0.186 | 0.270 | 0.295 | 0.254 | 0.336 | 0.212 | 0.212 | 0.365 | 0.266 | 0.237 | 0.212 | 0.696 |
| weather | 0.125 | 0.049 | 0.046 | **0.046** | 0.094 | 0.086 | 0.089 | 0.085 | 0.077 | 0.065 | 0.076 | 0.089 | 0.089 | 0.092 | 0.092 | 0.102 | 0.089 | 0.402 |
| | 0.25 | 0.054 | **0.051** | 0.052 | 0.101 | 0.105 | 0.102 | 0.067 | 0.087 | 0.075 | 0.086 | 0.108 | 0.108 | 0.094 | 0.109 | 0.133 | 0.111 | 0.334 |
| | 0.375 | **0.056** | 0.056 | 0.057 | 0.104 | 0.121 | 0.108 | 0.062 | 0.092 | 0.082 | 0.095 | 0.122 | 0.122 | 0.102 | 0.122 | 0.163 | 0.131 | 0.257 |
| | 0.5 | 0.062 | 0.064 | **0.061** | 0.110 | 0.141 | 0.114 | 0.065 | 0.105 | 0.090 | 0.100 | 0.136 | 0.137 | 0.107 | 0.138 | 0.209 | 0.152 | 0.254 |
| | Avg | 0.055 | 0.054 | **0.054** | 0.102 | 0.113 | 0.103 | 0.070 | 0.090 | 0.078 | 0.088 | 0.114 | 0.114 | 0.099 | 0.115 | 0.152 | 0.120 | 0.312 |
| Avg MAE | | **0.122** | 0.130 | 0.133 | 0.201 | 0.213 | 0.209 | 0.232 | 0.219 | 0.224 | 0.229 | 0.224 | 0.223 | 0.267 | 0.241 | 0.249 | 0.226 | 0.443 |
| Avg Rank | | **1.3** | 2.2 | 3.0 | 7.0 | 8.2 | 8.5 | 8.8 | 9.2 | 9.3 | 9.8 | 11.0 | 11.0 | 11.5 | 12.0 | 12.2 | 12.2 | 15.8 |

## M.3 ANOMALY DETECTION

The results for the anomaly detection task are summarized in the table 20, which compares the performance of various models across multiple datasets, including MSL, PSM, SMAP, SMD and SWaT. The evaluation metrics used are Accuracy (A), Recall (R), and Precision (P), with higher values indicating superior performance. Our proposed model consistently demonstrates state-of-the-art performance, achieving the highest scores in accuracy, recall, and precision on nearly all datasets. This is further validated by its overall average rank of 2.2, which is significantly better than the next best model, TimesNet, at 3.8. These findings collectively highlight the robust and superior capability of our model in effectively identifying anomalies across diverse time series data.

Table 20: Full results for the anomaly detection task. The A, R and P represent the accuracy, recall and precision (%) respectively. A higher value of A, R and P indicates a better performance.

| Model | MSL | | | PSM | | | SMAP | | | SMD | | | SWaT | | | Average | |
|---|---|---|---|---|---|---|---|---|---|---|---|---|---|---|---|---|---|
| | A | R | P | A | R | P | A | R | P | A | R | P | A | R | P | A | Rank |
| **Ours** | **97.91** | 86.85 | 92.82 | 97.83 | 93.34 | 98.30 | **97.93** | 90.62 | 93.09 | 98.76 | 76.92 | 92.01 | 97.83 | 82.92 | 98.78 | 98.05 | 2.2 |
| TimesNet | 96.47 | 75.29 | 89.55 | **98.57** | 96.30 | 98.52 | 93.62 | 56.41 | 90.01 | 98.77 | 81.54 | 87.89 | **98.15** | 93.54 | 91.45 | 97.12 | 3.8 |
| SSSD | 97.18 | 79.35 | 92.80 | 96.34 | 87.14 | 99.62 | 93.76 | 58.23 | 89.23 | **99.12** | 84.66 | 93.53 | 97.35 | 80.29 | 97.48 | 96.75 | 4.4 |
| MICN | 96.06 | 71.98 | 88.56 | 97.81 | 93.24 | 98.79 | 93.78 | 58.14 | 89.61 | 98.83 | 78.95 | 91.65 | 97.60 | 87.48 | 92.35 | 96.82 | 5.4 |
| DLinear | 96.48 | 75.30 | 89.69 | 98.17 | 94.69 | 98.65 | 93.51 | 56.38 | 88.82 | 98.71 | 75.84 | 91.65 | 96.75 | 80.36 | 91.88 | 96.72 | 7.4 |
| iTransformer | 95.00 | 62.63 | 86.15 | 96.99 | 90.26 | 98.77 | 93.62 | 56.33 | 90.07 | 98.61 | 73.45 | 91.31 | 97.47 | 86.29 | 92.39 | 96.34 | 8.2 |
| AnomalyT | 96.85 | 76.29 | 92.43 | 96.20 | 86.75 | 99.48 | 93.71 | 57.80 | 89.22 | 98.40 | 70.08 | 89.22 | 95.82 | 65.73 | 99.75 | 96.20 | 9.4 |
| LightTS | 96.33 | 74.08 | 89.32 | 96.54 | 88.54 | 98.84 | 93.56 | 55.80 | 90.12 | 98.69 | 75.41 | 91.55 | 96.25 | 69.16 | 99.93 | 96.27 | 9.4 |
| FiLM | 93.88 | 51.30 | 84.62 | 96.68 | 89.02 | 98.88 | 93.52 | 56.40 | 88.86 | 98.72 | 76.59 | 91.19 | 96.75 | 80.36 | 91.85 | 95.91 | 10.2 |
| KANAD | 96.42 | 72.82 | 91.43 | 95.15 | 83.54 | 98.81 | 93.56 | 56.31 | 89.43 | 98.71 | 76.25 | 91.27 | 95.83 | 65.73 | 99.95 | 95.93 | 10.8 |
| Reformer | 96.38 | 73.71 | 90.15 | 96.04 | 85.97 | 99.71 | 93.56 | 56.76 | 88.84 | 97.95 | 60.39 | 86.25 | 95.83 | 65.73 | 99.94 | 95.95 | 11.4 |
| Informer | 96.37 | 73.69 | 90.12 | 96.04 | 85.90 | 99.77 | 93.57 | 56.79 | 88.93 | 97.95 | 60.37 | 86.30 | 95.83 | 65.73 | 99.92 | 95.95 | 11.4 |
| Crossformer | 96.31 | 72.78 | 90.32 | 96.99 | 89.72 | 99.36 | 93.52 | 56.51 | 88.77 | 98.15 | 64.86 | 87.52 | 95.83 | 65.73 | 99.89 | 96.16 | 11.4 |
| Transformer | 96.33 | 73.66 | 89.70 | 95.24 | 83.15 | 99.62 | 94.27 | 61.37 | 90.86 | 97.80 | 65.24 | 78.32 | 95.87 | 66.10 | 99.81 | 95.90 | 11.6 |
| Autoformer | 96.63 | 75.80 | 90.71 | 94.16 | 78.95 | 99.99 | 94.38 | 62.53 | 90.65 | 97.81 | 65.10 | 78.45 | 95.81 | 65.56 | 99.96 | 95.76 | 11.8 |
| Pyraformer | 95.73 | 68.47 | 88.46 | 95.99 | 86.03 | 99.41 | 93.58 | 56.96 | 88.84 | 98.43 | 71.05 | 88.87 | 95.85 | 65.73 | 99.90 | 95.91 | 11.8 |
| FEDformer | 96.57 | 75.23 | 90.65 | 95.58 | 84.66 | 99.31 | 93.51 | 56.38 | 88.76 | 97.89 | 59.10 | 85.61 | 95.84 | 65.73 | 99.95 | 95.88 | 12.4 |
| ETSformer | 94.56 | 55.68 | 88.46 | 95.41 | 84.65 | 98.58 | 93.49 | 55.49 | 89.72 | 98.30 | 65.29 | 91.36 | 96.98 | 81.60 | 92.61 | 95.75 | 13.6 |

# N QUALITATIVE RESULTS

## N.1 FORECASTING

The forecasting visualizations in Figures 3 and 4 demonstrate the predictive capabilities of the proposed DiffKANformer model across various datasets. Figure 3 provides a comparative analysis on the ETTh1 dataset, where DiffKANformer's predictions exhibit a high degree of accuracy, closely following the ground truth values and outperforming the forecasting accuracy of other baseline models such as TimesNet, Reformer, and Nonstationary Transformer. These results collectively validate the effectiveness of the DiffKANformer architecture for long-term time series forecasting. Also shown in Figure 4, our model effectively captures the underlying patterns and trends of distinct time series, including electricity consumption (ECL, Caiso, Norpool) and weather data. The close alignment between the ground truth (green line) and the prediction (dashed orange line) in the forecasting window suggests the model's ability of forecasting different datasets accurately.

## N.2 IMPUTATION

To evaluate the effectiveness of our proposed method for imputation, we present qualitative comparisons through imputation visualizations across multiple datasets and masking scenarios. Figure 5 demonstrate the imputation performance of various time series models like DLinear, Crossformer, NonStationary Transformer on the Ettm1 dataset and Figure 6 Weather, Etth1, Etth2 and ECL at 50% and 25% mask ratios respectively. These visualizations clearly show that our method achieves superior imputation quality compared to baseline approaches, maintaining better temporal continuity and more accurately capturing the underlying patterns of the original time series.

## N.3 ANOMALY DETECTION

Figure 7 demonstrates the superior anomaly detection capabilities of our proposed method on the SMAP dataset across different baseline models including DLinear, Crossformer, and NonStationary Transformer. The visualization clearly shows that our method successfully identifies and highlights

the anomalous regions (indicated by the yellow shaded areas), while the baseline methods fail to detect these critical anomalies. This superior detection performance validates the effectiveness of our method in capturing meaningful temporal dependencies essential for accurate anomaly identification in time series data.

## O    SAMPLING TIME AND CONVERGENCE ANALYSIS

Our ablation studies (Tables 21, 22, 23) reveal critical insights into the computational overhead and convergence characteristics of KAN-based projections compared to traditional Markov diffusion models (without KAN projection).

Examining the training time complexity, KAN models exhibit substantially reduced training times across all configurations. For instance, at T=50 timesteps (Table 21), KAN requires only 295,227 seconds compared to Markov's 377,415 seconds, representing a 22% reduction in training time. This efficiency gain becomes more pronounced with increased timesteps: at T=400, KAN completes training in 171,007 seconds versus Markov's 315,110 seconds—a remarkable 46% improvement. This superlinear scaling advantage suggests that KAN's learnable activation functions converge more efficiently during the forward diffusion process, requiring fewer gradient updates to reach comparable performance.

The memory efficiency of KAN projections demonstrates consistent advantages, with peak memory consumption remaining stable around 90.2-90.3 MB across different timestep configurations, while Markov models show slight increases from 89.7 MB to 89.7 MB. This memory stability indicates that KAN's computational graph maintains bounded complexity regardless of the diffusion timestep parameter T.

The prediction horizon ablation (Table 22) reveals that KAN architectures maintain superior convergence rates even with extended prediction lengths. At Pred=720, KAN trains in 191,307 seconds versus Markov's 366,121 seconds a 48% reduction while achieving comparable test performance (40,705 vs 32,527 seconds). This indicates that KAN's learned basis functions generalize more effectively across longer temporal dependencies.

Most notably, the sequence length scaling (Table 23) demonstrates near linear computational complexity for KAN. As sequence length increases from 512 to 1024, KAN's training time scales from 184,162 to 175,398 seconds (actually decreasing due to optimization convergence), while Markov scales from 296,145 to 375,446 seconds. This sublinear scaling behavior for KAN suggests superior sample efficiency and faster convergence to optimal diffusion parameters.

These results conclusively demonstrate that KAN projections achieve faster convergence with lower computational overhead in the forward diffusion process. The consistent memory footprint, reduced training times (up to 48% improvement), and favorable scaling properties make KAN-based architectures particularly suitable for large scale diffusion models where training efficiency is paramount. The slight increase in test time for certain configurations is more than offset by the substantial training time reductions, making the overall computational budget significantly more favorable for KAN implementations.

Table 21: Computational complexity analysis for diffusion timestep (T) ablation. KAN projections demonstrate superior training efficiency with 22-46% reduction in training time compared to Markov models across all timestep configurations (T=50 to 400), while maintaining stable memory consumption ( 90.2 MB). The convergence advantage becomes more pronounced at higher timesteps, indicating KAN's efficient optimization in the forward diffusion process.

| Type | T | train time sec | test time sec | train peak mem MB | test peak mem MB |
|---|---|---|---|---|---|
| Markov | 50 | 377.4154 | 20.7349 | 89.6963 | 59.4868 |
| Markov | 100 | 349.3379 | 40.3780 | 89.6963 | 59.4868 |
| Markov | 200 | 270.5591 | 81.2419 | 89.6978 | 59.4883 |
| Markov | 400 | 315.1102 | 157.6585 | 89.7007 | 59.4912 |
| KAN | 50 | 295.2273 | 25.3280 | 90.2920 | 59.8442 |
| KAN | 100 | 169.3998 | 49.8523 | 90.2920 | 59.8442 |
| KAN | 200 | 158.0108 | 98.2846 | 90.2935 | 59.8457 |
| KAN | 400 | 171.0078 | 196.9544 | 90.2964 | 59.8486 |

Table 22: Computational complexity analysis for prediction horizon (Pred) ablation. KAN models exhibit significantly faster convergence across all prediction lengths, achieving up to 48% reduction in training time at Pred=720 (191,307s vs 366,121s). The consistent computational advantage demonstrates KAN's superior generalization capabilities for extended temporal dependencies, with stable memory footprint across varying prediction horizons.

| Type | Pred | train time sec | test time sec | train peak mem MB | test peak mem MB |
|---|---|---|---|---|---|
| Markov | 96 | 381.2194 | 42.1023 | 87.3667 | 58.1963 |
| Markov | 168 | 353.5391 | 40.4335 | 89.6963 | 59.4868 |
| Markov | 720 | 366.1210 | 32.3271 | 107.5386 | 69.3701 |
| Markov | 336 | 315.5612 | 38.7018 | 95.1255 | 62.4941 |
| KAN | 96 | 193.2493 | 51.0251 | 87.5962 | 58.3340 |
| KAN | 168 | 170.7544 | 50.0561 | 90.2920 | 59.8442 |
| KAN | 336 | 204.5689 | 47.2504 | 97.3374 | 63.8213 |
| KAN | 720 | 191.3073 | 40.7052 | 117.5020 | 75.3521 |

Table 23: Computational complexity analysis for sequence length (Seq) ablation. KAN projections show favorable scaling properties as sequence length increases from 512 to 1024, with training times remaining near-constant (184,162s to 175,398s) while Markov models exhibit substantial increases (296,145s to 375,446s). This sublinear scaling behavior indicates superior sample efficiency and faster convergence to optimal parameters, making KAN particularly suitable for long-sequence modeling tasks.

| Type | Seq | train time sec | test time sec | train peak mem MB | test peak mem MB |
|---|---|---|---|---|---|
| Markov | 512 | 296.1450 | 40.2757 | 88.7505 | 59.0869 |
| Markov | 720 | 349.0643 | 40.4680 | 89.6963 | 59.4868 |
| Markov | 1024 | 375.4464 | 40.5819 | 91.0786 | 60.0713 |
| KAN | 512 | 184.1622 | 49.0772 | 89.3462 | 59.4443 |
| KAN | 720 | 172.4618 | 49.2707 | 90.2920 | 59.8442 |
| KAN | 1024 | 175.3989 | 49.8541 | 91.6743 | 60.4287 |

## P    COMPARISION OF DiT-KAN AND DiT-MLP

The computational complexity analysis presented in Table 24 compares DiT-MLP and DiT-KAN architectures across varying sequence lengths. Our experiments reveal that DiT-KAN consistently outperforms DiT-MLP in terms of training efficiency. Specifically, at a sequence length of 512, DiT-KAN achieves a training time of 184.16 seconds compared to 209.77 seconds for DiT-MLP, representing a reduction of approximately 12%. This efficiency gap widens considerably as sequence length increases; at a sequence length of 1024, DiT-KAN requires only 175.40 seconds for training, whereas DiT-MLP demands 380.60 seconds, yielding a speedup factor of approximately 2.2×.

Notably, DiT-KAN exhibits remarkable stability in test time performance across different sequence lengths, maintaining values between 49.08 and 49.85 seconds regardless of input dimensionality. In contrast, DiT-MLP demonstrates comparable test times ranging from 40.47 to 41.69 seconds. While DiT-MLP shows marginally faster inference, the difference remains relatively modest compared to the substantial training time advantages offered by DiT-KAN.

Regarding memory consumption, both architectures exhibit similar peak memory usage during training and testing phases. DiT-KAN's training peak memory ranges from 89.35 to 91.67 MB, closely matching DiT-MLP's range of 89.32 to 91.65 MB. Similarly, test peak memory remains comparable across both methods, with differences of less than 1 MB in most configurations. These results suggest that DiT-KAN achieves its computational efficiency gains without imposing additional memory overhead, making it a practical and scalable alternative for diffusion transformer applications.

## Q    DIFFKANFORMER TO DDPM

In Appendix D, we explicitly derived each step used to obtain the loss formulation for our proposed method.

Table 24: Computational complexity analysis of DIT-KAN and DiT-MLP for various sequence lengths.

| Type | Seq | train time sec | test time sec | train peak mem MB | test peak mem MB |
|---|---|---|---|---|---|
| DiT-MLP | 512 | 209.7688 | 41.6930 | 89.3168 | 59.4267 |
| DiT-MLP | 720 | 349.0643 | 40.4680 | 89.6963 | 59.4868 |
| DiT-MLP | 1024 | 380.6010 | 41.2959 | 91.6450 | 60.4111 |
| DiT-KAN | 512 | 184.1622 | 49.0772 | 89.3462 | 59.4443 |
| DiT-KAN | 720 | 172.4618 | 49.2707 | 90.2920 | 59.8442 |
| DiT-KAN | 1024 | 175.3989 | 49.8541 | 91.6743 | 60.4287 |

From equation 5 in Appendix D.1 we can clearly see.

$$q_\phi(\mathbf{x}^t|\mathbf{x}, c) = \mathcal{N}(\mathbf{x}^t; \sqrt{\bar{\alpha}_t}KAN_\phi(\mathbf{x}, t) + (1 - \sqrt{\bar{\alpha}_t})c, (1 - \bar{\alpha}_t)I) \tag{23}$$

From (23), We can write,

$$\begin{aligned}
\mathbf{x}^t &= \sqrt{\bar{\alpha}_t}KAN_\phi(\mathbf{x}, t) + (1 - \sqrt{\bar{\alpha}_t})c + \sqrt{1 - \bar{\alpha}_t}\epsilon \\
&= \sqrt{\bar{\alpha}_t}KAN_\phi(\mathbf{x}, t) + (1 - \sqrt{\bar{\alpha}_t})c + \sqrt{1 - \alpha_t\bar{\alpha}_{t-1}}\epsilon \\
&= \sqrt{\bar{\alpha}_t}KAN_\phi(\mathbf{x}, t) + (1 - \sqrt{\bar{\alpha}_t})c + \sqrt{1 - \alpha_t}\epsilon + \sqrt{\alpha_t - \alpha_t\bar{\alpha}_{t-1}}\epsilon \\
&= \sqrt{\bar{\alpha}_t}KAN_\phi(\mathbf{x}, t) + (1 - \sqrt{\bar{\alpha}_t})c + \sqrt{1 - \alpha_t}\epsilon + \sqrt{\alpha_t}\sqrt{1 - \bar{\alpha}_{t-1}}\epsilon \\
&= \sqrt{\bar{\alpha}_t}KAN_\phi(\mathbf{x}, t) + (1 - \sqrt{\bar{\alpha}_t})c + \sqrt{1 - \alpha_t}\epsilon + \sqrt{\alpha_t}\left(\mathbf{x}^{t-1} - \sqrt{\bar{\alpha}_{t-1}}KAN_\phi(\mathbf{x}, t-1) - (1 - \sqrt{\bar{\alpha}_{t-1}})c\right) \\
&= \sqrt{\alpha_t}\mathbf{x}^{t-1} + \sqrt{\bar{\alpha}_t}\left(KAN_\phi(\mathbf{x}, t) - KAN_\phi(\mathbf{x}, t-1)\right) + (1 - \sqrt{\alpha_t})c + \sqrt{1 - \alpha_t}\epsilon
\end{aligned} \tag{24}$$

Here, if we replace KAN = 1d and c = 0 in the above equation, we get

$$\mathbf{x}^t = \sqrt{\alpha_t}\mathbf{x}^{t-1} + \sqrt{1 - \alpha_t}\epsilon \tag{25}$$

$$q_\phi(\mathbf{x}^t|\mathbf{x}, c) = \mathcal{N}(\mathbf{x}^t; \sqrt{\bar{\alpha}_t}\mathbf{x} + (1 - \bar{\alpha}_t)I) \tag{26}$$

Which is DDPM formulation.

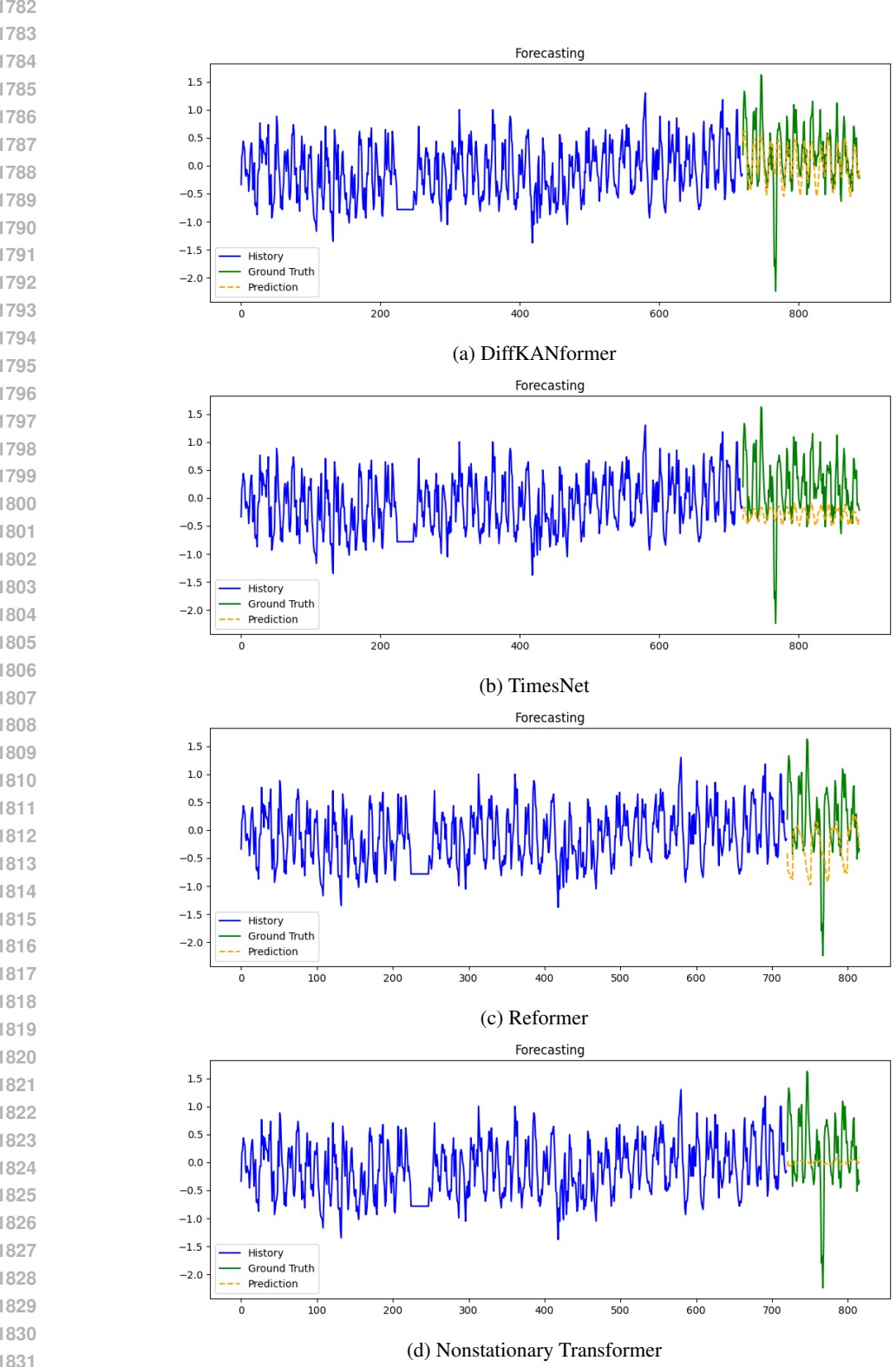

(a) DiffKANformer

(b) TimesNet

(c) Reformer

(d) Nonstationary Transformer

Figure 3: Forecasting visualizations of ETTh1 dataset predictions by (a) DiffKANformer, (b) Times-Net Wu et al. (2022), (c) Reformer Kitaev et al. (2020) and (d) Nonstationary Transformer Liu et al. (2022c)

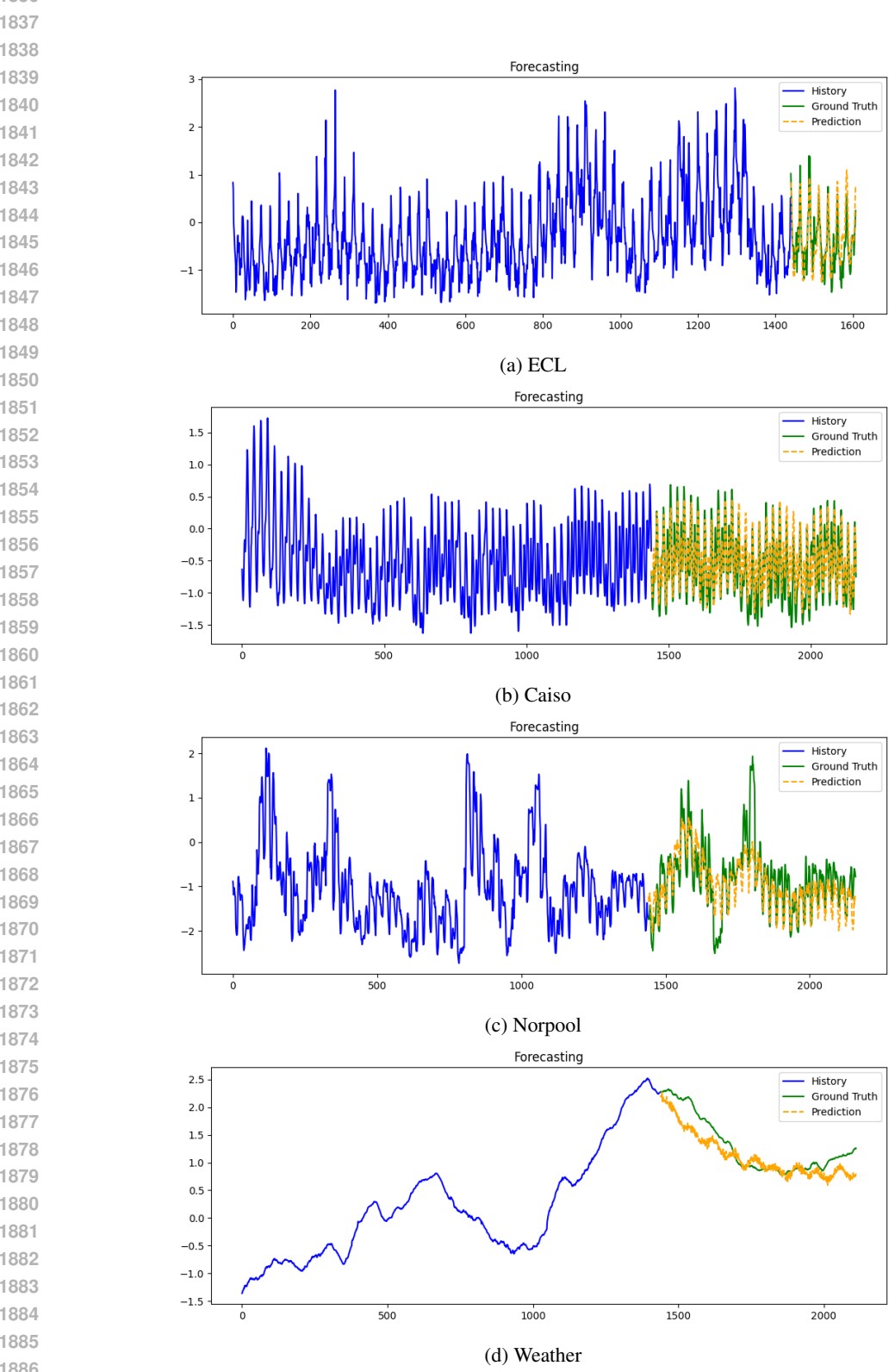

Figure 4: Forecasting visualizations results of different datasets for DiffKANformer

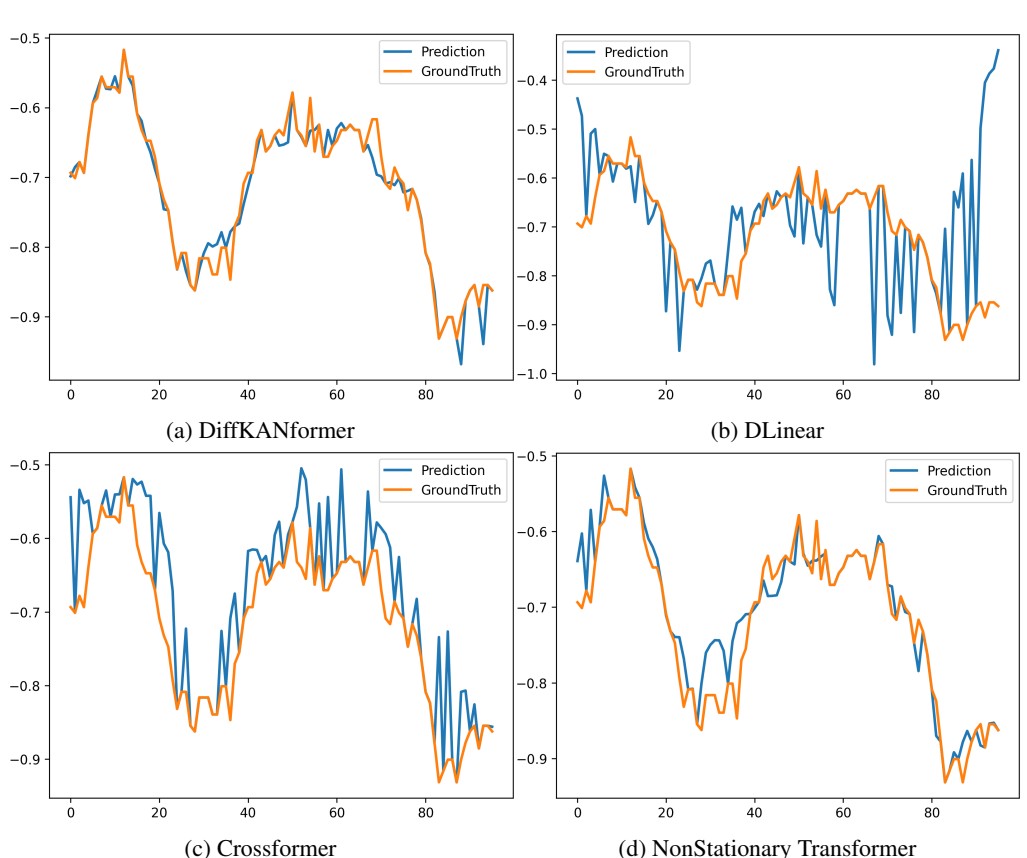

Figure 5: Imputation visualizations for ETTm1 dataset at 50% mask ratio for (a) Ours, (b) Dlinear (Zeng et al., 2023), (c) Crossformer (Zhang & Yan, 2023) and (d) NonStationary Transformer (Liu et al., 2022c)

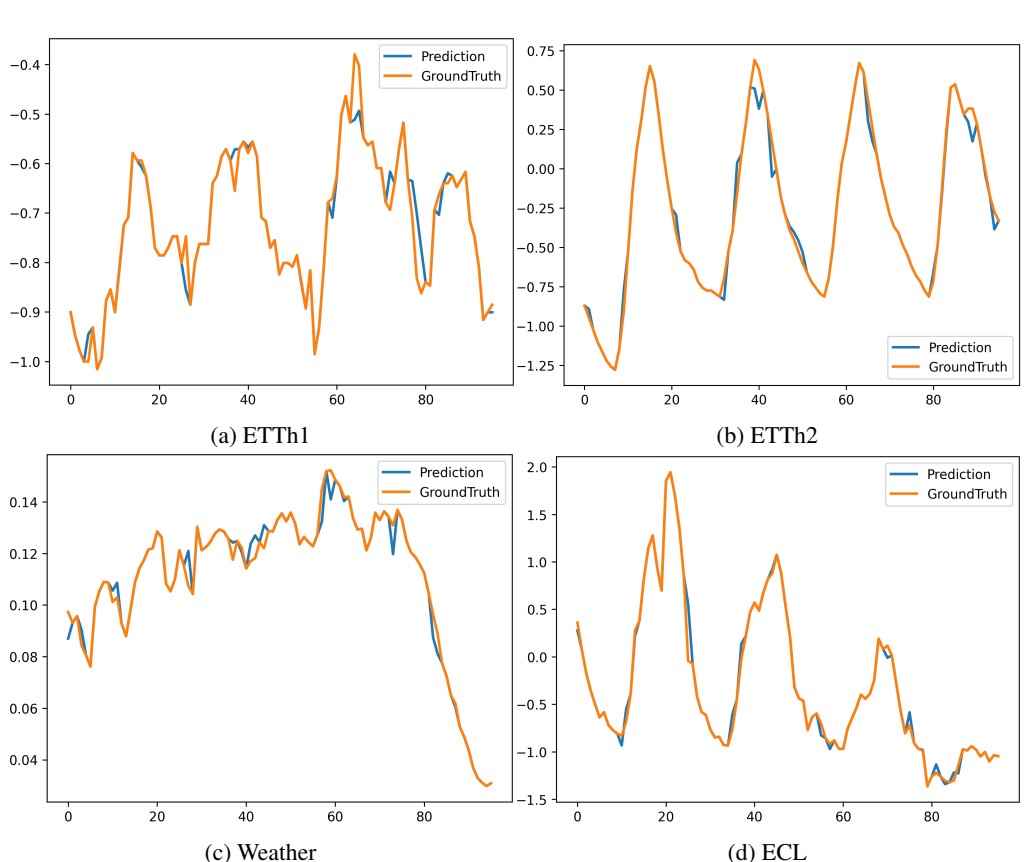

Figure 6: Imputation visualizations of DiffKANformer for ETTh1, ETTh2, Weather and ECL datasets at 25% mask ratio.

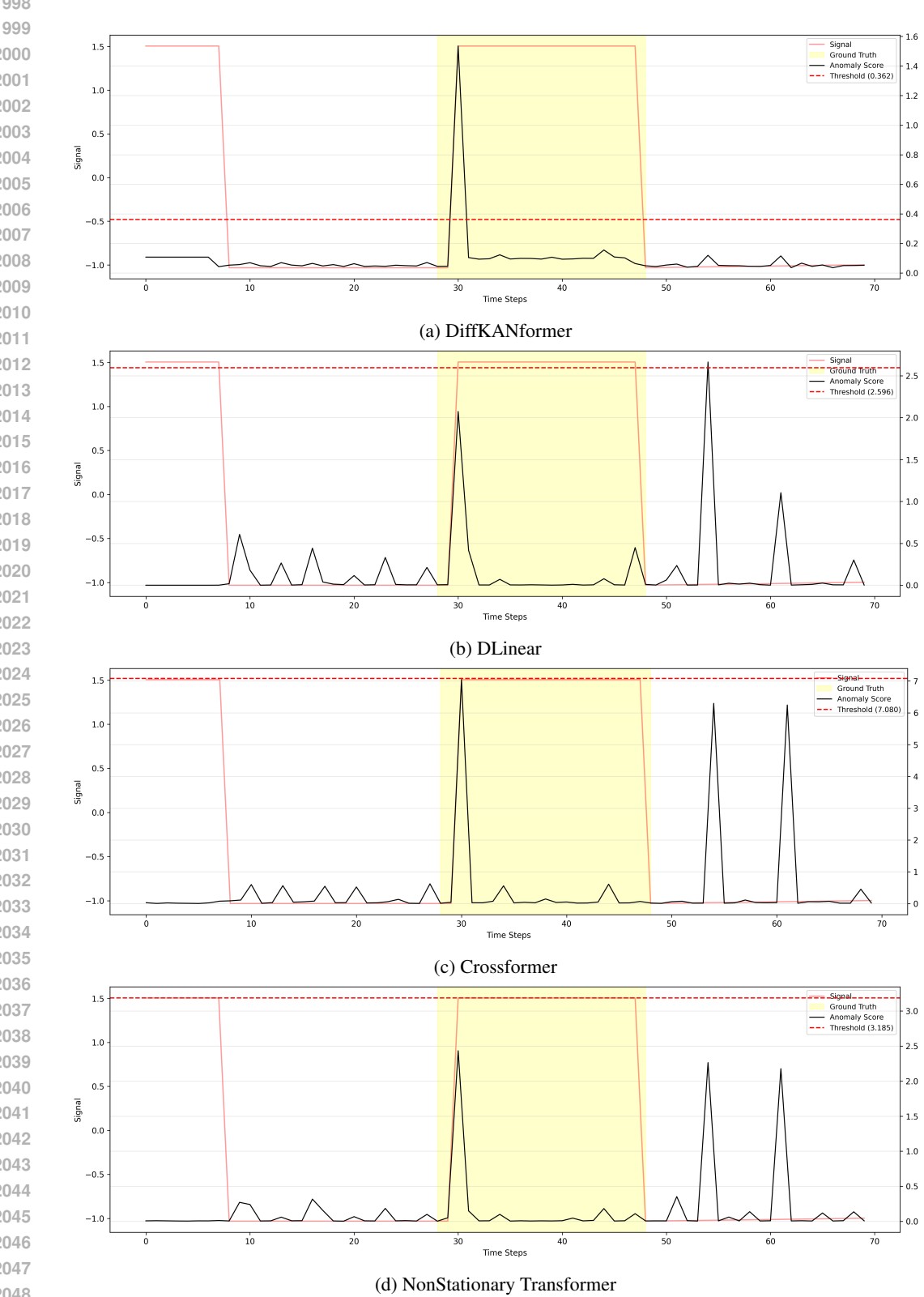

Figure 7: Anomaly Detection visualizations for SMAP dataset for (a) Ours, (b) Dlinear (Zeng et al., 2023), (c) Crossformer (Zhang & Yan, 2023) and (d) NonStationary Transformer (Liu et al., 2022c)

