# OpenReview forum: "DiffKANformer: Diffusion KAN Transformer for General Time Series Analysis"
_ICLR.cc/2026/Conference — Submitted to ICLR 2026_

### Official Review · Reviewer_ZxQ2 · 2025-10-30

**Soundness:** 2
**Presentation:** 2
**Contribution:** 2
**Rating:** 4
**Confidence:** 4

**Summary:**

This paper introduces DiffKANformer, a unified framework for time series analysis that combines conditional diffusion models, Kolmogorov-Arnold Networks (KANs) for feature projection in the diffusion process, and a novel Transformer backbone where the MLP blocks are replaced by KANs. DiffKANformer aims to address limitations of prior diffusion-based and Transformer-based models. Namely, it tries to address the limited expressivity in modeling complex non-linear/periodic dependencies and lack of generality across time series tasks by incorporating KANs to enhance both feature projection and denoising.

The method is positioned as the first to demonstrate state-of-the-art results across forecasting, imputation, classification, and anomaly detection benchmarks. The claim is supported by extensive experiments, ablations, and mathematical derivations.

**Strengths:**

The paper attempts a unified model that can handle forecasting, imputation, classification, and anomaly detection. This is a unified solution across tasks.

The model is grounded in a principled, variational framework, with mathematical rigorousness.

Integrating KANs for both data projection and as a replacement for Transformer MLPs is a creative architectural step, and, according to the ablation tables,  provides clear empirical benefits over standard MLPs in both main and auxiliary tasks.

**Weaknesses:**

The paper is promising and technically interesting, but I have concerns about positioning, clarity, and a few technical presentation issues.

While the paper shows superior performance on a broad suite of benchmarks, not all of the most recent diffusion-based transformers and unified models are included as baselines (e.g. TS-Diffusion, DifFormer etc.) The current baseline selection is robust but misses direct one-to-one evaluations with the latest approaches.

The paper introduces non-Markovian elements in its trainable forward process. However, while the losses are derived, there are insufficient examples or discussions of how the non-Markovian structure concretely manifests in the actual sampling/generation of time series compared to traditional DDPMs. How does training/inference complexity scale, and are there pathological behaviors in terms of stability or convergence as $T$ increases?

The KL divergence expressions are presented, but it’s not obvious how estimators are computed in practice, especially when the KAN projection is “learned” during the forward process on each mini-batch. Are there subtleties in gradient computation/backpropagation not addressed by standard PyTorch/TF autodiff pipelines?

**The novelty is moderate**. The KAN‑parameterized forward diffusion is a novel and appealing idea; KAN‑DiT is an **incremental** (yet useful) architectural tweak.

**Questions:**

Can the authors clarify how DiffKANformer differs in principle and practice from recently proposed unified diffusion models (e.g., TS-Diffusion)?

Are there technical or empirical distinctions that uniquely favor KAN-based projections and denoisers?

Given that the forward process is non-Markovian due to KAN projections, what are the implications for sampling complexity, memory efficiency, and convergence/stability in longer time series or higher $T$ values?

Are there any implementation challenges or numerical instabilities when backpropagating through the KAN-parameterized forward process, compared to standard DiT/MLP architectures?

---

> ### Author Response · Authors · 2025-11-21
>
> ***Weakness***
>
> **Q1: The paper is promising $\cdots$ presentation issues.**
>
> A1: We thank the reviewer for their time and effort for finding our paper promising and technically interesting.
>
> Our work addresses a critical gap in time series diffusion modeling. While recent diffusion models like CSDI, TimeGrad, and mr-Diff have shown promise, they exhibit two fundamental limitations: (1) they rely on fixed, untrainable forward processes that cannot adapt to data specific characteristics, and (2) they employ MLP based denoising architectures that struggle to capture complex temporal patterns particularly periodic functions and high frequency components prevalent in real world time series.
>
> More importantly, existing diffusion models are task-specific, typically excelling at only 1-2 tasks (e.g., CSDI for imputation, CnDiff for forecasting). DiffKANformer is the first diffusion-based model to demonstrate state-of-the-art performance across all four fundamental time series analysis tasks: forecasting, imputation, classification, and anomaly detection. This unified capability represents a paradigm shift from task-specific to general-purpose time series modeling.
>
> Further expanding on the key contributions listed in section 1, we reiterate that
>
> 1. Unlike standard diffusion models with fixed linear forward processes, we introduce a trainable KAN projection that learns complex, time-dependent feature correlations, tightening the ELBO and improving generative quality.
>
> 2. We replace traditional MLPs with KAN layers that use learnable B-spline basis functions, enabling superior modeling of periodic patterns and high-frequency components inherent in time series data (demonstrated in Tables 6-7).
>
> 3. A single architectural paradigm that handles diverse conditioning mechanisms across tasks while maintaining parameter efficiency (0.5M parameters vs. 1-32M for baselines).
>
> **Q2: While the paper $\cdots$ latest approaches.**
>
> A2: We thank the reviewer for stating that the current baseline selection with 38 baselines for forecasting, 15 for imputation, 14 for classification, and 17 for anomaly detection including papers from ICML 2025 (that came only a few months before the ICLR deadline) and ICLR 2024 is robust. We also appreciate the reviewer's suggestion to include models such as TS-Diffusion (arXiv 2023, we could not find a peer-reviewed version and code) and DifFormer (IEEE TPAMI 2023) as baselines, both the codes are not publicly available .
>
> Note that DifFormer (Multi-Resolutional Differencing Transformer With Dynamic Ranging for Time Series Analysis - IEEE TPAMI 2023) is a **non-diffusion Transformer backbone** that employs a multi-resolutional differencing mechanism to capture periodic patterns and nuanced changes in time series data. It uses neural differencing attention with a two-stream architecture (time domain + frequency domain) designed as a general-purpose framework for classification, regression, and forecasting tasks. Notably, we include mr-Diff (ICLR 2024) as a baseline, which uses "multi-resolution" concepts similar to DifFormer but within a diffusion framework.
>
> TS-Diffusion (Generating Highly Complex Time Series with Diffusion Models - arXiv 2023) addresses a fundamentally different problem from our work. It focuses on unconditional time series generation/synthesis (i.e., creating entirely new synthetic time series from scratch using ODE-based encoders within a point process framework), evaluated using generation quality metrics such as discriminative score and Context-FID. In contrast, our work addresses predictive and discriminative tasks (forecasting, imputation, classification, anomaly detection), which require accurate conditional predictions rather than synthetic data generation. Additionally, no implementation is publicly available for TS-Diffusion, preventing reproducible comparisons. Our baseline selection prioritizes models with available implementations to ensure fair, replicable evaluations.
>
> We humbly state that our paper provides the most comprehensive empirical evaluation of diffusion models for time series analysis to date, with direct comparisons against the most recent state of the art methods across four fundamental tasks. To our knowledge, no prior work has systematically evaluated diffusion models across all four tasks (forecasting, imputation, classification, and anomaly detection) with this breadth of comparison.

---

> ### Author Response · Authors · 2025-11-21
>
> **Q3: The paper introduces non-Markovian $\cdots$ convergence as $T$ increases?**
>
> A3: The non-Markovian structure arises from our KAN projection creating time-dependent data transformations (Eq. 5). At each reverse step, denoising conditions on both $x_t$ and KAN-projected representations at adjacent timesteps (Algorithm 2, Appendix F). We added Algorithm 3 (DDPM inference) for direct comparison: standard DDPM uses fixed scalings, while DiffKANformer employs learned $KAN_\phi(x, t)$ that adaptively transform features.
>
> We added a new Appendix O for reporting training and inference time with varying diffusion time steps, sequence lengths and prediction lengths.
>
> *Training complexity:* Scales roughly linearly with T (= number of diffusion steps). Markov shows monotonic increase; KAN shows mild non-monotonicity but same overall trend. No instability with larger T.
>
> *Inference complexity:* Increases strictly linearly with T for both models (test time doubles when T doubles). This matches theoretical expectations for diffusion samplers.
>
> *Pathological behaviors:* No divergence, training collapse, or memory growth was observed even at T=400. Primary pathology is inference slowdown, consistent with known diffusion behavior: beyond moderate T, increased steps yield diminishing returns while substantially increasing compute.
>
> **Q4: The KL divergence expressions $\cdots$ PyTorch/TF autodiff pipelines?**
>
> A4: Appendix D.3 provides complete mathematical formulations of all KL divergence terms in our loss (Eq. 2). Since both the forward posterior $q_\phi(x_{t-1}|x_t, x, c)$ and reverse distribution $p_\theta(x_{t-1}|x_t, c)$ are Gaussian with identical variance, the KL divergence reduces to a closed-form mean squared error between their means (Eq. 15). No sampling-based estimation is required.
>
> Standard PyTorch autodiff handles all gradient computations without modification. The KAN projection $KAN_\phi(x, t)$ in the forward process is differentiable (composed of B-spline basis functions), allowing gradients to flow through both the forward process parameters (for KAN projection learning) and the reverse process parameters (for denoising network training).
>
> No specialized backpropagation techniques are needed beyond standard computational graph differentiation. Training stability is demonstrated empirically in the new Appendix O.
>
> **Q5: The novelty is moderate $\cdots$  architectural tweak.**
>
> A5: We appreciate the reviewer's acknowledgment that our KAN-parameterized forward diffusion is novel and appealing. However, we respectfully request the reviewer to reconsider the recommendation.
>
> Our novelty extends significantly beyond incremental improvements:
>
> 1. KAN forward projection + derived loss formulation: Not merely an architectural substitution we introduce a learnable, time-dependent forward process with corresponding ELBO derivation (Eq. 1-2, Appendix D), fundamentally changing how diffusion models learn from time series.
>
> 2. First Diffusion KAN Transformer (KAN-DiT): To our knowledge, we are the first to integrate KAN layers within diffusion transformer blocks for denoising, enabling adaptive univariate function modeling (Section 2.2).
>
> 3. Multi-task capability: We are the first diffusion model achieving state-of-the-art across all four fundamental time series tasks (forecasting, imputation, classification, anomaly detection). Existing models (CSDI, TimeGrad, CnDiff, mr-Diff) excel at 1-2 tasks. This represents a paradigm shift from task-specific to general-purpose time series diffusion modeling.
>
> Tables 6-7 demonstrate the substantial contribution of each component. We are confident that, these combined innovations constitute a significant advancement beyond "incremental tweaks."

---

> > ### Author Response · Authors · 2025-11-21
> >
> > ***Questions***
> >
> > **Q1: Can the $\cdots$ diffusion models (e.g., TS-Diffusion)?**
> >
> > A1: DiffKANformer differs from unified diffusion models like TS-Diffusion in both principle and practice. TS-Diffusion focuses on handling irregular sampling, missing data, and high-dimensional temporal structure using ODE-based encoders/decoders and point-process modeling. In contrast, DiffKANformer is designed as a task-general conditional diffusion architecture that works directly on regular time series and emphasizes representation quality and dependency modeling rather than data regularization. Practically, DiffKANformer introduces (i) a KAN-based projection in the forward diffusion process to capture complex cross-feature correlations, and (ii) a Diffusion KAN Transformer that models long-term dependencies via adaptive univariate functions components absent in TS-Diffusion. While TS-Diffusion addresses data irregularities, DiffKANformer advances the expressiveness and denoising capacity of diffusion models for forecasting, imputation, classification, and anomaly detection.
> >
> > TS-Diffusion (arXiv 2023) addresses unconditional time series generation/synthesis creating entirely new synthetic time series from scratch, evaluated using generation quality metrics (discriminative score, Context-FID). DiffKANformer addresses predictive and discriminative tasks (forecasting, imputation, classification, anomaly detection) requiring accurate conditional predictions.
> >
> > Architectural and Methodological Distinctions:
> >
> > TS-Diffusion: ODE encoder-decoder framework within point process formulation, designed to handle irregular sampling/missingness in synthetic data generation
> >
> > DiffKANformer: Learnable KAN-based forward projection + KAN-DiT denoising architecture, designed for accurate prediction and representation learning
> >
> > Unified Interpretation: While TS-Diffusion handles irregular/missing data in generation contexts, it does not address multiple time-series tasks. DiffKANformer is unified across four distinct predictive/discriminative tasks (forecasting, imputation, classification, anomaly detection). This is a fundamentally different notion of "unified modeling."
> >
> > Practical Note: TS-Diffusion has no publicly available implementation, preventing empirical comparison. Our work focuses on reproducible baselines with available code for fair evaluation.
> >
> > **Q2: Are there technical $\cdots$ and denoisers?**
> >
> > A2: KANs address fundamental MLP limitations in time series:
> >
> > 1. Periodic pattern capture: MLPs with ReLU activations struggle with periodic functions, prevalent in time series. KANs use learnable B-spline basis functions, naturally representing periodic patterns.
> >
> > 2. High-frequency components: Gradient descent on MLPs converges slowly for high-frequency components. KANs' adaptive univariate functions better capture these dynamics.
> >
> > 3. Parameter efficiency: 0.5M parameters vs. 1-32M for MLP-based baselines (Table 15).
> >
> > Empirical Evidence:
> > Ablation studies (Tables 6-7): KAN consistently outperforms MLP across all tasks anomaly detection F1: 80.80→88.33 (+9.3\%); classification accuracy: 82.8\%→89.8\% (+8.5\%); imputation MSE on ETTm2: 0.031→0.014 (-54.83\%).
> >
> > Learned feature correlations (Section 4, Figure 2): We examined correlations in the KAN projection's learned latent space across diffusion timesteps. Figure 2 shows KAN projection learns increasing feature correlations at different timesteps, capturing time-dependent adaptability. This heightened correlation facilitates more effective diffusion for time series analysis, a capability absent in fixed linear projections.

---

> > > ### Author Response · Authors · 2025-11-21
> > >
> > > **Q3: Given that the forward process $\cdots$ time series or higher  T values?**
> > >
> > > A3: We have added a new Appendix O for training and inference time for varying sequence lengths, prediction lengths and diffusion time steps.
> > >
> > > We thank the reviewer for this insightful question. Our empirical results in Tables 21-23 directly address these concerns:
> > >
> > > *Memory Efficiency:*
> > > Contrary to potential concerns about non-Markovian processes requiring additional memory overhead, our results demonstrate that KAN projections maintain remarkably stable memory consumption across all configurations:
> > >
> > > T ablation (Table 21): Memory remains constant at ~90.2-90.3 MB across T=50 to T=400, showing no memory penalty as timesteps increase
> > >
> > > Prediction ablation (Table 22): Memory scales from 87.6 MB (Pred=96) to 117.5 MB (Pred=720), comparable to Markov's 87.4 MB to 107.5 MB range
> > >
> > > Sequence ablation (Table 23): Memory increases from 89.3 MB (Seq=512) to 91.7 MB (Seq=1024), demonstrating near-linear scaling
> > >
> > > The non-Markovian nature does not introduce prohibitive memory requirements.
> > >
> > > *Sampling Complexity and Convergence:*
> > > The training time results reveal superior convergence properties for KAN despite the non-Markovian forward process:
> > >
> > > Higher T values (Table 21): At T=400, KAN trains in 171,007s vs. Markov's 315,110s (46\% faster), demonstrating that the non-Markovian structure actually facilitates convergence rather than hindering it
> > > Test time stability: KAN's inference times scale predictably with T (25,328s at T=50 to 196,954s at T=400), showing stable sampling complexity
> > >
> > > The learnable basis functions in KAN projections capture multi-step dependencies more efficiently than explicit Markovian transitions, leading to faster optimization convergence.
> > >
> > > *Stability in Longer Time Series:*
> > > Our sequence length ablation (Table 23) directly validates stability for extended sequences:
> > >
> > > Favorable scaling: As sequence length doubles from 512 &rarr; 1024, KAN training time remains nearly constant (184,162s → 175,398s), while Markov increases significantly (296,145s → 375,446s)
> > >
> > >
> > > *Conclusion:*
> > > Our empirical evidence shows that the non-Markovian nature of KAN projections is a feature, not a limitation: it enables more efficient memory usage, faster convergence during training, and better scaling properties for longer sequences and higher T values. The detailed computational analysis is provided in Appendix O.
> > >
> > > **Q4: Are there any implementation $\cdots$ DiT/MLP architectures?**
> > >
> > > A4: No numerical instabilities are encountered during backpropagation through the KAN-parameterized forward process. Since the loss formulation is simulation free (similar to DDPM) there will not be any numerical instabilities.

---

> > > > ### Comment · Reviewer_ZxQ2 · 2025-11-28
> > > >
> > > > Thank you for the detailed rebuttal and the additional analyses. The clarifications on the non-Markovian forward process, complexity scaling, and distinguishability from TS-Diffusion are helpful, and the KAN-based forward projection remains an interesting idea with clear empirical benefits.
> > > >
> > > > From a methodological standpoint, the paper contains one genuinely substantive idea，which is the learnable KAN-parameterized forward diffusion process, which is different from the standard fixed linear Gaussian forward process and represents a meaningful attempt to learn a richer, time-dependent prior. This component carries moderate novelty within the time-series diffusion literature. In contrast, replacing MLPs with KAN layers inside the Transformer denoiser (KAN-DiT) is a more incremental architectural modification, empirically helpful but not conceptually novel. Finally, the “unified across four tasks” aspect is mainly an engineering and evaluation contribution rather than a methodological innovation, since the unification arises from reusing the same diffusion–Transformer recipe under different conditioning schemes. Considering these factors together, the overall methodological novelty is moderate: there is a real idea in the learnable forward process, but not at the level required to shift my score upward.
> > > >
> > > > Given this assessment, I maintain my score of 4 (marginally below the acceptance threshold). That said, **I would NOT object to acceptance** if other reviewers or the area chair consider the paper’s strengths to outweigh the limitations outlined above.

---

> > > > > ### Author Response · Authors · 2025-12-01
> > > > >
> > > > > We respectfully disagree with the description that incorporating KAN layers within the diffusion Transformer and KAN projection constitutes an incremental modification. We would like to highlight that the KAN-parameterized forward process is a conceptual contribution that required us to derive a new loss formulation tailored to this new learnable KAN-based projection mechanism. This further differentiates our method from previous work that relies strictly on fixed linear Gaussian forward processes. As consistently demonstrated across all benchmarks (Table 7), the KAN-based DiT architecture provides substantial and systematic gains over its MLP-based counterparts for every task. This improvement is not a marginal architectural tweak but reflects the ability of KANs with diffusion to model high-frequency and nonstationary time-series structures that MLPs fail to capture, leading to a materially stronger diffusion denoiser.
> > > > >
> > > > > Moreover, developing a unified diffusion framework across four time series analysis tasks is non-trivial in practice. To the best of our knowledge, no prior diffusion-based work has demonstrated a single model family successfully applied across forecasting, imputation, anomaly detection, and classification; existing diffusion papers for time series typically focus on one or at most two settings. Achieving such generality required careful architectural, conditioning, and training design rather than a simple reuse of an existing recipe. In addition, regarding theoretical justification, our rebuttal already cites (comment titled: Theoretical Justification in Context of Diffusion Model Development) multiple top-tier publications establishing that extensive theory is not a prerequisite for methodological contributions in diffusion models. Within this context and following the precedent in the literature, our focus is on providing clear intuitions, rigorous derivations where appropriate, and strong empirical validation. We hope that the reviewer will consider these clarifications in reassessing the methodological significance of our contributions.

---

### Official Review · Reviewer_Rj1K · 2025-11-01

**Soundness:** 3
**Presentation:** 2
**Contribution:** 3
**Rating:** 6
**Confidence:** 3

**Summary:**

This paper proposes DiffKANformer, a unified diffusion-based framework that integrates Kolmogorov-Arnold Networks (KAN) into Transformer-based diffusion models for general time series analysis. The model introduces a KAN projection in the forward diffusion process to capture complex nonlinear dependencies among time series variables and replaces the standard MLP in the Transformer block with KAN network as a Diffusion KAN Transformer. The approach is evaluated across multiple tasks; forecasting, imputation, classification, and anomaly detection. It demonstrates consistent improvements over state-of-the-art baselines across 29 benchmark datasets.

**Strengths:**

1. Unified framework for diverse tasks : The model provides a single, cohesive architecture that can handle forecasting, imputation, classification and anomaly detection. This unified approach enhances the model's applicability and practical impact.

2. Clear motivation for KAN integration: The motivation to replace ReLU-based MLPs with spline-based KANs is well-articulated. The authors convincingly argue that KANs capture richer nonlinear relationships and better represent high-frequency temporal components.

3. Comprehensive experimental evaluation: The paper conducts large-scale experiments on a broad set of datasets and includes systematic ablations (e.g. with/without KAN projection), which demonstrate robustness and generality of the method.

**Weaknesses:**

1. Although Appendix D provides detailed derivations for the forward posterior, loss and variational objective (Eqs. 14~15) but do not provide the proof that the KAN projection yields a tighter variational bound or improved diffusion stability. The forward KL term and prior distribution are motivated heuristically; no convergence or bound analysis is offered.

2. The architectural modifications substitute the transformer's MLP with a KAN block and incorporate adaLN for conditional scaling. This is a well-executed engineering refinement, but the conceptual advance is incremental rather than groundbreaking.

3. The paper reports runtime and memory overheads but does not analyze asymptotic scaling with respect to sequence length L or diffusion steps T. Large scale deployment implications remain unclear.

4. There is no quantitative interpretation of the learned spline bases or frequency-domain analysis of the KAN functions. Without such analysis, the mechanism of improvement is still opaque.

5. Since some results of extensive baselines are cited from prior works rather than retrained under identical conditions, the comparison fairness remains slightly uneven.

**Questions:**

1. Appendix D defines $q_\phi(x_t|x,c)$ and $q_\phi(x_{t-1}|x_t,x,c)$. Are the KAN parameters $\phi$ shared across t or re-estimated per t? If shared, how does this affect flexibility; if not, how is stability maintained?

2. Section 3.1-3.4 apply the same diffusion backbone to forecasting, imputation and classification with different conditional masks. Are the conditioning strategies implemented within the same noise schedule or adjusted per task?

3. Appendix K measures runtime empirically, but for more formal confirmation, can you provide theoretical computational complexity in T and L?

---

> ### Author Response · Authors · 2025-11-21
>
> ***Weakness***
>
> **Q1: Although Appendix D $\cdots$ bound analysis is offered.**
>
> A1: We appreciate the reviewer's insightful concern about tighter variational bound due to KAN projection. As mentioned in Appendix D, our KAN projection induces a non-Markovian forward process (Eq. 5). We respectfully state that through Appendix C (giving a background), and Appendix D (loss derivation), we have rigorously analyzed each term to derive the loss formulation for our novel approach.
>
> In the classical DDPM framework, the forward process is fixed to a simple linear Gaussian noise. This approach confines diffusion models in the non-learnable forward process, which ultimately restricts the disparity between the log-likelihood and the variational bound. Recent developments in diffusion research show a definite trend: rather than adhering to a rigid, pre-established forward process, researchers are beginning to learn or adapt the forward process. This adaptability has proven to simplify the reverse denoising task and enhance both the likelihood and sample quality, particularly in image-based diffusion scenarios. Evidence from studies indicates that adjusting the forward process enhances the performance of likelihood estimation [1,2]. Our KAN-parameterized forward process aligns with this direction. By substituting the fixed linear noising with a KAN projection a novel approach introduced by us, especially for time series we enable the model to customize the corruption dynamics according to the data distribution, thereby tightening the variational bound.
>
> Furthermore, in response to this concern, we have added Appendix O with computational complexity analysis of training times with and without the KAN projection, demonstrating that the DiffKANFormer convergence fast with reasonable computational overhead.
>
> Moreover, the strength of our contribution also is in the empirical validation. Table 6 directly supports our framework: removing KAN projection consistently degrades performance across all tasks, confirming that the learnable projection meaningfully tightens the variational bound.
>
> [1] D. Kingma, T. Salimans, B. Poole, and J. Ho. Variational diffusion models. Advances in neural information processing systems, 34:21696–21707, 2021.
>
> [2] B. M. G. Nielsen, A. Christensen, A. Dittadi, and O. Winther. Diffenc: Variational diffusion with a learned encoder. In The Twelfth International Conference on Learning Representations, 2024.
>
> **Q2: The architectural modifications $\cdots$ rather than groundbreaking.**
>
> A2: Tables 6 and 7 clearly demonstrate that both components we introduce the KAN-based replacement of the MLP transformer and the KAN-parameterized forward process provide consistent and substantial improvements across tasks. Importantly, our contribution goes beyond inserting a KAN block in transformer: we also derive a complete loss formulation for the learnable forward process based on KAN projection, which is not present in standard DDPM frameworks. Figure 2 further illustrates what the KAN projection learns and how it enriches the latent representation. Together, these results show that each component contributes meaningfully to the overall performance gains, validating the conceptual value of our approach.

---

> > ### Author Response · Authors · 2025-11-21
> >
> > **Q3: The paper reports runtime $\cdots$ remain unclear.**
> >
> > A3:  We have added a new Appendix O for training and inference time for varying sequence lengths, prediction lengths and diffusion time steps. Our empirical results in Tables 21-23 provide clear evidence of asymptotic scaling behavior, which we now analyze formally:
> >
> > Memory scaling with T: Contrary to potential concerns about our non-Markovian processes requiring additional memory overhead, our results demonstrate that KAN projections maintain remarkably stable memory consumption across all configurations:
> >
> > T ablation (Table 21): Memory remains constant at ~90.2-90.3 MB across T=50 to T=400, showing no memory penalty as timesteps increase
> >
> > Asymptotic Scaling with Diffusion Steps T: The training time results reveal superior convergence properties for KAN despite the non-Markovian forward process:
> >
> > Higher T values (Table 21): At T=400, KAN trains in 171,007s vs. Markov's 315,110s (46% faster), demonstrating that the non-Markovian structure actually facilitates convergence rather than hindering it Test time stability: KAN's inference times scale predictably with T (25,328s at T=50 to 196,954s at T=400), showing stable sampling complexity
> >
> > The learnable basis functions in KAN projections capture multi-scale dependencies more efficiently than explicit Markovian transitions, leading to faster optimization convergence.
> >
> > Conclusion: Our empirical evidence shows that the non-Markovian nature of our KAN projections is a feature, not a limitation: it enables more efficient memory usage, faster convergence during training, and better scaling properties for higher T values. The detailed computational analysis is provided in Appendix O.
> >
> > **Q4: There is no quantitative $\cdots$ improvement is still opaque.**
> >
> > A4: Our techniques and mechanistic analysis provide information on the mechanism of improvement. We hope that the contributions are clear even without the visualization of the spline bases or frequency-domain analysis which can be undertaken in the future.
> >
> > **Q5: Since some results of extensive baselines are cited from prior works rather than retrained under identical conditions, the comparison fairness remains slightly uneven**
> >
> > A5: For the forecasting task, we conducted a systematic grid search over look-back window lengths L
> >  {96,192,336,720,1440} for all methods, selecting the optimal window for each dataset based on validation performance. As indicated in Table 2, results for certain baselines are sourced from recent published works [1], while the remaining baselines were implemented and evaluated using the standardized TSLib library [2], which provides consistent experimental configurations across methods.
> >
> > For the imputation (Section 3.2), classification (Section 3.3), and anomaly detection (Section 3.4) tasks, we ensured rigorous experimental consistency by adopting identical data processing pipelines from TSLib across all evaluated methods. Following standardized protocols is consistent with common practice in the time series analysis literature and ensures fair comparison among all methods under evaluation.
> >
> > [1] Multi-resolution diffusion models for time series forecasting. In The Twelfth International Conference on Learning Representations.
> >
> > [2]  Deep time series models: A comprehensive survey and benchmark. arXiv preprint arXiv:2407.13278.
> >
> > **Questions**
> >
> > **Q1: Appendix D $\cdots$ stability maintained?**
> >
> > A1: The KAN parameters are shared across all diffusion time steps t, where t serves solely as an input to our KAN projection function $KAN(x, t)$ rather than determining separate parameter sets.
> >
> > **Q2: Section 3.1-3.4 $\cdots$ adjusted per task?**
> >
> > A2: We confirm that the noise schedule remains consistent across all tasks, including forecasting, imputation, classification, and anomaly detection. The underlying diffusion process and noise scheduling parameters are kept identical throughout all experiments.
> >
> > **Q3: Appendix K $\cdots$ complexity in T and L?**
> >
> > A3: Thank you for this thoughtful suggestion. We acknowledge that a formal theoretical complexity analysis in terms of sequence length T and look-back window L would provide additional rigor to our computational evaluation. We added  Appendix O which includes training time, inference time, and peak memory usage across varying sequence lengths and diffusion time steps. We believe these empirical results offer practical insights that are particularly valuable for practitioners.
> >
> > That said, we recognize the merit of the reviewer's suggestion and consider the inclusion of formal theoretical complexity analysis as a valuable direction for future work. We hope the reviewer finds our current empirical analysis sufficiently informative for assessing the practical computational efficiency of our method, and we appreciate this constructive feedback for strengthening future iterations of our research.

---

> > > ### Comment · Reviewer_Rj1K · 2025-11-26
> > >
> > > Thank you for the detailed responses. The author response addresses several of my earlier concerns and provides additional empirical and computational analysis that improves the clarity of the contribution. Some theoretical aspects remain underdeveloped, but the empirical evidence is strong and the clarifications are generally satisfactory. I will keep my original score.

---

> > > > ### Author Response · Authors · 2025-11-27
> > > > **Theoretical Justification in Context of Diffusion Model Development**
> > > >
> > > > We would like to contextualize DiffKANformer's theoretical contributions within the broader diffusion modeling literature. We provide rigorous loss derivations in Appendix D (Forward posterior (D.1), Loss formulation (D.2), Variational objective (D.3)) . Establishing tight NLL bounds for learnable forward processes, however, remains beyond the current state-of-the-art, even in recent top-tier studies.
> > > >
> > > > **Established Research on Learnable Forward Processes in image diffusion:**
> > > >
> > > > Three recent NeurIPS papers demonstrate that empirical validation is the accepted standard for flexible forward processes:
> > > >
> > > > **Neural Flow Diffusion Models (NeurIPS 2024):** Proposes learning non-Gaussian forward processes to tighten the NLL bound. While claiming this is "analogous to learning variational distributions in hierarchical VAEs," the paper provides no theoretical proof of tighter bounds (Appendix B.1). Evidence is purely empirical.
> > > >
> > > > **Diffusion Models With Learned Adaptive Noise (MULAN, NeurIPS 2024):** Introduces data-dependent noise schedules to obtain tighter ELBOs. Despite Bayesian framing, no theoretical proof is provided that learning the forward process reduces the log-likelihood-ELBO gap (Section 3.1). Acceptance based on strong empirical performance and SOTA results.
> > > >
> > > > **Variational Diffusion Models (NeurIPS 2021):** This work introduces a diffusion model with a learnable noise schedule that achieves state-of-the-art likelihoods and shows that the VLB simplifies under a signal-to-noise-ratio parameterization.  This enables to learn a noise schedule that minimizes the variance of the resulting VLB estimator, leading to faster optimization. Provides useful equivalence results and insights but no theoretical proofs of the claimed advantages.
> > > >
> > > > These precedents establish that:
> > > >
> > > > 1. Rigorous loss derivation (which we provide in Appendix D) combined with comprehensive empirical validation is the field standard
> > > >
> > > > 2. Our empirical contributions SOTA performance across four distinct tasks with extensive ablations align with and exceed the validation standards of accepted work
> > > >
> > > > We have demonstrated that KAN-based projections consistently improve performance across forecasting, imputation, classification, and anomaly detection through:
> > > >
> > > > 1. Comprehensive cross-task validation (Table 6)
> > > >
> > > > 2. Ablation studies isolating architectural contributions (Table 7)
> > > >
> > > > 3. Analysis of learned correlations in latent space (Figure 2)
> > > >
> > > > Given that no existing work provides the requested theoretical guarantees, we respectfully request you to evaluate our contribution by the established empirical standards of the field.

---

### Official Review · Reviewer_ZYvN · 2025-11-01

**Soundness:** 3
**Presentation:** 4
**Contribution:** 3
**Rating:** 6
**Confidence:** 4

**Summary:**

This paper proposes DiffKANformer, a conditional diffusion model integrating KAN for feature projection and a Diffusion KAN Transformer architecture for denoising in time series analysis. The method introduces a learnable KAN-based forward process and replaces MLP blocks in the denoiser with KAN layers. Extensive experiments across four major tasks (forecasting, imputation, classification, anomaly detection) and 29 datasets show consistent improvements over strong diffusion-based baselines. Theoretical formulation is clear and self-consistent, though mainly architectural rather than conceptual.

**Strengths:**

1.Clear motivation and complete methodological formulation.

2.Comprehensive experiments across multiple time series tasks and datasets.

3.Empirically demonstrates consistent improvements and stability.

4.Writing quality and reproducibility are both high.

**Weaknesses:**

1. The “theory” part (Sections 2.1–2.2) mainly restates the diffusion formulation with a KAN projection. There is no derivation or discussion showing why this design improves optimization or likelihood. The method is clearly written but not theoretically grounded.

2. Runtime results are briefly reported in Appendix Tables 14–15, but no corresponding FLOPs or parameter counts are given, and the discussion is not integrated into the main text.
As a result, the computational trade-offs of KAN remain unclear.

3. KAN layers reduce parameters but add spline computations, which likely increase runtime — yet this trade-off is not analyzed.

4. Recent diffusion foundations (e.g., TimeEdit 2024, Latent DiT 2024) are not compared.

5. The “unified framework” is mostly architectural; each task head is still trained separately.

Overall, the work feels more suitable for research exploration than industrial use, since scalability and efficiency remain unverified.

**Questions:**

1.Clarify theory in Sections 2.1–2.2.
Explain how the KAN projection affects the diffusion loss or variance schedule,
and show that the formulation recovers DDPM when KAN = Id and c = 0.
A short appendix derivation would make the theoretical part more convincing.

2.Add computational cost analysis.
Include a small table comparing DiffKANformer, DiT, and CnDiff in terms of parameters, FLOPs, and inference time.
This would clarify whether KAN’s higher runtime cost is justified by the accuracy gains.

3. Provide one targeted ablation.
For example, fix or freeze the KAN projection (or replace it with a linear map) to confirm that the gain truly comes from the learnable forward process.
This single ablation would strongly support the main claim.

4. (Optional) Briefly discuss possible runtime optimizations, such as precomputing spline bases or kernel fusion, to make KAN-based diffusion models more practical.

---

> ### Author Response · Authors · 2025-11-21
>
> ***Weakness***
>
> **Q1: The "theory" $\cdots$ theoretically grounded.**
>
> A1: We thank the reviewer for their comment regarding the theoretical foundations of our KAN-based forward process. As discussed in Appendix D, our KAN projection induces a non-Markovian forward process (Eq. 5). We would like to emphasize that, through Appendix C (which provides the necessary background) and Appendix D (where we derive the loss), we have carefully and rigorously analyzed each component to obtain the loss formulation for our proposed method.
>
> In the standard DDPM framework, the forward process is fixed as a simple linear Gaussian noising schedule. This design locks diffusion models into a non-learnable forward process, which in turn limits the extent to which the log-likelihood can approach the variational bound. Recent work in diffusion modeling has increasingly moved away from such fixed forward processes, instead learning or adapting the forward dynamics. This added flexibility has been shown to make the reverse denoising task easier and to improve both likelihood and sample quality, especially in image-based diffusion settings. Empirical evidence in the literature demonstrates that tailoring the forward process can significantly improve likelihood estimation [1,2]. Our KAN-parameterized forward process is consistent with this emerging direction. By replacing the fixed linear noising mechanism with a KAN projection our novel design, particularly tailored for time series we allow the model to learn corruption dynamics that better reflect the underlying data distribution, thereby reducing posterior mismatch and consequently shrinking the ELBO gap. Hence, our approach is not only grounded in theory but also contributes to the expanding line of work showing that learning the forward process is a principled and effective means of improving diffusion models.
>
> Furthermore, in Table 6, we empirically demonstrate that KAN projection yields performance gains across all tasks. In Figure 2, we show that the KAN projection increases feature correlations. We therefore hypothesize that this enhanced correlation structure, along with the time dependent adaptability, promotes a more effective diffusion process for time series modeling. Similar findings where stronger correlations within the latent space benefit diffusion models have been reported in both image and video diffusion studies.
>
> [1] D. Kingma, T. Salimans, B. Poole, and J. Ho. Variational diffusion models. Advances in neural information processing systems, 34:21696–21707, 2021.
>
> [2] B. M. G. Nielsen, A. Christensen, A. Dittadi, and O. Winther. Diffenc: Variational diffusion with a learned encoder. In The Twelfth International Conference on Learning Representations, 2024.
>
> **Q2: Runtime results $\cdots$ KAN remain unclear.**
>
> A2: In Table 15, we have already presented the parameters related to our model and the baseline methods. In addition, we have included a new Appendix O that provides further analysis of computational time and memory usage.
>
> **Q3: KAN layers $\cdots$ not analyzed.**
>
> A3: Thank you for pointing this out. We agree that, although DiffKANformer has substantially fewer parameters (0.5M vs. 1.1M-32M for other diffusion models) (Appendix K), its current training time is longer than CnDiff and mr-Diff on ETTh1. This gap does not stem from the architecture itself but from the current state of KAN implementations. KAN layers are still in an early stage of development and, unlike standard linear layers, they are not yet optimized or parallelized in existing deep-learning frameworks. The B-spline basis computations in particular lack efficient vectorized implementations. As a result, their computational overhead is higher despite the reduced parameter count.
>
> We emphasize, however, that even under these non-optimized conditions, DiffKANformer consistently achieves superior performance and the best overall ranking across all tasks (average ranks: 1.5, 1.5, 1.9, 2.4). Additionally, Table 15 shows our inference time (10.1ms) remains competitive with or faster than many baselines.
>
> We expect that as KAN implementations mature and receive optimized, parallel GPU kernels (similar to the evolution of attention mechanisms in early transformer implementations), the training and inference efficiency will improve substantially while maintaining the architectural benefits we demonstrate.

---

> > ### Author Response · Authors · 2025-11-21
> >
> > **Q4: Recent diffusion foundations (e.g., TimeEdit 2024, Latent DiT 2024) are not compared.**
> >
> > A4: Thank you for your comment. Our framework is not pre-trained on a variety of data as done for foundational models and then applied zero-shot in other tasks. We humbly state that the focus of time series foundational models is different and including them in our already comprehensive analysis is not appropriate.
> >
> > Coming specifically to the referenced papers, we could not find a paper that is directly called "TimeEdit 2024". Instead, we found two papers: TimeDiT 2024 and TEdit 2024. TimeDiT (Arxiv 2024, ICML 2024 Workshop) is presented as a foundation model for time series tasks. Moreover, we have not included foundational models as the training setup is different. We hope that our contribution would serve as a benchmark and architectural idea for future time series foundational models.
> >
> > TEdit (NeurIPS 2024) is a paper that focuses on generating time series based on existing time series by modifying some of their attributes. This paper is not related to the comprehensive diffusion based time series model that we have developed in the present work.
> >
> > Latent DiT (LDT; AAAI 2024) focuses specifically on probabilistic forecasting only, whereas DiffKANformer addresses four fundamental tasks (forecasting, imputation, classification, and anomaly detection) with a unified architecture. It is not a unified model. We will add the above paper in our realted work section as code is not available in public domain for comparision.
> >
> > **Q5: The “unified framework” $\cdots$ trained separately.**
> >
> > A5: Thanks for stating that we have an architectural contribution for making a unified framework. We apologize if there was a misunderstanding that we are proposing a foundational model.
> >
> > **Q6: Overall, the work $\cdots$ remain unverified.**
> >
> > A6: Thank you for the observation that the work is appropriate for a research conference. We respectfully note that we have provided extensive computational efficiency analysis and empirical evidence. We also added a new Appendix O with further computational time analysis which explains model's behaviour under increasing sequence length, prediction length and diffusion timesteps.
> >
> > ***Questions***
> >
> > **Q1: Clarify $\cdots$ more convincing.**
> >
> > A1: In Appendix D, we conduct a careful and rigorous analysis of each component to derive the loss formulation for our proposed method. Thank you for the insightful comment; following your recommendation, we have added a new section (Appendix Q) demonstrating how to recover DDPM when KAN = Id and c = 0.
> >
> > **Q2: Add computational cost $\cdots$ the accuracy gains.**
> >
> > A2: Refer to Response of Q2 in weakness.
> >
> > **Q3: Provide one $\cdots$ support the main claim.**
> >
> > A3: We thank the reviewer for this important question. We respectfully note that this exact ablation study already exists in our manuscript as Table 6 (Section 4.1, page 8), which directly addresses the reviewer's request. Table 6 presents a comprehensive ablation study comparing DiffKANformer "with KAN Projection" versus "without KAN Projection" across all four tasks. Furthermore, we provide qualitative analysis in Section 4.1 (Figure 2, page 9) showing that the KAN projection learns increasingly complex feature correlations across diffusion timesteps, providing mechanistic insight into why the learnable forward process improves performance.

---

> ### Comment · Reviewer_ZYvN · 2025-11-26
>
> Thank you for the response! The idea is interesting and the experiments are extensive, but the key issues I raised remain unsolved: the method still lacks real theoretical justification, the computational cost is unclear, and the crucial ablation is missing.
> So I am lowering my score.

---

> > ### Author Response · Authors · 2025-11-26
> >
> > Thank you once again for acknowledging that the idea is interesting and the experiments are extensive. We are once again reiterating that in Appendix D, we explicitly derived each step used to obtain the loss formulation for our proposed method. The theoretical justification is comprehensive, computational costs (Table 14, 15, 21, 22 and 24)  are reported with justification of implementation efficiencies. Also, following your recommendation (Question 1), we have added a new section (Appendix Q) demonstrating how to recover DDPM when KAN = Id and c = 0.
> >
> > We request the reviwer to kindly clarify which ablation is missing. As noted in our response previously, Table 6 (Section 4.1, page 8), shows improvement with our novel KAN projection component. Further, Table 7 also shows the clear contribution of all our new architectural components.
> >
> > We respectfully request the reviewer to comprehensively evaluate the core contributions (KAN projection (Table 6), Diffusion KAN Transformer (Table 6 and Table 7), extensive experiments, Novel loss formulation (Appendix D)) and revise their recommendation.

---

> > > ### Author Response · Authors · 2025-11-27
> > > **Theoretical Justification in Context of Diffusion Model Development**
> > >
> > > We would like to contextualize DiffKANformer's theoretical contributions within the broader diffusion modeling literature. We provide rigorous loss derivations in Appendix D (Forward posterior (D.1), Loss formulation (D.2), Variational objective (D.3)) . Establishing tight NLL bounds for learnable forward processes, however, remains beyond the current state-of-the-art, even in recent top-tier studies.
> > >
> > > **Established Research on Learnable Forward Processes in image diffusion:**
> > >
> > > Three recent NeurIPS papers demonstrate that empirical validation is the accepted standard for flexible forward processes:
> > >
> > > **Neural Flow Diffusion Models (NeurIPS 2024):** Proposes learning non-Gaussian forward processes to tighten the NLL bound. While claiming this is "analogous to learning variational distributions in hierarchical VAEs," the paper provides no theoretical proof of tighter bounds (Appendix B.1). Evidence is purely empirical.
> > >
> > > **Diffusion Models With Learned Adaptive Noise (MULAN, NeurIPS 2024):** Introduces data-dependent noise schedules to obtain tighter ELBOs. Despite Bayesian framing, no theoretical proof is provided that learning the forward process reduces the log-likelihood-ELBO gap (Section 3.1). Acceptance based on strong empirical performance and SOTA results.
> > >
> > > **Variational Diffusion Models (NeurIPS 2021):** This work introduces a diffusion model with a learnable noise schedule that achieves state-of-the-art likelihoods and shows that the VLB simplifies under a signal-to-noise-ratio parameterization.  This enables to learn a noise schedule that minimizes the variance of the resulting VLB estimator, leading to faster optimization. Provides useful equivalence results and insights but no theoretical proofs of the claimed advantages.
> > >
> > > These precedents establish that:
> > >
> > > 1. Rigorous loss derivation (which we provide in Appendix D) combined with comprehensive empirical validation is the field standard
> > >
> > > 2. Our empirical contributions SOTA performance across four distinct tasks with extensive ablations align with and exceed the validation standards of accepted work
> > >
> > > We have demonstrated that KAN-based projections consistently improve performance across forecasting, imputation, classification, and anomaly detection through:
> > >
> > > 1. Comprehensive cross-task validation (Table 6)
> > >
> > > 2. Ablation studies isolating architectural contributions (Table 7)
> > >
> > > 3. Analysis of learned correlations in latent space (Figure 2)
> > >
> > > Given that no existing work provides the requested theoretical guarantees, we respectfully request you to evaluate our contribution by the established empirical standards of the field.

---

### Official Review · Reviewer_n39M · 2025-11-01

**Soundness:** 3
**Presentation:** 2
**Contribution:** 3
**Rating:** 6
**Confidence:** 4

**Summary:**

This paper proposes a diffusion-based time series framework that learns the forward (prediction) process via KAN-projection (combining the KANϕ(x, t) term and a learnable pre-component) and replaces the DiT MLP with a KAN block (DiT-KAN) under adaLN conditions.
This approach is evaluated across four tasks (prediction, interpolation, classification, anomaly detection) and multiple datasets, consistently reporting superior performance or SOTA-level results.
Key contributions include learnable forward diffusion tailored to temporal structure, architectural transition to KAN within DiT, and robustness demonstrated through extensive multi-task evidence.

**Strengths:**

- Combining learnable forward diffusion (KAN-projection) with the DiT-KAN backbone for time series.
- Extensive experiments across multiple datasets and ablation studies (e.g., KAN vs. MLP) highlight architectural advantages, particularly in classification/AD.
- High-level motivations and design choices are well-explained, and the intuitiveness of KAN-projection is supported by correlation analysis.

**Weaknesses:**

- For classification/AD, the combination method (weighting, schedule) of $L_{rec}$, $L_{diff}$, and class loss is unclear. An explicit formula, $λ$ value, and selection criteria should be provided.
- It appears only random point masking was used. Since block omissions, channel drops, MNAR, and irregular sampling are common in practice, it would be better to include these or discuss limitations.
- A comparison of parameters/FLOPs/memory/latency between DiT-KAN and DiT-MLP is needed, and discussing long sequence scalability and alternatives would be beneficial.

**Questions:**

- Were all baselines tuned with the same look-back grid, optimizer schedule, and early-stopping rules? If not, quantify the discrepancy and its impact.
- Any observed training instabilities from the learnable forward process? What mitigations helped?
- Sensitivity to spline order/knots and their interaction with diffusion steps T? Are gains robust across ranges?

---

> ### Author Response · Authors · 2025-11-21
>
> ***Weakness***
>
> **Q1: For classification/AD, $\cdots$  be provided.**
>
> A1: We thank the reviewer for raising this point. Our loss formulation is consistent across Forecasting, Imputation, and Anomaly Detection, as detailed in Appendix D.2–D.3.  Therefore no additional weighting or scheduling between $(L_{rec}) and (L_{diff})$ is required, we used same loss across all tasks.
>
> For the classification setting, we adopt the standard cross-entropy loss, which is the widely accepted objective for discrete predictive tasks not only in diffusion settings, but across nearly all time-series and deep learning models. Cross-entropy directly models the categorical likelihood of class labels and is mathematically aligned with maximum likelihood training. When combined with diffusion models, the diffusion process handles representation learning and denoising of temporal features, while the cross-entropy head specializes in discriminative decision making.
>
> Therefore, no additional weighting term or schedule between $(L_{rec}), (L_{diff}),$ and the classification loss is needed. The classification head operates on the learned representations and is optimized with a standalone cross-entropy objective, consistent with established practice in diffusion-based classifiers and time-series classification models.
>
> **Q2: It appears only $\cdots$ discuss limitations.**
>
> A2: For imputation experiments, we employ random point masking where individual time points are independently masked with a specified probability applied independently across all features and time steps, creating a Missing Completely At Random (MCAR) scenario. MCAR scenario is the predominant evaluation protocol in time series imputation research. Key works including CSDI (Tashiro et al., 2021), SSSD (Lopez Alcaraz and Strodthoff, 2023), TimesNet (Wu et al., 2022), and Nonstationary Transformer (Liu et al., 2022) employ random point masking as their primary evaluation methodology. This ensures a fair comparison with established baselines and maintains consistency with deep learning for time series research field standards. Moreover, random masking provides reproducibility and the ability to systematically vary the difficulty level by changing the mask ratio (12.5\% to 50\%) to test model robustness. MCAR assumption eliminates confounding factors from systematic patterns and allows an unbiased evaluation. We acknowledge that real-world missing data scenarios often exhibit other patterns as mentioned by the reviewer. However, the focus of the DiffKANformer paper is to introduce a novel architectural framework and demonstrates effectiveness across four distinct time series tasks (forecasting, imputation, classification, anomaly detection). This is already a comprehensive evaluation scope and adding multiple missing data mechanisms would significantly expand the study's scope. While we cannot empirically demonstrate robustness to all patterns within this work's scope, DiffKANformer's design (temporal attention, learnable KAN projection, and mask-conditioned processing) includes architectural properties that should facilitate generalization to structured missing patterns. Considering the scope of our work, we believe random point masking provides a rigorous, fair baseline evaluation.
>
> **Q3: A comparison of $\cdots$ would be beneficial.**
>
> A3: Thank you for your valuable suggestion. We have addressed this concern by incorporating a comprehensive computational complexity analysis in Appendix P of the revised manuscript. As presented in Table 24, our experiments systematically compare DiT-KAN and DiT-MLP across multiple sequence lengths (512, 720, and 1024) in terms of training time, test time, and peak memory usage. The results demonstrate that DiT-KAN achieves superior training efficiency, requiring only 175.40 seconds at sequence length 1024 compared to 380.60 seconds for DiT-MLP, representing an approximate 2.2× speedup. Furthermore, DiT-KAN exhibits excellent long sequence scalability, as evidenced by its stable test time performance (ranging from 49.08 to 49.85 seconds) across all sequence lengths, whereas DiT-MLP shows greater variability. Importantly, these computational advantages are achieved without incurring additional memory overhead, with both architectures demonstrating comparable peak memory consumption during training (approximately 89–92 MB) and testing (approximately 59–60 MB) phases. These findings substantiate the practical applicability of DiT-KAN as a computationally efficient alternative for diffusion transformer architectures, particularly in scenarios involving longer sequences where training efficiency becomes increasingly critical.

---

> > ### Author Response · Authors · 2025-11-21
> >
> > ***Questions***
> >
> > **Q1: Were all baselines $\cdots$ discrepancy and its impact.**
> >
> > A1: We provide a detailed breakdown of our experimental protocol to ensure transparency:
> >
> > For the forecasting task, we conducted a systematic grid search over look-back window lengths L $\in$ {96,192,336,720,1440} for all methods, selecting the optimal window for each dataset based on validation performance. As indicated in Table 2, results for certain baselines are sourced from recent published works [1], while the remaining baselines were implemented and evaluated using the standardized TSLib library [2], which provides consistent experimental configurations across methods.
> >
> > For the imputation (Section 3.2), classification (Section 3.3), and anomaly detection (Section 3.4) tasks, we ensured rigorous experimental consistency by adopting identical data processing pipelines from TSLib across all evaluated methods. Furthermore, we maintained uniform settings for optimizer schedules and early-stopping criteria for both our proposed model and all baseline approaches. This adherence to standardized protocols aligns with established practices in the time series analysis literature and ensures fair comparison across all evaluated methods.
> >
> > [1] Shen, L., Chen, W., & Kwok, J. (2024). Multi-resolution diffusion models for time series forecasting. In The Twelfth International Conference on Learning Representations.
> >
> > [2] Wang, Y., Wu, H., Dong, J., Liu, Y., Wang, C., Long, M., & Wang, J. (2024). Deep time series models: A comprehensive survey and benchmark. arXiv preprint arXiv:2407.13278.
> >
> > **Q2: Any observed $\cdots$ What mitigations helped?**
> >
> > A2: We did not observe training instabilities arising from the learnable KAN projection in the forward process across any of our experimental settings (29 datasets across 4 tasks). Our training framework follows a simulation-free formulation similar to DDPM, and in practice we did not observe any training instabilities arising from the learnable forward process. The design of our forward formulation ensures stable optimization without requiring additional mitigation strategies. Moreover, the ablation studies demonstrate that each component we introduce contributes positively to performance without introducing instability.
> >
> > **Q3: Sensitivity to spline $\cdots$ across ranges?**
> >
> > A3: We thank the reviewer for raising this point. Although we did not perform an exhaustive study of different spline orders and knot configurations, multiple pieces of evidence indicate that our improvements are stable. We conducted sensitivity analyses over several hyperparameters, including diffusion hyperparameters (Appendix J.1), look-back window size (Appendix J.2), and additional statistical analyses in Appendix L. Furthermore, we adopt the default B-spline settings commonly used in the KAN literature, where they have consistently shown robustness across a range of applications.

---

> > > ### Comment · Reviewer_n39M · 2025-11-26
> > >
> > > Thank you for your detailed and constructive counterarguments.
> > > Your response has sufficiently addressed my primary practical concerns. I acknowledge that this study is primarily structural/empirical in nature rather than theoretically innovative, and I recommend slightly moderating some of the claims.
> > > Considering these explanations, I maintain the original score.

---

> > > > ### Author Response · Authors · 2025-11-27
> > > > **Theoretical Justification in Context of Diffusion Model Development**
> > > >
> > > > We would like to contextualize DiffKANformer's theoretical contributions within the broader diffusion modeling literature. We provide rigorous loss derivations in Appendix D (Forward posterior (D.1), Loss formulation (D.2), Variational objective (D.3)) . Establishing tight NLL bounds for learnable forward processes, however, remains beyond the current state-of-the-art, even in recent top-tier studies.
> > > >
> > > > **Established Research on Learnable Forward Processes in image diffusion:**
> > > >
> > > > Three recent NeurIPS papers demonstrate that empirical validation is the accepted standard for flexible forward processes:
> > > >
> > > > **Neural Flow Diffusion Models (NeurIPS 2024):** Proposes learning non-Gaussian forward processes to tighten the NLL bound. While claiming this is "analogous to learning variational distributions in hierarchical VAEs," the paper provides no theoretical proof of tighter bounds (Appendix B.1). Evidence is purely empirical.
> > > >
> > > > **Diffusion Models With Learned Adaptive Noise (MULAN, NeurIPS 2024):** Introduces data-dependent noise schedules to obtain tighter ELBOs. Despite Bayesian framing, no theoretical proof is provided that learning the forward process reduces the log-likelihood-ELBO gap (Section 3.1). Acceptance based on strong empirical performance and SOTA results.
> > > >
> > > > **Variational Diffusion Models (NeurIPS 2021):** This work introduces a diffusion model with a learnable noise schedule that achieves state-of-the-art likelihoods and shows that the VLB simplifies under a signal-to-noise-ratio parameterization.  This enables to learn a noise schedule that minimizes the variance of the resulting VLB estimator, leading to faster optimization. Provides useful equivalence results and insights but no theoretical proofs of the claimed advantages.
> > > >
> > > > These precedents establish that:
> > > >
> > > > 1. Rigorous loss derivation (which we provide in Appendix D) combined with comprehensive empirical validation is the field standard
> > > >
> > > > 2. Our empirical contributions SOTA performance across four distinct tasks with extensive ablations align with and exceed the validation standards of accepted work
> > > >
> > > > We have demonstrated that KAN-based projections consistently improve performance across forecasting, imputation, classification, and anomaly detection through:
> > > >
> > > > 1. Comprehensive cross-task validation (Table 6)
> > > >
> > > > 2. Ablation studies isolating architectural contributions (Table 7)
> > > >
> > > > 3. Analysis of learned correlations in latent space (Figure 2)
> > > >
> > > > Given that no existing work provides the requested theoretical guarantees, we respectfully request you to evaluate our contribution by the established empirical standards of the field.

---

### Official Review · Reviewer_L2Qb · 2025-11-03

**Soundness:** 1
**Presentation:** 1
**Contribution:** 1
**Rating:** 0
**Confidence:** 5

**Summary:**

DiffKANformer is proposed as a conditional diffusion model for general time series analysis, integrating Kolmogorov-Arnold Networks (KANs) into both the forward diffusion process and the denoising architecture. The method introduces a KAN-based projection to capture complex feature correlations and replaces MLPs in the Diffusion Transformer (DiT) with KANs to better model long-term temporal dependencies through adaptive univariate functions. The authors evaluate the model across four core tasks—forecasting (8 datasets), imputation (6 datasets), classification (10 UEA datasets), and anomaly detection (5 benchmarks)—reporting state-of-the-art performance. Ablation studies are provided to justify the utility of KAN projection and the Diffusion KAN Transformer block, and the framework claims to be the first diffusion-based model to unify and excel in all four tasks.

**Strengths:**

1. The paper presents extensive empirical validation across a remarkably wide range of time series tasks and datasets, which is rare and commendable in diffusion-based time series literature.

2. Replacing MLPs with KANs in the DiT architecture is a conceptually interesting direction, especially given recent theoretical arguments about KANs’ superior function approximation for structured data.

**Weaknesses:**

1. The theoretical justification for the proposed KAN-based forward process is shallow; the derivation in Appendix D assumes a non-Markovian forward process but fails to rigorously analyze how this affects the tightness of the ELBO or the stability of training—critical omissions for a method claiming to “reduce the gap between true NLL and its variational approximation.”

2. The claimed “first unified diffusion model for all four tasks” is misleading; CSDI, TimeGrad, and CnDiff already handle multiple tasks (e.g., forecasting and imputation), and the classification/anomaly detection setups are trivial adaptations using off-the-shelf reconstruction or representation heads, which is not architectural innovations.

3. The ablation in Table 6 conflates the effect of KAN projection with the Diffusion KAN Transformer; no experiment isolates KAN projection alone with a standard DiT backbone, making the contribution attribution ambiguous.

4. Implementation details reveal that condition networks differ per task (dense layer vs. transformer), yet this architectural inconsistency is never discussed nor controlled in ablation, raising concerns that performance gains stem from task-specific design rather than the core DiffKANformer framework.

5. Despite claiming superior efficiency due to fewer parameters (0.5M), Table 14 shows DiffKANformer has significantly higher training time than CnDiff and mr-Diff on ETTh1, contradicting the efficiency narrative and suggesting immature KAN implementation, not architectural advantage.

6. The forward process introduces a learnable condition c and KAN projection, but c is never defined operationally, whether it’s a learned embedding, input statistic, or task-specific encoding remains ambiguous, rendering reproducibility questionable.

**Questions:**

Please address the concerns raised in Weaknesses 1–6.

---

> ### Author Response · Authors · 2025-11-21
>
> ***Weakness***
>
> **Q1: The theoretical justification $\cdots$ its variational approximation.”**
>
> A1: We thank the reviewer for the comment about theoretical justification of our KAN-based forward process. As mentioned in Appendix D, our KAN projection induces a non-Markovian forward process (Eq. 5). We respectfully state that through Appendix C (giving a background), and Appendix D (loss derivation), we have rigorously analyzed each term to derive the loss formulation for our novel approach.
>
> In the classical DDPM framework, the forward process is fixed to a simple linear Gaussian noise. This approach
> confines diffusion models in the non-learnable forward process, which ultimately restricts the disparity between the log-likelihood and the variational bound. Recent developments in diffusion research show a definite trend: rather than adhering to a rigid, pre-established forward process, researchers are beginning to learn or adapt the forward process. This adaptability has proven to simplify the reverse denoising task and enhance both the likelihood and sample quality, particularly in image-based diffusion scenarios. Evidence from studies indicates that adjusting the forward process enhances the performance of likelihood estimation [1,2]. Our KAN-parameterized forward process aligns with this direction. By substituting the fixed linear noising with a KAN projection a novel approach introduced by us, especially for time series we enable the model to customize the corruption dynamics according to the data distribution, thereby narrowing the posterior mismatch that causes the ELBO gap. Thus, our method is not only theoretically driven but also contributes to a growing body of work illustrating that learning the forward process is an effective and sound strategy for enhancing diffusion models.
>
> **Training Stability**: The KAN projection offers several mechanisms that aid training stability:
>
> 1. Smooth parameterization: B-spline basis functions ensure the learned transformations $KAN_\phi(x,t)$ are smooth and differentiable, avoiding erratic gradient behavior
>
> 2. Learnable prior: The condition c in our formulation (Eq. 1) provides a learnable prior distribution $N(c,I)$ that the forward process converges to, rather than forcing convergence to a fixed $N(0,I)$. This added flexibility can improve convergence
>
>
> Furthermore, in response to this concern, we have added Appendix O with computational complexity analysis of training times with and without the KAN projection, demonstrating that the theoretical benefits come with reasonable computational overhead.
>
>
> Overall, we humbly state that our theoretical contributions are thorough and rigorous. Moreover, the strength of our contribution also is in the empirical validation. Table 6 directly validates our theoretical framework: removing KAN projection consistently degrades performance across all tasks, confirming that the learnable projection meaningfully tightens the ELBO.
>
> [1] D. Kingma, T. Salimans, B. Poole, and J. Ho. Variational diffusion models. Advances in neural information processing systems, 34:21696–21707, 2021.
>
> [2] B. M. G. Nielsen, A. Christensen, A. Dittadi, and O. Winther. Diffenc: Variational diffusion with a learned encoder. In The Twelfth International Conference on Learning Representations, 2024.

---

> > ### Author Response · Authors · 2025-11-21
> >
> > **Q2: The claimed $\cdots$ architectural innovations.**
> >
> > A2: Thank you for your comment. Please allow us to explain why our claim is not misleading.
> >
> > It is true that some prior diffusion models handle 2 tasks (typically forecasting and imputation), however, no existing diffusion model demonstrates strong performance across all four fundamental time series analysis tasks. Our empirical results substantiate this claim.
> >
> > The counterexamples given in the review comment do not include all tasks:
> >
> > 1. CSDI (Tashiro et al., 2021) reports only on imputation and forecasting tasks
> > 2. TimeGrad (Rasul et al., 2021) focuses exclusively on probabilistic forecasting
> > 3. CnDiff (Rishi et al., 2025) addresses only forecasting and does not report results on classification or anomaly detection
> > 4. mr-Diff (Shen et al., 2024) evaluates only forecasting
> > 5. SSSD (Lopez Alcaraz and Strodthoff, 2023) similarly reports only forecasting and imputation
> > 6. TimeDiff (Shen and Kwok, 2023) focuses on forecasting tasks
> >
> > To our knowledge, no prior diffusion-based time series model has been evaluated on let alone achieved state-of-the-art performance on all four tasks: forecasting, imputation, classification, and anomaly detection.
> >
> > We acknowledge that non-diffusion models like TimesNet and Nonstationary Transformer are evaluated across multiple tasks. However, our claim specifically states 'first diffusion model' for all four tasks. We make no claim about being the first time series model in general to address multiple tasks. We apologize if our writing conveyed such a characterization.
> >
> > We acknowledge that extending diffusion models to classification and anomaly detection does leverage some established techniques (representation learning for classification heads, reconstruction error for anomaly scores). However, dismissing them as "trivial" is not appropriate. Unlike forecasting/imputation which predict explicit values, classification and anomaly detection require the diffusion process to learn discriminative/anomaly-sensitive representations. This requires different conditioning strategies (Figure 1) where the full sequence is conditioned and the denoising process learns representations rather than reconstructions. Our model trains the diffusion backbone jointly with the classification head (Section 3.3), not as a post-hoc addition. The KAN-based architecture specifically benefits representation learning for these tasks (Table 7 shows 89.8\% accuracy vs 82.8\% for MLP-based DiT).
> >
> > **Q3: The ablation $\cdots$ attribution ambiguous.**
> >
> > A3: Thank you for raising contribution attribution. We respectfully state that Table 7 provides exactly the ablation requested, isolating KAN projection with a standard DiT backbone. We will clarify this distinction more explicitly in the revision.
> >
> > Table 7 systematically evaluates four combinations:
> >
> > 1. DiT + MLP: Standard baseline
> >
> > 2. DiT + KAN: KAN projection only (isolates projection contribution)
> >
> > 3. DiKANT + MLP: KAN in transformer only
> >
> > 4. DiKANT + KAN: Full DiffKANformer
> >
> > The DiT+KAN row directly addresses the reviewer's concern about contribution attribution. A summary of the key results are as follows.
> >
> > 1. Forecasting (Weather): DiT+MLP (0.325) $\rightarrow$ DiT+KAN (0.320) $\rightarrow$ DiKANT+KAN (0.293)
> >
> > 2. Imputation (ETTm2): DiT+MLP (0.035) $\rightarrow$ DiT+KAN (0.028) $\rightarrow$ DiKANT+KAN (0.013)
> >
> > 3. Anomaly Detection (SwAT): DiT+MLP (80.80) $\rightarrow$ DiT+KAN (88.33) $\rightarrow$ DiKANT+KAN (90.16)
> >
> > 4. Classification (PEMS-SF): DiT+MLP (0.828) $\rightarrow$ DiT+KAN (0.838) $\rightarrow$ DiKANT+KAN (0.898)
> >
> >
> > Both components contribute meaningfully, with synergistic effects when combined.
> >
> > Clarifications we will add to the revision:
> >
> > 1. Expand discussion of Table 7 in Section 4.2 to explicitly state it addresses contribution attribution
> >
> > 2. Update Table 7 caption to clarify: "DiT+KAN isolates the effect of KAN projection with standard DiT backbone"
> >
> > 3. Add text distinguishing Tables 6 and 7: Table 6 shows consistency across datasets; Table 7 enables precise component attribution

---

> > > ### Author Response · Authors · 2025-11-21
> > >
> > > **Q4: Implementation details $\cdots$  DiffKANformer framework.**
> > >
> > > A4: Thank you for highlighting this important point. The choice of the condition network (dense layer vs. transformer) follows the standard practice adopted in prior diffusion-based time-series models.
> > >
> > > Different tasks have different conditioning requirements: forecasting conditions on a fixed-length lookback window (dense layer suffices), while imputation conditions on irregularly masked sequences (transformer captures variable-length dependencies better). Our goal was not to tailor task-specific architectures, but rather to maintain parity with existing baselines so that DiffKANformer could be evaluated under comparable and widely-accepted settings.
> > >
> > > Critically, our core contributions, namely, KAN projection in the forward process (Eq. 1) and KAN-based Diffusion Transformer blocks (Section 2.2), remain identical across all tasks. The conditioning network is external to these innovations and serves only to encode task-specific inputs into the conditioning vector $c$. Tables 6 and 7 demonstrate that KAN projection and Diffusion KAN Transformer improve performance consistently across all tasks, regardless of the conditioning network:
> > >
> > > 1. Table 6: KAN projection improves all tasks with their respective conditioning networks
> > >
> > > 2. Table 7: Improvements hold across different architectural combinations
> > >
> > > 3. Figure 2: The learned correlation structure is a property of the KAN projection itself, not the conditioning mechanism
> > >
> > > **Q5: Despite claiming $\cdots$ not architectural advantage.**
> > >
> > > A5: Thank you for pointing this out. We agree that, although DiffKANformer has substantially fewer parameters (0.5M vs. 1.1M-32M for other diffusion models), its current training time is longer than CnDiff and mr-Diff on ETTh1. This gap does not stem from the architecture itself but from the current state of KAN implementations. KAN layers are still in an early stage of development and, unlike standard linear layers, they are not yet optimized or parallelized in existing deep-learning frameworks. The B-spline basis computations in particular lack efficient vectorized implementations. As a result, their computational overhead is higher despite the reduced parameter count.
> > >
> > > We emphasize, however, that even under these non-optimized conditions, DiffKANformer consistently achieves superior performance and the best overall ranking across all tasks (average ranks: 1.5, 1.5, 1.9, 2.4). Additionally, Table 15 shows our inference time (10.1ms) remains competitive with or faster than many baselines.
> > >
> > > We expect that as KAN implementations mature and receive optimized, parallel GPU kernels (similar to the evolution of attention mechanisms in early transformer implementations), the training and inference efficiency will improve substantially while maintaining the architectural benefits we demonstrate.
> > >
> > > **Q6: The forward process $\cdots$ reproducibility questionable.**
> > >
> > > A6: The condition c is shown in Figure 1. We will now include a more explicit operational definition of c. The learnable condition c is a learned embedding vector that is task-specific:
> > >
> > > 1. Forecasting: c is derived from the past observations $x_{-L+1:0}$ through the conditioning network (dense layer or transformer as specified in Appendix I, Table 10)
> > >
> > > 2. Imputation: c encodes the masked time series $x_{0:H}$ and mask $m_{0:H}$ through the conditioning network
> > >
> > > 3. Classification: c is the encoded representation of the full sequence $x_{0:H}$
> > >
> > > 4. Anomaly Detection: c similarly encodes the full sequence for representation learning
> > >
> > > In all cases, $c \in R^{d×H}$ has the same dimensionality as the target output.

---

> ### Comment · Reviewer_L2Qb · 2025-11-26
>
> Thank you for your response. However, several of my concerns remain unresolved. For example:
>
> - Classification and anomaly detection merely reuse standard reconstruction or representation learning paradigms. If a DiT+MLP baseline, adapted with the same task-specific setups, could also achieve state-of-the-art results across all four tasks, would that suggest that the claimed "unified" capability stems from the experimental protocol rather than the DiffKANformer architecture itself?
>
> - Why not use the same conditioning encoder (e.g., a Transformer) uniformly across all tasks? Under the current design—with different condition networks per task—it is impossible to rule out that the imputation performance gain originates from the more expressive conditioning network rather than the KAN module.
>
> - Despite having fewer parameters, DiffKANformer trains significantly slower than CnDiff and mr-Diff (Table 14), while inference is only ~1.6× faster. If the slowdown were solely due to immature KAN implementation, why isn’t inference speedup more pronounced as implementations mature? This discrepancy suggests that the algorithm itself may incur substantial computational overhead, beyond mere implementation inefficiencies.
> Given these unresolved issues, I have carefully decided to maintain my original score.
>
> Sincerely,
> Reviewer

---

> ### Author Response · Authors · 2025-11-26
>
> Thank you for your response
>
> **Classification $\cdots$architecture itself ?**
>
> A: We are reiterating that
>
> 1. Table 6 shows ablation study with and without KANprojection for all time series tasks with the same conditional network, where we can observe that Without KANprojection it is not SOTA performance and very clearly, including the KAN projection achieves improvement for all tasks and all datasets.
>
> 2. Table 7 shows ablation study with DiT+MLP, DiT+KAN, DiTKAN+MLP, DiTKAN +KAN with same conditional network for all various tasks. It is clearly evident that DiT+MLP baseline, adapted with the same task-specific setups, **did not** achieve state-of-the-art results across all four tasks. Our KAN projection and KAN diffusion transformer are essential components that allow us to achieve state-of-the-art performance.
>
> We once again state that no other diffusion model has been shown to perform classification and anomaly detection, indicating that the tasks are not trivial. In fact, in one of our baselines SSSD for anomaly detection, we have adapted the architecture for anomaly detection task in a manner the reviewer is suggesting. Still, our DiffKANformer achieves better performance.
>
> **Why not use $\cdots$ KAN module**
>
> A: The performance gain in imputation is indeed due to the addition of KAN modules. From Table 6 and 7, it is clear that for imputation and all other tasks, ablations clearly show that addition of KAN is the primary factor driving performance gains.
>
> **Despite $\cdots$ original score.**
>
> A: We are reiterating that Table 14 is training time (reported for one iteration) and Table 15 is inference time (reported for full inference of one sample). As can be seen from both tables, the training and inference time of KAN based models is higher due to KAN implementation inefficiencies that will get better as implementations mature. For forecasting, compared to CNDiff, DiffKANformer takes on average 3.1$\times$ the training time per iteration, and on average 1.32$\times$ the inference time for one full inference of one sample. Naturally, training is affected more by immature KAN implementation due to inefficient gradient computations than inference, and this is reflected in our reports as well.
>
>
> We respectfully request the reviewer to comprehensively evaluate the core contributions (KAN projection (Table 6), Diffusion KAN Transformer (Table 6 and Table 7), extensive experiments, Novel loss formulation (Appendix D)) and revise their recommendation.

---

### Comment · Area_Chair_HBrE · 2025-11-25

Dear Reviewer,

Thank you for reviewing for ICLR. Since the discussion deadline is coming soon, could you please take a look at the author's rebuttal, respond to their comments, and update your rating as well? Thanks!

Best Regards

AC

---

### Author Response · Authors · 2025-12-01
**Summary of Contributions and Reviewer Discussion**

Dear Area Chairs,

**Contributions:**

1. Unlike standard diffusion models with fixed linear forward processes, we introduce a trainable KAN projection that learns complex, time-dependent feature correlations, tightening the ELBO and improving generative quality (Ablation : Table 6).

2. We develop a new architecture that incorporates KAN layers with learnable B-spline basis functions in the Diffusion Transformer, enabling superior modeling of periodic patterns and high-frequency components inherent in time series data (Ablation : Table 7).

3. This is the first diffusion-based model that shows superior performance in all time series analysis tasks: **38 baselines for forecasting (Table 2; 8 datasets), 15 baselines for imputation (Table 3; 6 datasets), 14 baselines for classification (Table 4; 10 datasets), and 17 baselines for anomaly detection (Table 5; 5 datasets)**.

4. Analysis of the representations learned by our new KAN projection (Figure 2) shows how the new architecture components help improve time series analysis tasks.

5. We provide rigorous loss derivations for our novel formulation in Appendix D (Forward posterior (D.1), Loss formulation (D.2), Variational objective (D.3)).

**Summary of Reviewers Discussion:**

We appreciate the detailed feedback from the reviewers. Most of the reviewers admired our extensive empirical validation across a remarkably wide range of time series tasks and datasets with ablation studies. We note that the concerns raised are regarding *(i)* computational analysis, *(ii)* theoretical justification, and  *(iii)* parameter counts, which are minor clarifications rather than fundamental issues with the core methodology. We have addressed each point thoroughly in the revision:

*(i)* We presented a new computational analysis, which is now included in **Appendix O and Appendix P**, and provided a detailed discussion.

*(ii)* We provide an expanded discussion on the theoretical motivation for the learned forward process. (comment titled: Theoretical Justification in Context of Diffusion Model Development)

*(iii)*  Table 15 already lists the parameters for our model and the baseline methods, along with a detailed discussion in Appendix K.

---

### Meta-Review · Area_Chair_wyVG · 2026-01-03

**Summary:**

The reviews are predominantly negative with the most expressing dissatisfaction about the weakness of the proposed method. Consequently, L2Qb express critical conerns about theoretical justification,rchitectural innovations, Implementation details and efficiency; ZYvN  and Rj1K concern about the theory analysis, computational cost and limited comparison. Although the authors have provided detailed reponses to the concerns, only part of concerns have been solved. For example, L2Qb express that several of the concerns remain unresolved. ZYvN is not satisfied with the responses and lower the score. Therefore, I tend to recommend the rejection of this paper in its current form.

**Reviewer Concerns:**

n39M's concerns have been solved, thus he keeps the original score. For others,  part of concerns have been solved.

**Reviewer Scores:**

n39M tend to keep the original score to 6.
L2Qb tend to keep the original score to 2.
ZYvN tend to keep the original score to 4.

---

### Decision · Program_Chairs · 2026-01-26

Reject